# RIEMANNIAN OPTIMIZATION
# ON RELAXED INDICATOR MATRIX MANIFOLD

**Jinghui Yuan,   Fangyuan Xie,   Feiping Nie,**[*]   **Xuelong Li**
School of Artificial Intelligence, Optics and Electronics (iOPEN),
Northwestern Polytechnical University,
Xi'an 710072, P.R. China
`yuanjh@mail.nwpu.edu.cn;xiefangyuan@mail.nwpu.edu.cn`
`feipingnie@gmail.com;li@nwpu.edu.cn`

## ABSTRACT

The indicator matrix plays an important role in machine learning, but optimizing it is an NP-hard problem. We propose a new relaxation of the indicator matrix and compared with other existing relaxations, it can flexibly incorporate class information. We prove that this relaxation forms a manifold, which we call the Relaxed Indicator Matrix Manifold (RIM manifold). Based on Riemannian geometry, we develop a Riemannian toolbox for optimization on the RIM manifold. Specifically, we provide several methods of Retraction, including a fast Retraction method to obtain geodesics. We point out that the RIM manifold is a generalization of the double stochastic manifold, and it is much faster than existing methods on the double stochastic manifold, which has a complexity of $\mathcal{O}(n^3)$, while RIM manifold optimization is $\mathcal{O}(n)$ and often yields better results. We conducted extensive experiments, including image denoising, with millions of variables to support our conclusion, and applied the RIM manifold to Ratio Cut, we provide a rigorous convergence proof and achieve clustering results that outperform the state-of-the-art methods. Our Code is presented in Appendix H.

## 1   INTRODUCTION

Indicator matrices play a crucial role in machine learning (Mo et al., 2025; Li et al., 2024a; Tsitsulin et al., 2023), particularly in tasks such as clustering (Fan et al., 2022; Macgregor, 2024) and classification (Shi et al., 2024). For a problem with $n$ samples and $c$ classes, the indicator matrix $F \in \mathrm{Ind}^{n \times c}$, where $\mathrm{Ind}^{n \times c} = \{X \in \mathbb{R}^{n \times c} \mid X_{ij} \in \{0, 1\}, X1_c = 1_n\}$ and $1_c$ is the column vector of ones of size $c$. The optimization of indicator matrices, which can be seen as a 0-1 programming problem, is NP-hard (Schuetz et al., 2022; Gasse et al., 2022). Therefore, finding efficient methods to relax the indicator matrix for optimization is important.

Ng et al. (2001) relaxed the indicator matrix to the Steifel manifold, $F \in \{X \mid X^T X = I\}$, where $I$ is the identity matrix. This approach further developed spectral graph theory and led to the formulation of classic algorithms such as spectral clustering (Balestriero & LeCun, 2022; Macgregor & Sun, 2022). However, optimizing over the Steifel manifold always requires $\mathcal{O}(n^2 c)$ operations (Wen & Yin, 2013), making it challenging to scale for large datasets, and it can only provide an optimal solution for problems of the form $\mathrm{tr}(F^T LF)$, while in clustering, the resulting $F$ still needs post-processing through methods like K-means (Li et al., 2015; Mondal et al., 2021). An alternative relaxation is to make $F$ onto the single stochastic manifold, $F \in \{X \mid X1_c = 1_n, X > 0\}$ (Sun et al., 2015), which gave rise to well-known algorithms like Fuzzy K-means (Ferraro, 2024; Borlea et al., 2021). However, this approach has the drawback of not considering the total number of samples per class, which can lead to empty clusters or imbalanced class distributions (Ikotun et al., 2023; Hu et al., 2023). The most recent method is to relax the indicator matrix onto the double stochastic manifold, i.e., $F \in \{X \mid X1_c = 1_n, X^T 1_n = r, X > 0\}$ (Fettal et al., 2024; Yuan et al., 2024c). However, this approach also has significant drawbacks. The double stochastic manifold imposes overly strict requirements on the columns of $F$, as it necessitates knowing the true distribution of

---

[*]Corresponding author.

each class in the dataset as a prior, which is nearly impossible for unknown datasets. Additionally, optimization over the double stochastic manifold is extremely challenging, still requiring $\mathcal{O}(n^3)$ time (Douik & Hassibi, 2019; 2018), making it almost infeasible for large-scale datasets.

To solve above questions, we propose a new relaxation method, where $F \in \{X \mid X1_c = 1_n, l < X^T 1_n < u, X > 0\}$. In this approach, the constraints on the column sums are relaxed to lie within a specified range. This allows us to flexibly incorporate as much prior knowledge as possible into the model. When there is more prior knowledge, we can choose a tighter $(l, u)$ interval. Conversely, we can make it more relaxed. Specifically, when the column sums and the true distribution are known, we can set $l = u$ and $l$ to the true distribution (In fact, this does not lead to the absence of solutions, for further discussion, you see Appendix G). When no prior knowledge is available, we can set $l < 0$ and $u > n$, which means our relaxation is a generalization of both the single stochastic manifold and the double stochastic manifold, offering a more adaptable framework.

We prove that the set of relaxed indicator matrices forms a manifold, which we call the Relaxed Indicator Matrix Manifold (RIM manifold). Based on Riemannian geometry (Boumal, 2023; Fei et al., 2025), we have developed a Riemannian optimization toolbox (Boumal et al., 2014; Townsend et al., 2016) for running optimization on the RIM manifold. In particular, we provide three distinct Retraction methods, including one that allows for fast computation of geodesics (Nguyen, 2022; Jordan et al., 2022), enabling our algorithm to efficiently operate along the geodesic. Furthermore, we demonstrate that our algorithm, compared to existing Riemannian optimization methods on the double stochastic manifold, reduces the time complexity from $\mathcal{O}(n^3)$ to $\mathcal{O}(n)$. Furthermore, we have developed various Riemannian optimization algorithms that run on the RIM manifold.

We designed a series of large-scale experiments with millions of optimization variables to validate our algorithm. These experiments include comparisons with state-of-the-art optimization algorithms on both convex and non-convex problems like image denoising (Takemoto et al., 2022; Zhou et al., 2024). In particular, we applied the Ratio Cut model (Veldt, 2023; Hagen & Kahng, 1992) to the RIM manifold. When $l = u$, our algorithm is 70-200 times faster than those based on the double stochastic manifold for large-scale problems with millions of variables, and it achieves lower loss results. In general, the algorithms on the RIM manifold outperform the latest optimization algorithms in both loss function values and time. Additionally, the Ratio Cut clustering metric on the RIM manifold exceeds that of the latest clustering algorithms.

Overall, our contributions include:

- We propose a novel relaxation method for the indicator matrix, which allows for the full utilization of varying levels of prior information from the dataset, and we proved that the relaxed matrix forms a manifold.

- We develope a Riemannian optimization toolbox for manifolds, providing three Retraction algorithms, including a fast method for obtaining geodesics on the RIM manifold. We also demonstrated that the RIM manifold can replace methods on the double stochastic manifold, reducing the time complexity from $\mathcal{O}(n^3)$ to $\mathcal{O}(n)$.

- We conducte lots of experiments with millions of variables, demonstrating the speed and efficiency of our algorithm. Our method outperforms the double stochastic manifold by 70-200 times in large-scale experiments, yielding better results and shorter time on various problems compared to latest optimization methods. We apply the RIM manifold to Ratio Cut and achieve superior clustering performance compared to the state-of-the-art methods.

## 2 PRELIMINARIES

The Preliminaries section consists of four parts: an introduction to the notations, a brief overview of Riemannian optimization, and an introduction to the single stochastic manifold, double stochastic manifold, and Steifel manifold, as well as machine learning methods on these manifolds. All the notations used in this paper follows the standard conventions of Riemannian optimization, and important symbols are introduced in the main text. Due to space limitations, the Preliminaries can be found in Appendix B.

## 3 RIEMANNIAN TOOLBOX

### 3.1 DEFINITION OF THE RELAXED INDICATOR MATRIX MANIFOLD

The optimization of indicator matrix $F \in \text{Ind}^{n \times c}$, where typically $n \gg c$, is an NP-hard optimization problem. Three relaxation methods have already been introduced. The Steifel manifold $F \in \{X \mid X^T X = I\}$ always requires $\mathcal{O}(n^2 c)$ time complexity (Wen & Yin, 2013) and can only yield an analytical optimal solution in the form of $\text{tr}(F^T L F)$, while in clustering, the resulting $F$ still needs post-processing through methods like K-means. The single stochastic manifold $F \in \{X \mid X1_c = 1_n, X > 0\}$ does not impose any constraints on the column sums of $F$, which may lead to empty or imbalanced classes and cannot incorporate column sum information into the model. The double stochastic manifold $F \in \{X \mid X1_c = 1_n, X^T 1_n = r, X > 0\}$, on the other hand, has a time complexity of $\mathcal{O}(n^3)$, and the constraints on the column sums are too strict, often making it impossible to obtain the sum of the column. Therefore, we propose a new relaxation method:

$$F \in \{X \mid X1_c = 1_n, l < X^T 1_n < u, X > 0\} \tag{1}$$

Introducing $l$ and $u$ allows us to incorporate as much information as possible into the model. Additionally, when $l < 0$ and $u > n$, our relaxation reduces to $\{X \mid X1_c = 1_n, X > 0\}$. When $u = l = r$, our relaxation becomes $\{X \mid X1_c = 1_n, X^T 1_n = r, X > 0\}$. Thus, our relaxation generalizes the previously mentioned approaches. Importantly, our relaxation forms an embedded submanifold of the Euclidean space.

**Lemma 1.** *Our relaxed indicator matrix set $\mathcal{M} = \{X \mid X1_c = 1_n, l < X^T 1_n < u, X > 0\}$ forms an embedded submanifold of the Euclidean space, with $\dim \mathcal{M} = (n-1)c$. We refer to it as the Relaxed Indicator Matrix Manifold. The proof is included in A.1*

### 3.2 RIEMANNIAN OPTIMIZATION TOOLBOX FOR THE RIM MANIFOLD

In this section, we will establish an optimized Riemannian toolbox for the RIM manifold. To transform the embedded submanifold (Zhang et al., 2024; Lee & Lee, 2012) $\mathcal{M}$ into a Riemannian submanifold (Lee, 2018; Gulbahar, 2021), it is necessary to equip $\mathcal{M}$ with an inner product $\langle \cdot, \cdot \rangle_X$. Mishra et al. (2021) adopt the Fisher information (Ly et al., 2017; Rissanen, 1996) metric for manifolds. However, an alternative approach is to directly restrict the Euclidean inner product onto the manifold. The reason for doing so is seen in F. This restriction allows for a straightforward derivation of the Riemannian gradient (Huang & Wei, 2022) from the Euclidean gradient and the method lies in enabling an intuitive and convenient Retraction mapping.

**Lemma 2.** *By restricting the Euclidean inner product $\langle U, V \rangle = \sum_{i=1}^{n} \sum_{j=1}^{c} U_{ij} V_{ij}$ onto the RIM manifold $\mathcal{M}$, the tangent space of $\mathcal{M}$ at $X$ is given by $T_X \mathcal{M} = \{U \mid U1_c = 0\}$. For any function $\mathcal{H}$, if its Euclidean gradient is $\text{Grad}\,\mathcal{H}(F)$, the Riemannian gradient $\text{grad}\,\mathcal{H}(F)$ is expressed as following. The proof is included in A.2*

$$\text{grad}\,\mathcal{H}(F) = \text{Grad}\,\mathcal{H}(F) - \frac{1}{c}\text{Grad}\,\mathcal{H}(F)1_c 1_c^T. \tag{2}$$

To further obtain second-order information of a function, it is necessary to equip the manifold $\mathcal{M}$ with a Riemannian connection (Epstein, 1975). We select the unique connection that ensures the Riemannian Hessian $\text{hess}\,\mathcal{H}$ is symmetric and compatible with the inner product as the Riemannian connection. The following theorem formalizes this:

**Lemma 3.** *For the manifold $\mathcal{M}$, there exists a unique connection that is compatible with the inner product and ensures that the Riemannian Hessian mapping is self-adjoint. This connection is given by following. $\bar{\nabla}$ is the Riemannian connection in Euclidean space. The proof is included in A.3*

$$\nabla_V U = \bar{\nabla}_V U - \frac{1}{c}\bar{\nabla}_V U 1_c 1_c^T. \tag{3}$$

The Riemannian Hessian mapping can be directly derived from the above Riemannian connection.

**Lemma 4.** *For the manifold $\mathcal{M}$ equipped with the connection $\nabla_V U$, the Riemannian Hessian mapping satisfies following. $\text{Hess}\,\mathcal{H}$ is the Riemannian Hessian in Euclidean space. The proof is included in A.4*

$$\text{hess}\,\mathcal{H}[V] = \text{Hess}\,\mathcal{H}[V] - \frac{1}{c}\text{Hess}\,\mathcal{H}[V]1_c 1_c^T. \tag{4}$$

A Retraction (Boumal, 2023; Absil et al., 2008) is a mapping $R_X(tV)$ that maps from the tangent space of $\mathcal{M}$ at $X$ to the manifold $\mathcal{M}$, i.e., $R_X(tV) : T_X\mathcal{M} \to \mathcal{M}$. A Retraction is used to generate a curve $\gamma(t) = R_X(tV)$, starting at $X$ and moving in the initial direction given by $V$, allowing $X$ to move along the manifold. Specifically, $R_X(tV)$ should satisfy $R_X(0) = X$ and $\frac{d}{dt}R_X(tV)\big|_{t=0} = V$. If $\frac{D}{dt}\gamma'(t)\big|_{t=0} = 0$, then $\gamma(t)$ forms a geodesic, where $\frac{D}{dt}$ represents the Levi-Civita derivative (Berz, 1996). Geodesics provide better convergence guarantees for optimization algorithms on manifolds (Vishnoi, 2018). The following theorem presents a method for obtaining geodesics.

**Theorem 1.** *Let $R_X(tV) = argmin_{F \in \mathcal{M}}\|F - (X + tV)\|_F^2$, $X \in \mathcal{M}$. Then*

$$argmin_{F \in \mathcal{M}}\|F - (X + tV)\|_F^2 = \max(0, X + tV - \nu(t)1_c^T - 1_n\omega^T(t) + 1_n\rho^T(t)) \quad (5)$$

*where $\nu(t), \omega^T(t), \rho^T(t)$ are Lagrange multipliers. Moreover, there exists $\delta > 0$ such that for $t \in (0, \delta)$, $-\nu(t)1_c^T - 1_n\omega(t)^T + 1_n\rho(t)^T = 0$, and the Retraction satisfies the following. Where $\frac{D}{dt}$ denotes the Levi-Civita derivative.*

$$R_X(0) = X, \quad \frac{d}{dt}R_X(tV)\big|_{t=0} = V, \quad \frac{D}{dt}R'_X(tV)\big|_{t=0} = 0 \quad (6)$$

*Thus, $R_X(tV)$ is a geodesic. The proof is included in A.5.*

The essence of solving the Retraction is to compute the orthogonal projection $argmin_{F \in \mathcal{M}}\|F - (X + tV)\|_F^2$, which can be addressed from two perspectives: the primal problem and the dual problem.

**Theorem 2.** *$\mathcal{M} = \Omega_1 \cap \Omega_2 \cap \Omega_3$, where $\Omega_1 = \{X \mid X > 0, X1_c = 1_n\}$, $\Omega_2 = \{X \mid X^T1_n > l\}$, and $\Omega_3 = \{X \mid X^T1_n < u\}$. The primal problem can be solved using the Dykstras (Tibshirani, 2017; Boyle & Dykstra, 1986) algorithm by iteratively projecting onto $\Omega_1$, $\Omega_2$, and $\Omega_3$. Specifically:*

*$Proj_{\Omega_1}(X) = \left(X_{ij} + \eta_i\right)_+$, where $\eta$ is determined by $Proj_{\Omega_1}(X)1_c = 1_n$.*

*$Proj_{\Omega_2}(X)$ and $Proj_{\Omega_3}(X)$ are defined similarly. For example,*

$$Proj_{\Omega_2}(X^j) = \begin{cases} X^j, & if\ (X^j)^T1_n > l_j, \\ \frac{1}{n}(l_j - 1_n^TX^j)1_n + X^j, & if\ (X^j)^T1_n \le l_j, \end{cases} \quad (7)$$

*where $X^j$ is the $j$-th column of $X$, and $l_j$ is the $j$-th element of the column vector $l$. The proof is included in A.6.*

Please note that Dykstras algorithm is a projection-based retraction algorithm (Absil & Malick, 2012), which in fact seeks the best point in the set $\{X \mid X1_c = 1_n, l \le X^T1_n \le u, X \ge 0\}$. This design allows the algorithm to remain effective even when $l = u$. A detailed discussion of this can be found in Appendix G.

Another approach is the dual gradient ascent method. We have proven the following theorem.

**Theorem 3.** *Solving the primal problem is equivalent to solving the following dual problem:*

$$\max_{\omega \ge 0, \rho \ge 0} \mathcal{L} = \frac{1}{2}\|\max(0, X + tV - \nu1_c^T - 1_n\omega^T + 1_n\rho^T)\|_F^2 - \langle \nu, 1_n\rangle - \langle \omega, u\rangle + \langle \rho, l\rangle \quad (8)$$

*where $\nu$, $\omega$, and $\rho$ are Lagrange multipliers. The partial derivatives of $\mathcal{L}$ with respect to $\nu$, $\omega$, and $\rho$ are known, and gradient ascent can be used solving $\nu$, $\omega$, and $\rho$. Finally, $R_X(tV)$ can be obtained using $\max(0, X + tV - \nu1_c^T - 1_n\omega^T + 1_n\rho^T)$. The partial derivatives are following. The proof is included in A.7.*

$$\begin{cases} \frac{\partial\mathcal{L}}{\partial\nu} = \max(0, X + tV - \nu1_c^T - 1_n\omega^T + 1_n\rho^T)1_c - 1_n \\ \frac{\partial\mathcal{L}}{\partial\omega} = \max(0, X + tV - \nu1_c^T - 1_n\omega^T + 1_n\rho^T)^T1_n - u \\ \frac{\partial\mathcal{L}}{\partial\rho} = -\max(0, X + tV - \nu1_c^T - 1_n\omega^T + 1_n\rho^T)^T1_n + l \end{cases} \quad (9)$$

Additionally, we propose a Retraction method based on a variant of the Sinkhorn algorithm (Xie et al., 2025; Cuturi, 2013). This approach also attempts to map a matrix onto the RIM manifold using two diagonal matrices. The following theorem illustrates this property. However, it is equivalent to solving an optimal transport problem with an entropy regularization parameter, whose choice may not be well justified.

Table 1: Time complexity comparison $(n \gg c)$.

| Operation | RIM Manifold | | | Doubly Stochastic Manifold | | | Speedup factor |
|---|---|---|---|---|---|---|---|
| | Additions | Multiplications | Total | Additions | Multiplications | Total | |
| Riemannian Gradient | $\mathcal{O}(nc)$ | $\mathcal{O}(n)$ | $\mathcal{O}(n)$ | $\mathcal{O}(n^3)$ | $\mathcal{O}(n^3)$ | $\mathcal{O}(n^3)$ | $\mathcal{O}(n^2)$ |
| Retraction | $\mathcal{O}(nc)$ | $\mathcal{O}(nc)$ | $\mathcal{O}(nc)$ | $\mathcal{O}(nc)$ | $\mathcal{O}(nc)$ | $\mathcal{O}(nc)$ | $\mathcal{O}(1)$ |
| Riemannian Hessian | $\mathcal{O}(nc)$ | $\mathcal{O}(n)$ | $\mathcal{O}(n)$ | $\mathcal{O}(n^3)$ | $\mathcal{O}(n^3)$ | $\mathcal{O}(n^3)$ | $\mathcal{O}(n^2)$ |

**Theorem 4.** *The Sinkhorn-based Retraction is defined as*

$$R_X^s(tV) = \mathcal{S}(X \odot \exp(tV \oslash X)) = \mathrm{diag}(p^*)(X \odot \exp(tV \oslash X))\,\mathrm{diag}(q^* \odot w^*) \qquad (10)$$

*where $p^*, q^*, w^*$ are vectors, $\exp(\cdot)$ denotes element-wise exponentiation, and $\mathrm{diag}(\cdot)$ converts a vector into a diagonal matrix. The vectors $p^*, q^*, w^*$ are obtained by iteratively updating the following equations:*

$$\begin{cases} p^{(k+1)} = 1_n \oslash \left( (X \odot \exp(tV \oslash X)) \left( q^{(k)} \odot w^{(k)} \right) \right), \\ q^{(k+1)} = \max\left( l \oslash \left( (X \odot \exp(tV \oslash X))^T p^{(k+1)} \odot w^{(k)} \right), 1_c \right), \\ w^{(k+1)} = \min\left( u \oslash \left( (X \odot \exp(tV \oslash X))^T p^{(k+1)} \odot q^{(k+1)} \right), 1_c \right). \end{cases} \qquad (11)$$

*This iterative procedure ensures the mapping onto the RIM manifold. The solution $R_X^s(tV) = \mathrm{diag}(p^*)(X \odot \exp(tV \oslash X))\,\mathrm{diag}(q^* \odot w^*)$ is equivalent to solving the dual-bound optimal transport problem* (12) *with an entropy regularization parameter of 1. The proof is included in* A.8.

$$R_X^s(tV) = argmin_{F \in \mathcal{M}} \left\langle F, -\log(X \odot \exp(tV \oslash X)) \right\rangle + \delta\big|_{\delta=1} \sum_{i=1}^{n} \sum_{j=1}^{c} \left( F_{ij} \log(F_{ij}) - F_{ij} \right) \qquad (12)$$

Based on the Riemannian toolbox for the RIM manifold, we have developed Riemannian Gradient Descent (RIMRGD), Riemannian Conjugate Gradient (RIMRCG), and Riemannian Trust-Region (RIMRTR) methods on the RIM manifold. The algorithmic procedures are provided in Appendix C.

### 3.3 COMPARISON ANALYSIS OF TIME COMPLEXITY

When $u = l$, the RIM manifold reduces to the doubly stochastic manifold and provides a fast way for solving problems on the doubly stochastic constraint. Existing optimization methods on the doubly stochastic manifold are extremely time-consuming. This section provides a comparative analysis of the time complexity between the RIM manifold and the doubly stochastic manifold.

First, we discuss the Riemannian gradient. The computation of the Riemannian gradient on the RIM manifold is given by $\mathrm{grad}\,\mathcal{H}(F) = \mathrm{Grad}\,\mathcal{H}(F) - \frac{1}{c}\,\mathrm{Grad}\,\mathcal{H}(F)1_c1_c^T$. Here, the term $\mathrm{Grad}\,\mathcal{H}(F)1_c1_c^T$ involves summing each column, dividing by $c$, and then replicating it across $c$ columns. This requires $2nc$ additions and $n$ divisions.

For the doubly stochastic manifold, the Riemannian gradient is $(n = c)$:

$$\begin{cases} \mathrm{grad}\,\mathcal{H}(F) = \gamma - \left( \alpha 1_n^T + 1_n 1_n^T \gamma - 1_n \alpha^T F \right) \odot F, \\ \alpha = \left( I - FF^T \right)^\dagger \left( \gamma - F\gamma^T \right) 1_n, \quad \gamma = \mathrm{Grad}\,\mathcal{H}(F) \odot F. \end{cases} \qquad (13)$$

The term $FF^T \in \mathbb{R}^{n \times n}$, and computing its pseudo-inverse $(I - FF^T)^\dagger$ requires at least $n^3$ additions or multiplications. Further computing the Riemannian gradient involves at least $n^3$ operations. When $n \neq c$, we need to solve a linear system of $(n + c)$ dimensions still takes $\mathcal{O}(n^3)$ time (where $n \gg c$).

For the Retraction operation, the time complexity is $\mathcal{O}(nc)$, which scales linearly with the number of variables. For the computation of the Riemannian Hessian, the RIM manifold also requires only $\mathcal{O}(nc)$ additions and $\mathcal{O}(c)$ multiplications. In contrast, the Hessian mapping on the doubly stochastic manifold has a highly complex expression (186), requiring at least $\mathcal{O}(n^3)$ additions and multiplications.

We summarize the time complexity in Table 1, including the complexity of each operation and the speedup factor. We will conduct extensive experiments to verify the acceleration effect.

## 4 RIM MANIFOLD FOR GRAPH CUT

In this section, we apply the RIM manifold to graph cut problems, using Max Cut (Shinde et al., 2021; Wang et al., 2022) and Ratio Cut (Chen et al., 2022b; Nie et al., 2024) as examples. Max Cut

and Ratio Cut are both well-known graph partitioning algorithms, and their loss functions are given by $\mathcal{H}_m(F) = -tr(F^T SF)$ for the Max Cut, and $\mathcal{H}_r(F) = tr(F^T LF(F^T F)^{-1})$ for the Ratio Cut. $S$ is the similarity matrix, and $L$ is the Laplacian matrix (Nie et al., 2016; 2014). The constraint is $F \in \text{Ind}^{n \times c}$, and we relax this constraint on the RIM manifold.

First, the Euclidean gradient of $-tr(F^T SF)$ is $\text{Grad}(-tr(F^T SF)) = -SF$, and its corresponding Riemannian gradient is $\text{grad}\,\mathcal{H}_m(F) = -SF + \frac{1}{c}SF1_c1_c^T$. According to Theorem 4, the Riemannian Hessian expression is $\text{hess}\,\mathcal{H}_m[V] = \text{Hess}\,\mathcal{H}_m[V] - \frac{1}{c}\text{Hess}\,\mathcal{H}_m[V]1_c1_c^T$ Moreover, because we know that:

$$\text{Hess}\,\mathcal{H}_m[V] = \lim_{t \to 0} \frac{\text{Grad}\,\mathcal{H}_m(F + tV) - \text{Grad}\,\mathcal{H}_m(F)}{t} = \lim_{t \to 0} \frac{-S(F + tV) + SF}{t} = -SV \quad (14)$$

Therefore, we show that $\text{hess}\left(-tr(F^T SF)\right)[V]$ can be represented as following:

$$\text{hess}\left(-tr(F^T SF)\right)[V] = -SV + \frac{1}{c}SV1_c1_c^T \quad (15)$$

Now we apply the RIM manifold to the Ratio Cut problem. Ratio Cut is an important graph partitioning method with the objective function $tr(F^T LF(F^T F)^{-1})$, subject to $F \in \text{Ind}^{n \times c}$. The relaxed optimization problem is formulated as:

$$\min_{F \in \mathcal{M}} tr(F^T LF(F^T F)^{-1}), \quad \mathcal{M} = \{X \mid X1_c = 1_n, l < X^T 1_n < u, X > 0\} \quad (16)$$

The following theorem provides the expressions for the Euclidean gradient and the Euclidean Hessian map of the Ratio Cut.

**Theorem 5.** *The loss function for the Ratio Cut is given by $\mathcal{H}_r(F) = tr(F^T LF(F^T F)^{-1})$. Then, the Euclidean gradient of the loss function with respect to $F$ is following. The proof is included in A.9.*

$$Grad\mathcal{H}_r(F) = 2\left(LF(F^T F)^{-1} - F(F^T F)^{-1}(F^T LF)(F^T F)^{-1}\right) \quad (17)$$

*Given the substitutions $(F^T F)^{-1} = J$ and $F^T LF = K$, the Euclidean Hessian map for the loss function is:*

$$Hess\mathcal{H}_r[V] = 2\big(LVJ - LFJ(V^T F + F^T V)J - VJKJ + FJ(V^T F + F^T V)JKJ \quad (18)$$

$$- FJ(V^T LF + F^T LV)J + FJKJ(V^T F + F^T V)J\big) \quad (19)$$

The above theorem provides the Euclidean gradient of Ratio Cut. Although computing $(F^T F)^{-1}$ requires inversion, where $F^T F \in \mathbb{R}^{c \times c}$, the inversion complexity is only $\mathcal{O}(c^3)$ and $c \ll n$. Next, we will perform graph cut optimization on the RIM manifold, comparing the loss results and runtime with various state-of-the-art algorithms, as well as evaluating the effectiveness of graph cut for clustering.

We further prove that the graph cut problem defined on the RIM manifold is always Lipschitz continuous. More specifically, we have the following theorem.

**Theorem 6.** *For any graph cut problem expressed as $\mathcal{H}(F) = tr((F^T LF)(F^T WF)^{-1})$, where $W$ is any symmetric matrix, the Euclidean gradient $Grad\mathcal{H}(F)$ is bounded, and satisfies:*

$$\|Grad\mathcal{H}(F)\|_{\circledS} \leq 2\left(\frac{\|L\|_{\circledS}\sqrt{n}}{\alpha} + \frac{\|W\|_{\circledS}\|L\|_{\circledS}n^{3/2}}{\alpha^2}\right), \quad \alpha = \frac{\sigma_{\min}(W) \cdot l^2}{n} \quad (20)$$

*where $\sigma_{\min}(W)$ is the smallest singular value of the matrix $W$ and $\|\cdot\|_{\circledS}$ denotes the spectral norm. This implies that $\mathcal{H}(F)$ is Lipschitz continuous.*

In addition, we provide **convergence theorems for graph cut** optimization on the RIM manifold using Riemannian optimization. Specifically, in Proof A.11, we show that the graph cut problem is Lipschitz smooth and provide an analysis of its convergence rate. We also analyze the relationship between finding the optimal solution via the doubly stochastic manifold and the RIM manifold under the assumptions of strong convexity and L-smoothness. The proof can be found in Appendix A.12.

## 5 EXPERIMENTS

In this section, we will conduct extensive experiments to evaluate the performance of Riemannian optimization on the RIM manifold and address several key questions of interest.

- **Question 1:** For the RIM manifold, this paper proposes three different Retraction methods. Which method is the most efficient? Which Retraction is recommended for use?
- **Question 2:** When $l = u$, does the Riemannian optimization algorithm on the RIM manifold outperform the Riemannian optimization algorithm on the doubly stochastic manifold in terms of effectiveness and speed?
- **Question 3:** For non-convex optimization problems, we evaluate whether optimization on the RIM manifold is faster or more effective compared to other state-of-the-art methods? As examples, we consider a classic non-convex graph cut problem Ratio Cut.
- **Question 4:** When relaxing the graph cut problem onto the RIM manifold (followed by discretization), can common clustering metrics(ACC,NMI,ARI) achieve better values?

## 5.1 EXPERIMENTAL SETUPS

### 5.1.1 EXPERIMENT 1 SETUP

To determine which of the three Retraction methods is more efficient, we randomly select a large number of matrices $V \in T_X \mathcal{M}$, i.e., generate a large number of tangent vectors, and set $t = 1$. Then, we apply the three Retraction methods to generate points on the RIM manifold $\mathcal{M}$. To ensure the experiment's validity, we vary the matrix dimensions $V \in \mathbb{R}^{n \times c}$, where $n$ takes values from $\{500, 1000, 3000, 5000, 7000, 10000\}$ and $c$ takes values from $\{5, 10, 50, 100, 500, 1000\}$. The lower and upper bounds are set as $l = 0.9 \left\lfloor \frac{n}{c} \right\rfloor$ and $u = 1.1 \left\lfloor \frac{n}{c} \right\rfloor$, respectively, as well as $l = u = \frac{n}{c}$. We then calculate the computation time for the three Retraction methods and compare them. For large-scale problems, we recommend using the faster Retraction method. If the computation times are nearly identical, we recommend using the norm-based Retraction, as it yields geodesics with better properties.

### 5.1.2 EXPERIMENT 2 SETUP

To answer the second question, we need to compare Riemannian optimization methods on the RIM manifold with optimization methods on the doubly stochastic manifold under the condition $l = u$. To this end, we design two optimization problems, including both convex and non-convex cases.

The first problem is a norm approximation problem. Specifically, we randomly generate a matrix $A \in \mathbb{R}^{n \times c}$ with sizes $n \in \{5000, 7000, 10000\}$, $c \in \{5, 10, 20, 50, 70, 100\}$ and solve the following optimization problem. We compare the runtime and loss function values of the two manifolds.

$$\min_{F \in \mathcal{M}} \|F - A\|_F^2, \quad \mathcal{M} = \{X \mid X1_c = A1_c, X^T 1_n = A^T 1_c, X > 0\} \tag{21}$$

The second problem is an image denoising task based on the classical total variation (TV) regularization model. The RIM-TV model is given by

$$\begin{cases} \min_{F \in \mathcal{M}} \frac{1}{2} \|F - \tilde{A}\|_F^2 + \xi \sum_{i,j} \left( |F_{i,j+1} - F_{i,j}| + |F_{i+1,j} - F_{i,j}| \right) \\ \mathcal{M} = \{X \mid X > 0, X1_c = \tilde{A}1_c, X^T 1_n = \tilde{A}^T 1_n\} \end{cases} \tag{22}$$

Here, $\xi$ is the total variation (TV) regularization coefficient, $A$ is the original image obtained from the dataset, and $\tilde{A}$ is the noisy image generated by adding Gaussian white noise to $A$. The image $A_{ij}$ is in$(0, 1)$, $\xi$ is chosen from the set $\{0.3,0.7\}$, and the variance of the added Gaussian noise is chosen from the set $\{0.3, 0.5, 0.9\}$. We will compare the speed and objective function values of the algorithm when running on the RIM manifold versus the doubly stochastic manifold. More experimental details can be found in Appendix D.1. When comparing the RIM manifold with the doubly stochastic manifold, the only difference lies in the manifold toolbox used, all other components, including the line search method and stopping criteria, are kept exactly the same.

### 5.1.3 EXPERIMENT 3 SETUP

To answer the third question, we apply the RIM manifold to Ratio Cut and conduct experiments on 8 real datasets (as shown in Appendix D.3.4). The values of $l$ and $u$ are set as $l = u = \frac{n}{c}$ and $l = 0.9 \left\lfloor \frac{n}{c} \right\rfloor$, $u = 1.1 \left\lfloor \frac{n}{c} \right\rfloor$, respectively. For $l = u = \frac{n}{c}$, we compare seven algorithms: Riemannian Gradient Descent (RIMRGD), Riemannian Conjugate Gradient (RIMRCG), Riemannian

Table 2: Table of Execution Time when $l = u$ for Different Retraction Algorithms($s$)

| Row&Col | Dual | | | | | | Sinkhorn | | | | | | Dykstras | | | | | |
|---|---|---|---|---|---|---|---|---|---|---|---|---|---|---|---|---|---|---|
| | 500 | 1000 | 3000 | 5000 | 7000 | 10000 | 500 | 1000 | 3000 | 5000 | 7000 | 10000 | 500 | 1000 | 3000 | 5000 | 7000 | 10000 |
| 5 | 0.012 | 0.024 | 0.054 | 0.084 | 0.108 | 0.140 | **0.004** | 0.009 | 0.048 | 0.132 | 0.169 | 0.499 | 0.006 | **0.007** | **0.011** | **0.019** | **0.028** | **0.038** |
| 10 | 0.022 | 0.036 | 0.075 | 0.112 | 0.141 | 0.188 | **0.002** | 0.006 | 0.036 | 0.087 | 0.166 | 0.343 | 0.006 | **0.005** | **0.014** | **0.023** | **0.031** | **0.043** |
| 50 | 0.074 | 0.095 | 0.791 | 1.307 | 1.886 | 2.766 | **0.002** | **0.008** | 0.043 | 0.125 | 0.228 | 0.474 | 0.005 | **0.008** | **0.023** | **0.039** | **0.053** | **0.074** |
| 100 | 0.012 | 0.174 | 1.597 | 2.962 | 3.831 | 5.710 | **0.003** | **0.008** | 0.056 | 0.140 | 0.288 | 0.580 | 0.006 | 0.010 | **0.031** | **0.060** | **0.072** | **0.106** |
| 500 | 0.054 | 0.122 | 8.597 | 14.32 | 20.16 | 23.77 | **0.013** | **0.030** | 0.237 | 0.629 | 1.155 | 2.265 | 0.016 | 0.033 | **0.096** | **0.168** | **0.223** | **0.318** |
| 1000 | 0.102 | 0.178 | 17.26 | 28.56 | 40.55 | 56.56 | **0.034** | 0.082 | 0.446 | 1.038 | 1.931 | 3.614 | **0.034** | **0.067** | **0.219** | **0.384** | **0.556** | **0.789** |

Trust Region (RIMRTR), Frank-Wolfe Algorithm (FWA) (Jaggi, 2013; Weber & Sra, 2023; Yurtsever & Sra, 2022), Projected Gradient Descent (PGD) (Shen & Chen, 2023; Chen & Wainwright, 2015), Riemannian Gradient Descent on the Double Stochastic Manifold (DSRGD) (Tripuraneni et al., 2018), and Riemannian Conjugate Gradient on the Double Stochastic Manifold (DSRCG) (Sato, 2022). For $l = 0.9 \left\lfloor \frac{n}{c} \right\rfloor$ and $u = 1.1 \left\lfloor \frac{n}{c} \right\rfloor$, we only compare RIMRGD, RIMRCG, RIMRTR, FWA, and PGD. The optimization results of these algorithms are then compared. More experimental details can be found in Appendix D.2.

### 5.1.4 EXPERIMENT 4 SETUP

To answer the fourth question, we compare the Ratio Cut algorithm on the RIM manifold with ten clustering algorithms. We again choose 8 real datasets with different types, including images, tables, waveforms, etc. (as shown in Appendix D.3.4), and conduct large-scale validation using 10 comparison algorithms (listed in D.3). We evaluate the clustering performance using three metrics: clustering accuracy (ACC) (Yuan et al., 2024a;b), normalized mutual information (NMI) (Ren et al., 2024), and adjusted Rand index (ARI) (Ronen et al., 2022). For the similarity matrix, we use the k-nearest neighbor (k-NN) (Li et al., 2024b; Zhu et al., 2022) Gaussian kernel function (Wang et al., 2009; Chen et al., 2021) and construct the Gaussian kernel function using the mean Euclidean distance. For the parameter $k$, each comparison algorithm is tested by searching for the best value of $k$ within the range $k = [8, 10, 12, 14, 16]$. More experimental details can be found in Appendix D.3.

## 5.2 EXPERIMENTAL RESULTS

### 5.2.1 RESULT OF EXPERIMENTAL 1

The data for Experiment 1 when $l = u$ is presented in Table 2. The horizontal axis indicates the methods used, while the vertical axis represents the number of columns, and the horizontal axis represents the number of rows of the experimental matrix. The table entries represent the time required for Retraction, measured in seconds. The fastest method is highlighted in **red**. As observed, when the matrix is small, the Sinkhorn method is faster. However, as the matrix size increases, the Dykstras method shows significant advantages and produces the geodesic. Therefore, we recommend using the Dykstras method to obtain the Retraction curve. More data can be found in Appendix E.1.

### 5.2.2 RESULT OF EXPERIMENTAL 2

Table 3 shows the time and final loss required by the Riemannian Trust Region method to solve convex optimization problems of different scales. It can be seen that, for problems of varying sizes, the RIMTRT significantly outperforms the DSTRT in both time consumption and final loss. Therefore, we have highlighted the RIM manifold results in **red**. Data for the Riemannian Gradient Descent and Riemannian Conjugate Gradient methods can be found in Table 13 and Table 14.

For the second part of the experiment, Figure 1 shows the comparison of denoising results using the TV algorithm on the RIM manifold and the doubly stochastic manifold with a noise level of 0.3. In this case, $\xi = 0.3$. On the RIM manifold, the running time was **29.77s**, and the loss value decreased to **1.05e5**, while on the doubly stochastic manifold, the time was **85.33s**, and the loss value was **1.17e5**. By observing the images, it is evident that the image obtained using the doubly stochastic manifold has noticeable noise when zoomed in, while the image on the RIM manifold is smoother. Additional data and images can be found in Figure 4.

In addition, we evaluated the denoising results obtained from the RIM manifold and the doubly stochastic manifold using multiple metrics (Valsesia et al., 2020). The resulting table is shown Table 4. It can be observed that the denoising performance of the RIM manifold consistently surpasses that of the doubly stochastic manifold. In addition, it is important to note that the relative values rather

Table 3: Cost and Time on the RIM Manifold and Doubly Stochastic Manifold(RTR).

| Row&Col | RIM Manifold | | | | | | Doubly Stochastic Manifold | | | | | |
|---|---|---|---|---|---|---|---|---|---|---|---|---|
| | Cost | | | Time | | | Cost | | | Time | | |
| Size | 5000 | 7000 | 10000 | 5000 | 7000 | 10000 | 5000 | 7000 | 10000 | 5000 | 7000 | 10000 |
| 5 | 3.09E-23 | 8.28E-20 | 2.09E-20 | 0.265 | 0.355 | 0.516 | 4.38E-11 | 3.96E-10 | 2.89E-10 | 9.530 | 12.85 | 31.97 |
| 10 | 1.91E-20 | 9.58E-20 | 3.80E-20 | 0.283 | 0.464 | 0.690 | 1.91E-10 | 3.66E-10 | 4.12E-10 | 16.25 | 17.04 | 32.72 |
| 20 | 1.02E-19 | 1.22E-23 | 8.29E-19 | 0.366 | 0.562 | 0.691 | 9.49E-10 | 6.04E-10 | 1.17E-09 | 18.77 | 35.15 | 26.77 |
| 50 | 7.66E-20 | 2.13E-18 | 2.20E-20 | 0.602 | 0.844 | 1.087 | 3.08E-09 | 1.99E-09 | 1.57E-09 | 38.55 | 64.27 | 111.1 |
| 70 | 1.85E-20 | 2.84E-18 | 7.49E-19 | 0.791 | 0.983 | 1.352 | 2.59E-09 | 1.65E-09 | 3.18E-09 | 70.47 | 121.0 | 77.28 |
| 100 | 1.31E-19 | 5.04E-20 | 1.26E-17 | 0.990 | 1.324 | 1.721 | 1.78E-09 | 2.18E-09 | 2.83E-09 | 91.40 | 121.3 | 241.4 |

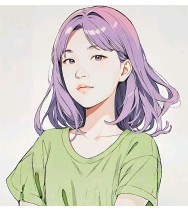 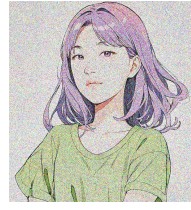 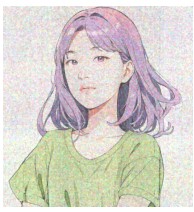 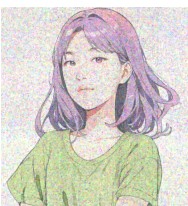

| (a) Origin | (b) Noisy Image | (c) RIM Result | (d) DS Result |
|---|---|---|---|

Figure 1: Image Denoising Results, Noise Coefficient 0.3, $\xi = 0.3$.

than the absolute values—of metrics such as LPIPS should be the focus, since TV regularization is not a state-of-the-art denoising method. Our goal here is to compare RIM against DSM.

### 5.2.3 RESULT OF EXPERIMENTAL 3

When $l = u$, the time and loss for the seven comparison algorithms are presented in Table 5. We have marked the algorithm names on the RIM manifold in blue, the shortest time in **red**, and the lowest loss in bright red. It can be observed that the optimization algorithms on the RIM manifold achieved most of the top positions. Figure 2 shows the loss decrease curves for some datasets. More results can be found in Appendix E.3.

### 5.2.4 RESULT OF EXPERIMENTAL 4

For Experiment 4, Table 6 records the performance of 12 comparison algorithms across 8 real-world datasets based on clustering accuracy (ACC), normalized mutual information (NMI), and adjusted Rand index (ARI). Our algorithm is marked in blue, and the best-performing algorithm is marked in **red**. It can be observed that performing Ratio Cut on the RIM manifold leads to superior results compared to the most advanced algorithms. More results can be found in Appendix E.4.

## 6 LIMITATIONS

We acknowledge that our study still has several limitations that warrant further investigation. First, the RIM manifold is a relaxation of the indicator matrix, and for an NP-hard non-convex indicator-matrix optimization problem, our analysis can only demonstrate that the RIM manifold outperforms existing relaxations such as the single stochastic, doubly stochastic, and Stiefel manifolds in various aspects. It remains challenging to provide a precise bound between the optimal solution of the relaxed problem and that of the original NP-hard problem under general conditions. Second, the RIM manifold employs projection onto a closed set as its retraction. When strict adherence to an open set is required, a small correction term $\varepsilon$. Finally, additional experimental results could be included in future work to further demonstrate the superiority of the RIM manifold.

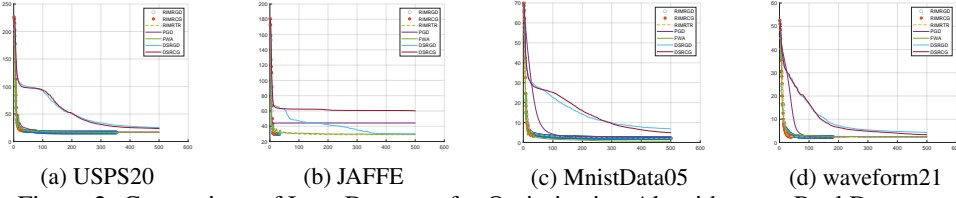

| (a) USPS20 | (b) JAFFE | (c) MnistData05 | (d) waveform21 |
|---|---|---|---|

Figure 2: Comparison of Loss Decrease for Optimization Algorithms on Real Datasets.

Table 4: Comparison of Denoising Performance between the RIM Manifold and the DSM

| (noise, $\xi$) Metric | (0.3, 0.3) RIM | DSM | (0.3, 0.7) RIM | DSM | (0.5, 0.3) RIM | DSM | (0.5, 0.7) RIM | DSM | (0.9, 0.3) RIM | DSM | (0.9, 0.7) RIM | DSM |
|---|---|---|---|---|---|---|---|---|---|---|---|---|
| MSE | 0.012 | 0.015 | 0.020 | 0.027 | 0.022 | 0.026 | 0.023 | 0.032 | 0.085 | 0.106 | 0.044 | 0.051 |
| PSNR | 19.27 | 18.33 | 17.06 | 15.66 | 16.56 | 15.78 | 16.32 | 14.92 | 10.73 | 9.751 | 13.59 | 12.89 |
| SSIM | 0.502 | 0.412 | 0.434 | 0.256 | 0.282 | 0.247 | 0.327 | 0.209 | 0.107 | 0.096 | 0.183 | 0.148 |
| LPIP | 0.561 | 0.671 | 0.719 | 0.929 | 0.742 | 0.824 | 0.775 | 0.968 | 0.969 | 1.020 | 0.803 | 1.020 |

Table 5: Time and Loss of Different Optimization Algorithms on Ratio Cut when $l = u$

| Datasets&Methods | DSRGD Time | Cost | DSRCG Time | Cost | FWA Time | Cost | PGD Time | Cost | RIMRGD Time | Cost | RIMRCG Time | Cost | RIMRTR Time | Cost |
|---|---|---|---|---|---|---|---|---|---|---|---|---|---|---|
| COIL20 | 8.978 | 28.17 | 11.90 | 28.41 | 10.49 | 41.12 | 6.967 | 31.53 | 1.145 | 24.83 | **0.685** | 27.46 | 14.20 | **22.48** |
| Digit | 8.650 | 2.751 | 11.87 | 2.312 | 9.196 | **0.492** | 6.077 | 0.953 | 7.058 | 0.942 | **0.886** | 1.319 | 13.73 | 1.089 |
| JAFFE | 2.224 | 30.06 | 2.774 | 60.16 | 0.303 | 29.39 | 2.725 | 44.35 | 0.149 | 29.56 | **0.119** | 29.92 | 1.982 | **28.94** |
| MSRA25 | 9.901 | 2.775 | 11.94 | 2.249 | 9.687 | 1.845 | 6.954 | 1.221 | 2.221 | 1.636 | **1.957** | **1.009** | 17.74 | 1.070 |
| PalmData25 | 43.39 | 737.1 | 54.48 | 1054 | 88.35 | 561.1 | 23.74 | 642.2 | 9.506 | **456.0** | **2.583** | 642.3 | 18.77 | 516.3 |
| USPS20 | 9.238 | 25.52 | 12.65 | 23.58 | 10.37 | 16.76 | 6.842 | 17.32 | 5.257 | 16.46 | **0.735** | 19.91 | 12.59 | **16.31** |
| Waveform21 | 11.16 | 4.328 | 13.76 | 3.313 | 17.81 | 2.457 | 8.645 | 2.392 | 4.094 | **2.385** | **1.237** | 2.434 | 8.508 | 2.390 |
| MnistData05 | 18.16 | 6.834 | 23.60 | 4.894 | 26.29 | **0.619** | 14.96 | 2.520 | 16.43 | 2.126 | **1.724** | 3.325 | 35.93 | 2.154 |

Table 6: Mean clustering performance of compared methods on real-world datasets.

| Metric | Method | COIL20 | Digit | JAFFE | MSRA25 | PalmData25 | USPS20 | Waveform21 | MnistData05 |
|---|---|---|---|---|---|---|---|---|---|
| ACC | KM | 53.44 | 58.33 | 72.16 | 49.33 | 70.32 | 55.51 | 50.38 | 53.86 |
| | CDKM | 52.47 | 65.82 | 80.85 | 59.63 | 76.05 | 57.68 | 50.36 | 54.24 |
| | Rcut | 78.14 | 74.62 | 84.51 | 56.84 | 87.03 | 57.83 | 51.93 | 62.80 |
| | Ncut | 78.88 | 76.71 | 83.76 | 56.23 | 86.76 | 59.20 | 51.93 | 61.14 |
| | Nystrom | 51.56 | 72.08 | 75.77 | 52.85 | 76.81 | 62.55 | 51.49 | 55.91 |
| | BKNC | 57.11 | 60.92 | 93.76 | **65.47** | 86.74 | 62.76 | 51.51 | 52.00 |
| | FCFC | 59.34 | 43.94 | 71.60 | 54.27 | 69.38 | 58.23 | 56.98 | 54.41 |
| | FSC | **82.76** | 79.77 | 81.69 | 56.25 | 82.27 | 67.63 | 50.42 | 57.76 |
| | LSCR | 65.67 | 78.14 | 91.97 | 53.82 | 58.25 | 63.07 | 56.19 | 57.15 |
| | LSCK | 62.28 | 78.04 | 84.98 | 54.41 | 58.31 | 61.86 | 54.95 | 58.57 |
| | RIMRcut | 79.72 | **82.53** | **96.71** | 56.64 | **90.85** | **70.28** | **74.80** | **65.55** |
| NMI | KM | 71.43 | 58.20 | 80.93 | 60.10 | 89.40 | 54.57 | 36.77 | 49.57 |
| | CDKM | 71.16 | 63.64 | 87.48 | 63.83 | 91.94 | 55.92 | 36.77 | 49.23 |
| | Rcut | 86.18 | 75.28 | 90.11 | 71.64 | 95.41 | 63.84 | 37.06 | 63.11 |
| | Ncut | 86.32 | 76.78 | 89.87 | 71.50 | 95.26 | 64.46 | 37.06 | **63.22** |
| | Nystrom | 66.11 | 70.13 | 82.53 | 57.77 | 93.09 | 59.00 | 36.95 | 48.53 |
| | BKNC | 69.80 | 59.37 | 92.40 | 69.30 | 95.83 | 57.10 | 36.94 | 44.56 |
| | FCFC | 74.05 | 38.33 | 80.30 | 63.34 | 89.47 | 55.71 | 22.89 | 48.75 |
| | FSC | **91.45** | **80.98** | 90.43 | 70.60 | 94.62 | **74.75** | 36.76 | 58.33 |
| | LSCR | 74.67 | 75.07 | 93.13 | 68.06 | 81.84 | 62.36 | 33.37 | 52.82 |
| | LSCK | 74.02 | 76.53 | 87.89 | 67.97 | 81.70 | 65.23 | 36.92 | 59.14 |
| | RIMRcut | 85.63 | 80.05 | **96.24** | **71.76** | **96.50** | 69.08 | **42.14** | 59.35 |
| ARI | KM | 50.81 | 45.80 | 66.83 | 34.66 | 65.06 | 43.57 | 25.56 | 37.18 |
| | CDKM | 48.11 | 52.74 | 76.36 | 37.70 | 71.73 | 45.59 | 25.56 | 36.79 |
| | Rcut | 73.73 | 65.81 | 81.70 | 46.35 | 84.76 | 51.99 | 25.31 | 51.32 |
| | Ncut | 74.30 | 68.21 | 81.30 | 45.90 | 84.25 | 52.72 | 25.31 | 50.51 |
| | Nystrom | 45.96 | 59.50 | 69.85 | 38.07 | 76.23 | 50.01 | 25.03 | 38.21 |
| | BKNC | 49.96 | 48.98 | 87.96 | 54.78 | 85.56 | 48.43 | 25.02 | 32.89 |
| | FCFC | 54.41 | 25.50 | 65.73 | 40.42 | 66.03 | 46.32 | 22.89 | 36.86 |
| | FSC | **79.46** | **73.03** | 80.26 | 43.99 | 79.67 | **61.71** | 25.10 | 44.78 |
| | LSCR | 57.68 | 67.21 | 86.76 | 43.31 | 48.70 | 52.64 | 25.12 | 41.46 |
| | LSCK | 54.59 | 68.70 | 77.37 | 42.18 | 48.58 | 52.54 | 26.47 | 46.48 |
| | RIMRcut | 73.98 | **75.01** | **93.32** | **46.82** | **88.49** | 56.06 | **42.89** | **52.87** |

## 7 CONCLUSION

This paper presents a new relaxation for indicator matrices and proves that it forms a Riemannian manifold. We have constructed a Riemannian toolbox for optimization on the RIM manifold. In particular, we introduce multiple methods for Retraction, including one that operates quickly along the geodesic. The paper demonstrates that optimization on the RIM manifold is useful for machine learning and it is a fast method $\mathcal{O}(n)$ that can replace the existing double stochastic manifold optimization with a time complexity of $\mathcal{O}(n^3)$. Through large-scale experiments from multiple perspectives, we have proven the effectiveness and speed of optimization on the RIM manifold.

## 8 ACKNOWLEDGEMENT

This work was supported by the National Natural Science Foundation of China under Grant 62576277.

## 9 STATEMENT

For the reproducibility of this paper, we have submitted the complete anonymized code with fixed random seeds, as detailed in Appendix H. In addition, large language models (LLMs) were only used for language polishing.

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

## CONTENTS

# A  PROOFS OF THEOREMS

## A.1  PROOF OF LEMMA 1

Our relaxed indicator matrix set $\mathcal{M} = \{X \mid X1_c = 1_n, l < X^T1_n < u, X > 0\}$ forms an embedded submanifold of the Euclidean space, with $\dim \mathcal{M} = (n-1)c$. We refer to it as the Relaxed Indicator Matrix Manifold.

*Proof.* The set $\mathcal{M}$ can be viewed as the intersection of three sets: $\mathcal{M} = \{X \mid X1_c = 1_n, l < X^T1_n < u, X > 0\} = \Omega_1 \cap \Omega_2 \cap \Omega_3$, where $\Omega_1 = \{X \mid X > 0, X1_c = 1_n\}$, $\Omega_2 = \{X \mid X^T1_n > l\}$, and $\Omega_3 = \{X \mid X^T1_n < u\}$. Consider the differential of the local defining function for the set $\Omega_1$, i.e.

$$D(X1_c - 1_n)[V] = \lim_{t \to 0} \frac{(X + tV)1_c - 1_n - (X1_c - 1_n)}{t} = \lim_{t \to 0} \frac{tV1_c}{t} = V1_c \qquad (23)$$

Consider the null space of $D(X1_c - 1_n)[V]$, given by $\mathrm{Ker}(D(X1_c - 1_n)[V]) = \{V \mid V1_c = 0\}$. The dimension of this null space is

$$\dim(\mathrm{Ker}(D(X1_c - 1_n)[V])) = nc - c = (n-1)c \qquad (24)$$

In addition, since $\Omega_2 = \{X \mid X^T1_n > l\}$ and $\Omega_3 = \{X \mid X^T1_n < u\}$, take $\Omega_2$ as an example. For any directional vector $U$, there must exist $\delta_U > 0$ such that $(X + \delta_U U)^T1_n > l$. Thus, both $\Omega_2$ and $\Omega_3$ are open sets. According to Theorem ([Petersen, 2006](#)), $\Omega_1$ forms a manifold, and $\Omega_2$ and $\Omega_3$ are open sets. The intersection of an open set with a manifold remains a manifold. Therefore, $\mathcal{M} = \{X \mid X1_c = 1_n, l < X^T1_n < u, X > 0\} = \Omega_1 \cap \Omega_2 \cap \Omega_3$ is still a manifold, and $\dim(\mathcal{M}) = \dim(\mathrm{Ker}(D(X1_c - 1_n)[V])) = (n-1)c$.

We refer to $\mathcal{M}$ as the Relaxed Indicator Matrix manifold, abbreviated as the RIM manifold.

## A.2  PROOF OF LEMMA 2

By restricting the Euclidean inner product $\langle U, V \rangle = \sum_{i=1}^{n} \sum_{j=1}^{c} U_{ij}V_{ij}$ onto the RIM manifold $\mathcal{M}$, the tangent space of $\mathcal{M}$ at $X$ is given by $T_X\mathcal{M} = \{U \mid U1_c = 0\}$. For any function $\mathcal{H}$, if its Euclidean gradient is $\mathrm{Grad}\,\mathcal{H}(F)$, the Riemannian gradient $\mathrm{grad}\,\mathcal{H}(F)$ is expressed as following.

$$\mathrm{grad}\,\mathcal{H}(F) = \mathrm{Grad}\,\mathcal{H}(F) - \frac{1}{c}\mathrm{Grad}\,\mathcal{H}(F)1_c1_c^T. \qquad (25)$$

*Proof.* According to the definition of tangent space,

$$T_X\mathcal{M} = \mathrm{Ker}(D(X1_c - 1_n)[U]) = \{U \mid U1_c = 0\} \qquad (26)$$

Let $\mathrm{Grad}\mathcal{H}$ be the gradient of $\mathcal{H}$ in the Euclidean space. Then, $\mathrm{Grad}\mathcal{H} = \mathrm{Grad}\mathcal{H}_{\parallel} + \mathrm{Grad}\mathcal{H}_{\perp}$, where $\mathrm{Grad}\mathcal{H}_{\parallel}$ represents the component of $\mathrm{Grad}\mathcal{H}$ parallel to $T_X\mathcal{M}$, and $\mathrm{Grad}\mathcal{H}_{\perp}$ represents the component perpendicular to $T_X\mathcal{M}$.

By the definition of the Riemannian gradient,

$$D\mathcal{H}[V] = \langle \mathrm{Grad}\mathcal{H}, V \rangle = \langle \mathrm{grad}\mathcal{H}, V \rangle_X, V \in T_X\mathcal{M} \qquad (27)$$

Here, $\langle \mathrm{grad}\mathcal{H}, V \rangle_X$ denotes the inner product equipped on the manifold at $X$. When $\langle \mathrm{grad}\mathcal{H}, V \rangle_X$ coincides with the Euclidean inner product, we have

$$\langle \mathrm{Grad}\mathcal{H}, V \rangle = \langle \mathrm{Grad}\mathcal{H}_{\parallel}, V \rangle + \langle \mathrm{Grad}\mathcal{H}_{\perp}, V \rangle = \langle \mathrm{Grad}\mathcal{H}_{\parallel}, V \rangle = \langle \mathrm{grad}\mathcal{H}, V \rangle_X \qquad (28)$$

for $V \in T_X\mathcal{M}$, since $\langle \mathrm{Grad}\mathcal{H}_{\perp}, V \rangle = 0$ for $V \in T_X\mathcal{M}$. By the Ritz representation theorem, in this case, $\mathrm{grad}\mathcal{H}$ is the orthogonal projection of $\mathrm{Grad}\mathcal{H}$ onto the tangent space. The next step is to solve the optimization problem:

$$\min_{U \in \{U | U1_c = 0\}} \mathcal{L} = \min_{U \in \{U | U1_c = 0\}} \|U - \mathrm{Grad}\mathcal{H}\|_F^2 \qquad (29)$$

The Lagrangian function for the optimization problem is given by: $\mathcal{L} = \frac{1}{2}\|U - \mathrm{Grad}\mathcal{H}\|_F^2 + \alpha^T(U1_c)$. Taking the gradient with respect to $U$, we have:

$$\nabla_U\mathcal{L} = U - \mathrm{Grad}\mathcal{H} + \alpha1_c^T = 0 \qquad (30)$$

Solving for $U$, we obtain$U = \text{Grad}\mathcal{H} - \alpha 1_c^T$.. Since $U1_c = 0$, substituting $U$ gives $\text{Grad}\mathcal{H}1_c - \alpha 1_c^T 1_c = \text{Grad}\mathcal{H}1_c - c\alpha = 0$, which implies $\alpha = \frac{1}{c}\text{Grad}\mathcal{H}1_c$. Therefore, the Riemannian gradient is following.

$$\text{grad}\mathcal{H} = \text{argmin}_{U \in \{U|U1_c=0\}}\|U - \text{Grad}\mathcal{H}\|_F^2 = \text{Grad}\mathcal{H} - \frac{1}{c}\text{Grad}\mathcal{H}1_c1_c^T \tag{31}$$

## A.3 PROOF OF LEMMA 3

For the manifold $\mathcal{M}$, there exists a unique connection that is compatible with the inner product and ensures that the Riemannian Hessian mapping is self-adjoint. This connection is given by following. $\bar{\nabla}$ is the Riemannian connection in Euclidean space.

$$\nabla_V U = \bar{\nabla}_V U - \frac{1}{c}\bar{\nabla}_V U1_c1_c^T. \tag{32}$$

*Proof.* First, we need to prove that the connection is compatible with the inner product, which means proving $W\langle U, V\rangle = \langle \nabla_W U, V\rangle + \langle U, \nabla_W V\rangle$. We have the following equation

$$W\langle U, V\rangle = D(\langle U, V\rangle)[W] = D\left(\sum_{i=1}^n \sum_{j=1}^n U_{ij}V_{ij}\right)[W] = \sum_{i=1}^n \sum_{j=1}^n D(U_{ij}V_{ij})[W]$$

$$= \sum_{i=1}^n \sum_{j=1}^n (V_{ij}D(U_{ij})[W_{ij}] + U_{ij}D(V_{ij})[W_{ij}]) = \langle U, D(V)[W]\rangle + \langle D(U)[W], V\rangle$$

$$= \langle U, D(V)[W] - \frac{1}{c}D(V)[W]1_c1_c^T\rangle + \langle D(U)[W] - \frac{1}{c}D(U)[W]1_c1_c^T, V\rangle$$

$$+ \langle U, \frac{1}{c}D(V)[W]1_c1_c^T\rangle + \langle \frac{1}{c}D(U)[W]1_c1_c^T, V\rangle. \tag{33}$$

Since the standard inner product in Euclidean space is chosen, we have

$$\langle U, \frac{1}{c}D(V)[W]1_c1_c^T\rangle = \frac{1}{c}\text{tr}(U^T D(V)[W]1_c1_c^T) = \frac{1}{c}\text{tr}(D(V)[W]1_c1_c^T U^T) \tag{34}$$

$$= \frac{1}{c}\text{tr}(D(V)[W]1_c(U1_c)^T) = 0 \tag{35}$$

The last step equals zero because $U \in T_X\mathcal{M}$, which implies that $U1_c = 0$. In the Euclidean space, the connection $\bar{\nabla}_V U$ is defined as $D(U)[V]$. Furthermore, we have:

$$W\langle U, V\rangle = \langle U, D(V)[W] - \frac{1}{c}D(V)[W]1_c1_c^T\rangle + \langle D(U)[W] - \frac{1}{c}D(U)[W]1_c1_c^T, V\rangle \tag{36}$$

$$= \langle U, \bar{\nabla}_W V - \frac{1}{c}\bar{\nabla}_W V1_c1_c^T\rangle + \langle \bar{\nabla}_W U - \frac{1}{c}\bar{\nabla}_W U1_c1_c^T, V\rangle \tag{37}$$

$$= \langle U, \nabla_W V\rangle + \langle V, \nabla_W U\rangle \tag{38}$$

The second step is to prove that the Hessian map obtained from the connection is self-adjoint. That is, we need to prove $[U, V] = \nabla_U V - \nabla_V U$, where $[U, V]$ is the Lie bracket, and $[U, V]f = U(V(f)) - V(U(f))$. with $f$ being a smooth scalar field on the manifold $\mathcal{M}$. $U$ and $V$ are tangent vectors of the RIM manifold $\mathcal{M}$, i.e., $U, V \in T_X\mathcal{M}$. Let $\bar{U}$ and $\bar{V}$ be smooth extensions of $U$ and $V$ in the neighborhood of $\mathcal{M}$, satisfying $\bar{U}|_{\mathcal{M}} = U$ and $\bar{V}|_{\mathcal{M}} = V$. We have $[\bar{U}, \bar{V}] = D\bar{V}[\bar{U}] - D\bar{U}[\bar{V}] = \bar{\nabla}_{\bar{U}}\bar{V} - \bar{\nabla}_{\bar{V}}\bar{U}$. Thus, we can prove that:

$$[U, V] = [\bar{U}, \bar{V}]|_{\mathcal{M}} \tag{39}$$

$$= (\bar{\nabla}_{\bar{U}}\bar{V} - \bar{\nabla}_{\bar{V}}\bar{U})|_{\mathcal{M}} \tag{40}$$

$$= \text{Proj}_{\mathcal{M}}(\bar{\nabla}_{\bar{U}}\bar{V} - \bar{\nabla}_{\bar{V}}\bar{U})|_{\mathcal{M}} \tag{41}$$

$$= \bar{\nabla}_U V - \frac{1}{c}\bar{\nabla}_U V1_c1_c^T - \bar{\nabla}_V U + \frac{1}{c}\bar{\nabla}_V U1_c1_c^T \tag{42}$$

$$= \nabla_U V - \nabla_V U. \tag{43}$$

This equality, $(\bar{\nabla}_{\bar{U}}\bar{V} - \bar{\nabla}_{\bar{V}}\bar{U})|_{\mathcal{M}} = \text{Proj}_{\mathcal{M}}(\bar{\nabla}_{\bar{U}}\bar{V} - \bar{\nabla}_{\bar{V}}\bar{U})|_{\mathcal{M}}$, holds because $[U, V]$ is defined in the tangent space of $\mathcal{M}$. Therefore, the expression $(\bar{\nabla}_{\bar{U}}\bar{V} - \bar{\nabla}_{\bar{V}}\bar{U})|_{\mathcal{M}}$ and its projection onto the tangent space of $\mathcal{M}$ must be equal.

## A.4 PROOF OF LEMMA 4

For the manifold $\mathcal{M}$ equipped with the connection $\nabla_V U = \bar{\nabla}_V U - \frac{1}{c}\bar{\nabla}_V U 1_c 1_c^T$, the Riemannian Hessian mapping satisfies following.

$$\text{hess}\,\mathcal{H}[V] = \text{Hess}\,\mathcal{H}[V] - \frac{1}{c}\text{Hess}\,\mathcal{H}[V]1_c 1_c^T. \tag{44}$$

*Proof.* The Riemannian Hessian is defined as

$$hess\mathcal{H}[U] = \nabla_U \text{grad}\,\mathcal{H} = \nabla_U \left(\text{Grad}\,\mathcal{H} - \frac{1}{c}\text{Grad}\,\mathcal{H}1_c 1_c^T\right). \tag{45}$$

Using the definition of the Riemannian connection $\nabla$, we have

$$\text{hess}\,\mathcal{H}[U] = \nabla_U \text{grad}\,\mathcal{H} = D\left(\text{Grad}\,\mathcal{H} - \frac{1}{c}\text{Grad}\,\mathcal{H}1_c 1_c^T\right)[U] \tag{46}$$

$$= \lim_{t\to 0}\frac{\text{Grad}\,\mathcal{H}(X + tU) - \text{Grad}\,\mathcal{H}(X)}{t} - \lim_{t\to 0}\frac{\text{Grad}\,\mathcal{H}(X + tU)1_c 1_c^T - \text{Grad}\,\mathcal{H}(X)1_c 1_c^T}{ct} \tag{47}$$

$$= \text{Hess}\,\mathcal{H}[V] - \frac{1}{c}\text{Hess}\,\mathcal{H}[V]1_c 1_c^T \tag{48}$$

## A.5 PROOF OF THEOREM 1

Let $R_X(tV) = \text{argmin}_{F\in\mathcal{M}}\|F - (X + tV)\|_F^2$, $X \in \mathcal{M}$. Then

$$\text{argmin}_{F\in\mathcal{M}}\|F - (X + tV)\|_F^2 = \max(0, X + tV - \nu(t)1_c^T - 1_n\omega^T(t) + 1_n\rho^T(t)) \tag{49}$$

where $\nu(t), \omega^T(t), \rho^T(t)$ are Lagrange multipliers. Moreover, there exists $\delta > 0$ such that for $t \in (0, \delta)$, $-\nu(t)1_c^T - 1_n\omega(t)^T + 1_n\rho(t)^T = 0$, and the Retraction satisfies the following. Where $\frac{D}{dt}$ denotes the Levi-Civita derivative.

$$R_X(0) = X, \quad \frac{d}{dt}R_X(tV)\big|_{t=0} = V, \quad \frac{D}{dt}R'_X(tV)\big|_{t=0} = 0 \tag{50}$$

Thus, $R_X(tV)$ is a geodesic.

*Proof.* First, the Lagrangian dual function of the original problem is as follows:

$$\mathcal{L}(F, \nu, \omega, \rho) = \frac{1}{2}\|F - (X + tV)\|_F^2 - \nu^T(F1_c - 1_n) - \omega^T(F^T1_n - u) + \rho^T(l - F^T1_n) \tag{51}$$

Where $\nu$, $\omega$, and $\rho$ are the corresponding Lagrange multipliers, satisfying $\nu \geq 0$, $\omega \geq 0$, $\rho \geq 0$. Let $\frac{\partial\mathcal{L}}{\partial F} = 0$, then we have the following formular:

$$\frac{\partial\mathcal{L}}{\partial F} = F - X + \nu 1_c^T + 1_n\omega^T - 1_n\rho^T - tV = 0, \tag{52}$$

That is, $F = X + tV - \nu 1_c^T - 1_n\omega^T + 1_n\rho^T$. Since $F$ lies on the manifold $\mathcal{M}$ and $F \geq 0$, the final result is:

$$F^* = \text{argmin}_{F\in\mathcal{M}}\|F - (X + tV)\|_F^2 = \max\left(0, X + tV - \nu 1_c^T - 1_n\omega^T + 1_n\rho^T\right). \tag{53}$$

It can be proven that $F^*$ satisfies the KKT conditions of the original problem. For different $t$, the values of the Lagrange multipliers $\nu, \omega, \rho$ vary, and they are functions of $t$: $\nu(t), \omega(t), \rho(t)$. The next step is to prove the three properties of the second-order Retraction, $R_X(0) = X, \frac{d}{dt}R_X(tV)\big|_{t=0} = V, \frac{D}{dt}R'_X(tV)\big|_{t=0} = 0$. First, consider $R_X(0) = \text{argmin}_{F\in\mathcal{M}}\|F - X\|_F^2$.

Since $X \in \mathcal{M}$, we have $R_X(0) = F^*(0) = X$. Additionally, since $F^*(0) = \max\left(0, X + tV - \nu 1_c^T - 1_n \omega^T + 1_n \rho^T\right)\big|_{t=0} = \max\left(0, X - \nu(0)1_c^T - 1_n \omega(0)^T + 1_n \rho(0)^T\right)$. We know that $X = \max\left(0, X - \nu(0)1_c^T - 1_n \omega(0)^T + 1_n \rho(0)^T\right)$.

According to the definition, we calculate:

$$\frac{d}{dt} R_X(tV)\big|_{t=0} = \lim_{t \to 0} \frac{F^*(t) - F^*(0)}{t} = \lim_{t \to 0} \frac{\max\left(0, X + tV - \nu 1_c^T - 1_n \omega^T + 1_n \rho^T\right) - F^*(0)}{t} \tag{54}$$

Since $X \in \mathcal{M}$, we know that $X_{ij} > 0$, $X1_c = 1_n$, and $l < X1_n < u$. Furthermore, since $V \in T_X\mathcal{M}$, there exists a $\delta > 0$ such that for $t \in (0, \delta)$, we still have $(X + tV)_{ij} > 0$, $(X + tV)1_c = 1_n$, and $l < (X + tV)1_n < u$. This means that for $t \in (0, \delta)$, we have $R_X(tV) = \arg\min_{F \in \mathcal{M}} \|F - (X + tV)\|_F^2$, and since $(X + tV) \in \mathcal{M}$, it follows that $R_X(tV) = F^*(t) = X + tV$. Therefore, we have:

$$\frac{d}{dt} R_X(tV)\big|_{t=0} = \lim_{t \to 0} \frac{F^*(t) - F^*(0)}{t} = \lim_{t \to 0} \frac{X + tV - X}{t} = V \tag{55}$$

For $\frac{D}{dt} R'_X(tV)$, first consider $\frac{d}{dt} R'_X(tV)\big|_{t=0} = \lim_{t \to 0} \frac{1}{t}\left(\frac{d}{dt} F^*(t) - \frac{d}{dt} F^*(0)\right)$. Since there exists an interval $(0, \delta)$ such that $F^*(t) = \max\left(0, X + tV - \nu 1_c^T - 1_n \omega^T + 1_n \rho^T\right) = X + tV$, and within $(0, \delta)$, without loss of generality, we can assume that $\nu(t)1_c^T - 1_n \omega(t)^T + 1_n \rho(t)^T = 0$, and within this interval, $X + tV > 0$. Thus, within $(0, \delta)$, we have $\frac{d}{dt} \max\left(0, X + tV - \nu 1_c^T - 1_n \omega^T + 1_n \rho^T\right) = V$. Therefore, $\frac{d}{dt} R'_X(tV)\big|_{t=0} = \lim_{t \to 0} \frac{1}{t}(V - V) = 0$. Thus, the Levi-Civita derivative, compatible with the connection, is $\frac{D}{dt} R'_X(tV)\big|_{t=0} = 0$. This concludes the proof.

## A.6 PROOF OF THEOREM 2

$\mathcal{M} = \Omega_1 \cup \Omega_2 \cup \Omega_3$, where $\Omega_1 = \{X \mid X > 0, X1_c = 1_n\}$, $\Omega_2 = \{X \mid X^T 1_n > l\}$, and $\Omega_3 = \{X \mid X^T 1_n < u\}$. The primal problem can be solved using the Dykstras (Tibshirani, 2017; Boyle & Dykstra, 1986) algorithm by iteratively projecting onto $\Omega_1$, $\Omega_2$, and $\Omega_3$. Specifically:

$\text{Proj}_{\Omega_1}(X) = \left(X_{ij} + \eta_i\right)_+$, where $\eta$ is determined by $\text{Proj}_{\Omega_1}(X)1_c = 1_n$.

$\text{Proj}_{\Omega_2}(X)$ and $\text{Proj}_{\Omega_3}(X)$ are defined similarly. For example,

$$\text{Proj}_{\Omega_2}(X^j) = \begin{cases} X^j, & \text{if } (X^j)^T 1_n > l_j, \\ \frac{1}{n}(l_j - 1_n^T X^j)1_n + X^j, & \text{if } (X^j)^T 1_n \leq l_j, \end{cases} \tag{56}$$

where $X^j$ is the $j$-th column of $X$, and $l_j$ is the $j$-th element of the column vector $l$.

*Proof.* Consider first the orthogonal projection on $\Omega_1$, which is to solve the optimization problem: $F = \arg\min_{F \in \Omega_1} \|F - X\|_F^2$ where $\Omega_1 = \{X \mid X > 0, X1_c = 1_n\}$. The Lagrange function for this problem, incorporating the equality constraint $X1_c = 1_n$ and the inequality constraint $X > 0$, is:

$$\mathcal{L}(F, \eta, \Theta) = \frac{1}{2}\|F - X\|_F^2 - \eta^T(F1_c - 1_n) - \sum_{i,j} \Theta_{ij} F_{ij} \tag{57}$$

where $\eta \in \mathbb{R}^n$ are Lagrange multipliers for the equality constraints, and $\Theta_{ij} \geq 0$ are multipliers for the non-negativity constraints.

Since the constraints are separable row-wise, we optimize each row $F_i$ independently. The row-wise Lagrangian is $\mathcal{L}_i(F_i, \eta_i, \Theta_i) = \frac{1}{2}\|F_i - X_i\|_2^2 - \eta_i(F_i 1_c - 1) - \sum_j \Theta_{ij} F_{ij}$. Taking the gradient with respect to $F_i$ and setting it to zero:

$$F_i - X_i - \eta_i 1_c^T - \Theta_i = 0 \quad \Rightarrow \quad F_i = X_i + \eta_i 1_c^T + \Theta_i \tag{58}$$

By complementary slackness, $\Theta_{ij} F_{ij} = 0$. If $F_{ij} > 0$, then $\Theta_{ij} = 0$, implying $F_{ij} = X_{ij} + \eta_i$. If $F_{ij} = 0$, then $X_{ij} + \eta_i + \Theta_{ij} = 0$ with $\Theta_{ij} \geq 0$, hence $X_{ij} + \eta_i \leq 0$. Thus, the optimal solution is:

$$F_{ij}^* = \max(X_{ij} + \eta_i, 0) = (X_{ij} + \eta_i)_+ \tag{59}$$

The multiplier $\eta_i$ is determined by the equality constraint $F_i^* 1_c = 1 \rightarrow \sum_{j=1}^{c} (X_{ij} + \eta_i)_+ = 1$

For the projection onto $\Omega_2$, consider the optimization problem: $F^* = \text{argmin}_{F \in \Omega_2} \|F - X\|_F^2$ where $\Omega_2 = \{X \mid X^T 1_n > l\}$. For each column $X^j$, solve: $\min_{F^j} \|F^j - X^j\|_2^2 \quad \text{s.t.} \quad (F^j)^T 1_n > l_j$.

If $(X^j)^T 1_n > l_j$, the constraint is already satisfied: $\text{Proj}_{\Omega_2}(X^j) = X^j$

If $(X^j)^T 1_n \leq l_j$, introduce the Lagrangian:

$$\mathcal{L}(F^j, \lambda) = \frac{1}{2}\|F^j - X^j\|_2^2 + \lambda\big(l_j - (F^j)^T 1_n\big), \quad \lambda \geq 0 \tag{60}$$

Taking the gradient of $F^j$, we have the following:

$$\nabla_{F^j}\mathcal{L} = F^j - X^j - \lambda 1_n = 0 \quad \Rightarrow \quad F^j = X^j + \lambda 1_n \tag{61}$$

Substitute into the binding constraint $(F^j)^T 1_n = l_j$:

$$(X^j + \lambda 1_n)^T 1_n = l_j \quad \Rightarrow \quad \lambda = \frac{1}{n}\big(l_j - (X^j)^T 1_n\big) \tag{62}$$

Thus, the projection is:

$$\text{Proj}_{\Omega_2}(X^j) = X^j + \frac{1}{n}\big(l_j - (X^j)^T 1_n\big)1_n \tag{63}$$

Combining both cases, we have that

$$\text{Proj}_{\Omega_2}(X^j) = \begin{cases} X^j, & \text{if } (X^j)^T 1_n > l_j, \\ \frac{1}{n}(l_j - 1_n^T X^j)1_n + X^j, & \text{if } (X^j)^T 1_n \leq l_j, \end{cases} \tag{64}$$

Similarly, for the projection onto $\Omega_3$, we can follow the same procedure and obtain:

$$\text{Proj}_{\Omega_3}(X^j) = \begin{cases} X^j, & \text{if } (X^j)^T 1_n < u_j, \\ \frac{1}{n}(u_j - 1_n^T X^j)1_n + X^j, & \text{if } (X^j)^T 1_n \geq u_j, \end{cases} \tag{65}$$

where $X^j$ is the $j$-th column of $X$, and $u_j$ is the $j$-th element of the column vector $u$.

The ultimate goal is to perform an orthogonal projection onto the intersection of three convex sets, $\Omega_1, \Omega_2, \Omega_3$. This can be achieved using the von Neumann iterative projection theorem. However, the von Neumann iterative projection can only guarantee convergence to $\Omega_1 \cap \Omega_2 \cap \Omega_3$, but it does not ensure the orthogonal projection, i.e., the solution to the Retraction problem. To address this, we introduce Dykstras's projection algorithm, which performs a linear correction to the von Neumann projection algorithm at each step, ensuring that it achieves the orthogonal projection onto $\Omega_1 \cap \Omega_2 \cap \Omega_3$. The algorithm flowchart for Dykstras's projection algorithm for the intersection of $d$ convex sets is shown below.

---
**Algorithm 1:** Dykstras's Algorithm for Projection onto the Intersection of Convex Sets

---
**Input:** Closed convex sets $\Omega_1, \Omega_2, \ldots, \Omega_d$ and point $y \in \mathbb{R}^{n \times c}$
**Output:** Sequence of iterates $u^{(k)}$ converging to the projection onto $\Omega_1 \cap \cdots \cap \Omega_d$

1   Initialize $u^{(0)} = y$, $z_1^{(0)} = \cdots = z_d^{(0)} = 0$;
2   **while** *not converged* **do**
3     $u_0^{(k)} = u_d^{(k-1)}$;
4     **for** *i = 1 to d* **do**
5       $u_i^{(k)} = \text{Proj}_{\Omega_i}(u_{i-1}^{(k)} + z_i^{(k-1)})$;
6       $z_i^{(k)} = u_{i-1}^{(k)} + z_i^{(k-1)} - u_i^{(k)}$;
7     **end**
8     $k \leftarrow k + 1$;
9   **end**
10   **return** $u^{(k)}$;

---

The algorithm iteratively performs $\text{Proj}_{\Omega_1}(\cdot)$, $\text{Proj}_{\Omega_2}(\cdot)$, and $\text{Proj}_{\Omega_3}(\cdot)$, and at each step, a linear correction using $u^{(k)}$ is applied. This ensures the final result is the orthogonal projection onto the intersection $\Omega_1 \cap \Omega_2 \cap \Omega_3$.

A.7    PROOF OF THEOREM 3

Solving the primal problem is equivalent to solving the following dual problem:

$$\max_{\omega \geq 0, \rho \geq 0} \mathcal{L} = \frac{1}{2} \| \max(0, X + tV - \nu 1_c^T - 1_n \omega^T + 1_n \rho^T) \|_F^2 - \langle \nu, 1_n \rangle - \langle \omega, u \rangle + \langle \rho, l \rangle \quad (66)$$

where $\nu$, $\omega$, and $\rho$ are Lagrange multipliers. The partial derivatives of $\mathcal{L}$ with respect to $\nu$, $\omega$, and $\rho$ are known, and gradient ascent can be used solving $\nu$, $\omega$, and $\rho$. Finally, $R_X(tV)$ can be obtained using $\max(0, X + tV - \nu 1_c^T - 1_n \omega^T + 1_n \rho^T)$. The partial derivatives are following.

$$\begin{cases} \dfrac{\partial \mathcal{L}}{\partial \nu} = \max(0, X + tV - \nu 1_c^T - 1_n \omega^T + 1_n \rho^T) 1_c - 1_n \\[2mm] \dfrac{\partial \mathcal{L}}{\partial \omega} = \max(0, X + tV - \nu 1_c^T - 1_n \omega^T + 1_n \rho^T)^T 1_n - u \\[2mm] \dfrac{\partial \mathcal{L}}{\partial \rho} = - \max(0, X + tV - \nu 1_c^T - 1_n \omega^T + 1_n \rho^T)^T 1_n + l \end{cases} \quad (67)$$

*Proof.* According to the previous theorem, we know that

$$F^* = \max \left(0, X + tV - \nu 1_c^T - 1_n \omega^T + 1_n \rho^T \right) \quad (68)$$

Substituting $F^*$ into the Lagrangian function, we obtain

$$\mathcal{L}(\nu, \omega, \theta) = \frac{1}{2} \left\| \max \left(0, X + tV - \nu 1_c^T - 1_n \omega^T + 1_n \rho^T \right) - X - tV \right\|_F^2 \quad (69)$$

$$+ \nu^T \max \left(0, X + tV - \nu 1_c^T - 1_n \omega^T + 1_n \rho^T \right) 1_c - \nu^T 1_n \quad (70)$$

$$+ \omega^T \max \left(0, X + tV - \nu 1_c^T - 1_n \omega^T + 1_n \rho^T \right)^T 1_n - \omega^T u \quad (71)$$

$$- \rho^T \max \left(0, X + tV - \nu 1_c^T - 1_n \omega^T + 1_n \rho^T \right)^T 1_n + \rho^T l \quad (72)$$

Among the Lagrange multipliers $\nu, \omega, \rho$, we have $\omega \geq 0$ and $\rho \geq 0$.

$\star$ If $\left(X + tV - \nu 1_c^T - 1_n \omega^T + 1_n \rho^T \right) < 0$, then $\max \left(0, X + tV - \nu 1_c^T - 1_n \omega^T + 1_n \rho^T \right) = 0$, which further leads to

$$\mathcal{L}(\nu, \omega, \rho) = \frac{1}{2} \| X + tV \|_F^2 - \nu^T 1_n - \omega^T u + \rho^T l \quad (73)$$

At this point, a simple differentiation yields:

$$\frac{\partial}{\partial \nu} \mathcal{L}(\nu, \omega, \rho) = -1_n, \quad \frac{\partial}{\partial \omega} \mathcal{L}(\nu, \omega, \rho) = -u, \quad \frac{\partial}{\partial \rho} \mathcal{L}(\nu, \omega, \rho) = l \quad (74)$$

$\star$ If $\left(X + tV - \nu 1_c^T - 1_n \omega^T + 1_n \rho^T \right) \geq 0$, then $\max \left(0, X + tV - \nu 1_c^T - 1_n \omega^T + 1_n \rho^T \right) = X + tV - \nu 1_c^T - 1_n \omega^T + 1_n \rho^T$. It is worth noting that $\nu^T \max \left(0, X + tV - \nu 1_c^T - 1_n \omega^T + 1_n \rho^T \right) 1_c \in \mathbb{R}$ is a real number, that is,

$$\nu^T \max \left(0, X + tV - \nu 1_c^T - 1_n \omega^T + 1_n \rho^T \right) 1_c \quad (75)$$

$$= \mathrm{tr} \left(\nu^T \max \left(0, X + tV - \nu 1_c^T - 1_n \omega^T + 1_n \rho^T \right) 1_c \right) \quad (76)$$

$$= \mathrm{tr} \left( \max \left(0, X + tV - \nu 1_c^T - 1_n \omega^T + 1_n \rho^T \right)^T \nu 1_c^T \right) \quad (77)$$

$$= \left\langle \max \left(0, X + tV - \nu 1_c^T - 1_n \omega^T + 1_n \rho^T \right), \nu 1_c^T \right\rangle. \quad (78)$$

 At this point, we have

$$\mathcal{L}(\nu, \omega, \rho) = \frac{1}{2} \left\| \nu 1_c^T + 1_n \omega^T - 1_n \rho^T \right\|_F^2 \quad - \langle \nu, 1_n \rangle - \langle \omega, u \rangle + \langle \rho, l \rangle \quad (79)$$

$$+ \left\langle X + tV - \nu 1_c^T - 1_n \omega^T + 1_n \rho^T, \nu 1_c^T + 1_n \omega^T - 1_n \rho^T \right\rangle \quad (80)$$

$$= \frac{1}{2} \left\| \nu 1_c^T + 1_n \omega^T - 1_n \rho^T \right\|_F^2 \quad - \langle \nu, 1_n \rangle - \langle \omega, u \rangle + \langle \rho, l \rangle \quad (81)$$

$$+ \left\langle X + tV, \nu 1_c^T + 1_n \omega^T - 1_n \rho^T \right\rangle - \left\| \nu 1_c^T + 1_n \omega^T - 1_n \rho^T \right\|_F^2 \quad (82)$$

$$= -\frac{1}{2}\left\|\nu 1_c^T + 1_n\omega^T - 1_n\rho^T\right\|_F^2 \quad - \langle \nu, 1_n \rangle - \langle \omega, u \rangle \tag{83}$$

$$+ \langle \rho, l \rangle + \langle X + tV, \nu 1_c^T + 1_n\omega^T - 1_n\rho^T \rangle \tag{84}$$

At this point, taking derivatives of the Lagrangian with respect to the multipliers $\nu, \omega, \rho$, we obtain

$$\begin{cases} \dfrac{\partial\mathcal{L}}{\partial\nu} = (X + tV - \nu 1_c^T - 1_n\omega^T + 1_n\rho^T)1_c - 1_n, \\[2mm] \dfrac{\partial\mathcal{L}}{\partial\omega} = (X + tV - \nu 1_c^T - 1_n\omega^T + 1_n\rho^T)^T 1_n - u, \\[2mm] \dfrac{\partial\mathcal{L}}{\partial\rho} = -(X + tV - \nu 1_c^T - 1_n\omega^T + 1_n\rho^T)^T 1_n + l. \end{cases} \tag{85}$$

Finally, by consolidating the two cases, we obtain

$$\begin{cases} \dfrac{\partial\mathcal{L}}{\partial\nu} = \max(0, X + tV - \nu 1_c^T - 1_n\omega^T + 1_n\rho^T)1_c - 1_n, \\[2mm] \dfrac{\partial\mathcal{L}}{\partial\omega} = \max(0, X + tV - \nu 1_c^T - 1_n\omega^T + 1_n\rho^T)^T 1_n - u, \\[2mm] \dfrac{\partial\mathcal{L}}{\partial\rho} = -\max(0, X + tV - \nu 1_c^T - 1_n\omega^T + 1_n\rho^T)^T 1_n + l. \end{cases} \tag{86}$$

After obtaining the gradient, the dual problem can be solved by a simple dual gradient ascent method. It should be noted that the multipliers $\omega$ and $\rho$ have non-negative constraints, so projection onto the constraints is needed. Specifically, after each gradient ascent step, $\omega$ and $\rho$ should be projected onto the non-negative constraint. Once $\nu, \omega$, and $\rho$ are obtained, $F^*$ can be derived using

$$F^* = \max\left(0, X + tV - \nu 1_c^T - 1_n\omega^T + 1_n\rho^T\right) \tag{87}$$

The algorithm flow is as follows:

---

**Algorithm 2:** Dual Gradient Projection Ascent Method

---

**Input:** Initial values: $\nu_0, \omega_0, \rho_0$
Step size $\kappa > 0$
Constraints: $\omega \geq 0, \rho \geq 0$
**Output:** Optimized multipliers: $\nu^*, \omega^*, \rho^*$

1   Initialize $\nu = \nu_0, \omega = \omega_0, \rho = \rho_0$;
2   **while** *not converged* **do**
3     **Compute Gradient:**;
4     $\frac{\partial\mathcal{L}}{\partial\nu}, \frac{\partial\mathcal{L}}{\partial\omega}, \frac{\partial\mathcal{L}}{\partial\rho}$;
5     **Update multipliers:**;
6     $\nu \leftarrow \nu + \kappa \cdot \frac{\partial\mathcal{L}}{\partial\nu}$;
7     $\omega \leftarrow \omega + \kappa \cdot \frac{\partial\mathcal{L}}{\partial\omega}$;
8     $\rho \leftarrow \rho + \kappa \cdot \frac{\partial\mathcal{L}}{\partial\rho}$;
9     **Project onto constraints:**;
10    $\omega \leftarrow \max(0, \omega)$;
11    $\rho \leftarrow \max(0, \rho)$;
12   **end**
13   **return** *Final values $\nu, \omega, \rho$*;

---

### A.8   Proof of Theorem 4

The Sinkhorn-based Retraction is defined as

$$R_X^s(tV) = \mathcal{S}(X \odot \exp(tV \oslash X)) = \operatorname{diag}(p^*)(X \odot \exp(tV \oslash X))\operatorname{diag}(q^* \odot w^*) \tag{88}$$

where $p^*, q^*, w^*$ are vectors, $\exp(\cdot)$ denotes element-wise exponentiation, and $\operatorname{diag}(\cdot)$ converts a vector into a diagonal matrix. The vectors $p^*, q^*, w^*$ are obtained by iteratively updating the following

equations:

$$\begin{cases} p^{(k+1)} = 1_n \oslash \left( (X \odot \exp(tV \oslash X)) \left( q^{(k)} \odot w^{(k)} \right) \right), \\ q^{(k+1)} = \max \left( l \oslash \left( (X \odot \exp(tV \oslash X))^T \, p^{(k+1)} \odot w^{(k)} \right), 1_c \right), \\ w^{(k+1)} = \min \left( u \oslash \left( (X \odot \exp(tV \oslash X))^T \, p^{(k+1)} \odot q^{(k+1)} \right), 1_c \right). \end{cases} \quad (89)$$

This iterative procedure ensures the mapping onto the RIM manifold. The solution $R_X^s(tV) = \operatorname{diag}(p^*)(X \odot \exp(tV \oslash X)) \operatorname{diag}(q^* \odot w^*)$ is equivalent to solving the dual-bound optimal transport problem (12) with an entropy regularization parameter of 1.

$$R_X^s(tV) = \operatorname{argmin}_{F \in \mathcal{M}} \left\langle F, -\log(X \odot \exp(tV \oslash X)) \right\rangle + \delta \big|_{\delta=1} \sum_{i=1}^n \sum_{j=1}^c \left( F_{ij} \log(F_{ij}) - F_{ij} \right) \quad (90)$$

*Proof.* Introduce Lagrange multipliers $\eta \in \mathbb{R}^n$ (for equality $F 1_c = 1_n$), and $\lambda, \nu \in \mathbb{R}^c, \lambda, \nu > 0$ (for inequalities $F^T 1_n > l$, $F^T 1_n < u$). The Lagrangian is:

$$\mathcal{L}(F, \eta, \lambda, \nu) = \left\langle F, -\log(X \odot \exp(tV \oslash X)) \right\rangle + \sum_{i,j} \left( F_{ij} \log(F_{ij}) - F_{ij} \right)$$
$$+ \eta^T (F 1_c - 1_n) + \lambda^T (l - F^T 1_n) + \nu^T (F^T 1_n - u). \quad (91)$$

Differentiate $\mathcal{L}$ with respect to $F_{ij}$ and set to zero, we have

$$-\log \left( X_{ij} \exp \left( \frac{tV_{ij}}{X_{ij}} \right) \right) + \log F_{ij} + \eta_i - \lambda_j + \nu_j = 0 \quad (92)$$

Simplify using $\log(X_{ij} \exp(tV_{ij}/X_{ij})) = \log X_{ij} + tV_{ij}/X_{ij}$:

$$-X_{ij} - \frac{tV_{ij}}{X_{ij}} + \log F_{ij} + \eta_i - \lambda_j + \nu_j = 0 \quad (93)$$

Solve for $F_{ij}$:

$$F_{ij}^* = X_{ij} \exp \left( \frac{tV_{ij}}{X_{ij}} - \eta_i + \lambda_j - \nu_j \right) = X_{ij} \exp \left( \frac{tV_{ij}}{X_{ij}} \right) e^{-\eta_i + \lambda_j - \nu_j} \quad (94)$$

Since $\lambda$ and $\nu$ are positive, we introduce the following variable substitutions:

$$\begin{cases} p = e^{-\eta}, \\ q = e^{\lambda}, & e^{\lambda} \geq 1_n, \\ w = e^{-\nu}, & e^{-\nu} \leq 1_n. \end{cases} \quad (95)$$

Writing the component-wise form into matrix form, we have the following formula.

$$F^* = \operatorname{diag}(p) \left( X \odot \exp(tV \oslash X) \right) \operatorname{diag}(q \odot w). \quad (96)$$

To construct the iterative format, we first consider the equality constraints. Substitute $F$ into $F 1_c = 1_n$:

$$\operatorname{diag}(p) \left( X \odot \exp(tV \oslash X) \right) \operatorname{diag}(q \odot w) 1_c = 1_n \Rightarrow \operatorname{diag}(p) \left( X \odot \exp(tV \oslash X) \right) (q \odot w) = 1_n \quad (97)$$

Further, we can derive the iterative update formula for the row equality constraints.

$$p = 1_n \oslash \left( (X \odot \exp(tV \oslash X)) (q \odot w) \right) \Rightarrow p^{(k+1)} = 1_n \oslash \left( (X \odot \exp(tV \oslash X)) \left( q^{(k)} \odot w^{(k)} \right) \right) \quad (98)$$

Next, considering the column constraint $F^T 1_n > l$, substituting $F$, we obtain:

$$\operatorname{diag}(q \odot w) \left( X \odot \exp(tV \oslash X) \right)^T \operatorname{diag}(p)^T 1_n > l \Rightarrow (q \odot w) \odot \left( (X \odot \exp(tV \oslash X))^T p \right) > l \quad (99)$$

By the complementary slackness condition, we obtain:

$$\lambda_j \left[ (q \odot w) \odot \left( (X \odot \exp(tV \oslash X))^T p \right) - l \right]_j = 0 \quad (100)$$

At this point, we discuss the complementary slackness condition.

$$\begin{cases} \left[(q \odot w) \odot \left((X \odot \exp(tV \oslash X))^T p\right)\right]_j \neq l_j, & \lambda_j = 0 \Rightarrow q_j = 1 \\ \left[(q \odot w) \odot \left((X \odot \exp(tV \oslash X))^T p\right)\right]_j = l_j, & \Rightarrow q_j = \left(l \oslash \left((X \odot \exp(tV \oslash X))^T p \odot w\right)\right)_j \end{cases}$$
(101)

The element-wise iterative update formula is then derived as follows.

$$q_j = \max\left(l \oslash \left((X \odot \exp(tV \oslash X))^T p \odot w\right), 1_c\right)_j$$
(102)

$$\Rightarrow q_j^{(k+1)} = \max\left(l \oslash \left((X \odot \exp(tV \oslash X))^T p \odot w\right), 1_c\right)_j$$
(103)

$$\Rightarrow q^{(k+1)} = \max\left(l \oslash \left((X \odot \exp(tV \oslash X))^T p \odot w\right), 1_c\right)$$
(104)

Considering the column constraint $F^T 1_n < u$, substituting $F$, we obtain:

$$\mathrm{diag}(q \odot w)\,(X \odot \exp(tV \oslash X))^T \mathrm{diag}(p)^T 1_n < u \Rightarrow (q \odot w) \odot \left((X \odot \exp(tV \oslash X))^T p\right) < u$$
(105)

By the complementary slackness condition for upper bounds:

$$\nu_j \left[u - (q \odot w) \odot \left((X \odot \exp(tV \oslash X))^T p\right)\right]_j = 0$$
(106)

This leads to two cases:

$$\begin{cases} \left[(q \odot w) \odot \left((X \odot \exp(tV \oslash X))^T p\right)\right]_j \neq u_j, & \nu_j = 0 \Rightarrow w_j = 1 \\ \left[(q \odot w) \odot \left((X \odot \exp(tV \oslash X))^T p\right)\right]_j = u_j, & \Rightarrow w_j = \left(u \oslash \left(\left((X \odot \exp(tV \oslash X))^T p\right) \odot q\right)\right)_j \end{cases}$$
(107)

The element-wise update rule is then:

$$w_j = \min\left(u \oslash \left(\left((X \odot \exp(tV \oslash X))^T p\right) \odot q\right), 1_c\right)_j$$
(108)

$$\Rightarrow w_j^{(k+1)} = \min\left(u \oslash \left(\left((X \odot \exp(tV \oslash X))^T p^{(k+1)}\right) \odot q^{(k+1)}\right), 1_c\right)_j$$
(109)

$$\Rightarrow w^{(k+1)} = \min\left(u \oslash \left(\left((X \odot \exp(tV \oslash X))^T p^{(k+1)}\right) \odot q^{(k+1)}\right), 1_c\right)$$
(110)

The final update formula can be obtained as follows.

$$\begin{cases} p^{(k+1)} = 1_n \oslash \left((X \odot \exp(tV \oslash X))\,(q^{(k)} \odot w^{(k)})\right), \\ q^{(k+1)} = \max\left(l \oslash \left((X \odot \exp(tV \oslash X))^T p^{(k+1)} \odot w^{(k)}\right), 1_c\right), \\ w^{(k+1)} = \min\left(u \oslash \left((X \odot \exp(tV \oslash X))^T p^{(k+1)} \odot q^{(k+1)}\right), 1_c\right). \end{cases}$$
(111)

It is easy to verify that the result derived from Sinkhorn is indeed a Retraction (Douik & Hassibi, 2019). It can be seen that the $F$ obtained through the Retraction $R_X^s(tV)$ minimizes the inner product with $\log(X \odot \exp(tV \oslash X))$ under the entropy regularization coefficient of 1. On one hand, this entropy regularization is introduced merely to facilitate computation via the Sinkhorn theorem. On the other hand, the regularization coefficient being 1 lacks practical significance. Moreover, this Retraction is not a second-order Retraction, making its theoretical justification in terms of convergence properties less rigorous compared to the norm-minimizing Retraction. Therefore, the norm-minimizing Retraction is recommended.

## A.9 PROOF OF THEOREM 5

**Theorem 9.** The loss function for the Ratio Cut is given by $\mathcal{H}_r(F) = tr(F^T L F (F^T F)^{-1})$. Then, the Euclidean gradient of the loss function with respect to $F$ is:

$$\mathrm{Grad}\mathcal{H}_r(F) = 2\left(LF(F^T F)^{-1} - F(F^T F)^{-1}(F^T L F)(F^T F)^{-1}\right)$$
(112)

Given the substitutions $(F^T F)^{-1} = J$ and $F^T LF = K$, the Euclidean Hessian map for the loss function is:

$$\text{Hess}\mathcal{H}_r[V] = 2\big(LVJ - LFJ(V^T F + F^T V)J - VJKJ + FJ(V^T F + F^T V)JKJ \quad (113)$$
$$- FJ(V^T LF + F^T LV)J + FJKJ(V^T F + F^T V)J\big) \quad (114)$$

*Proof.* Let the objective function be $\mathcal{H}_r(F) = \text{tr}(F^T LFJ)$, where $J = (F^T F)^{-1}$. Apply a small perturbation $\delta F$ to $F$, yielding the variation:

$$\delta\mathcal{H}_r = \text{tr}\left((\delta F^T)LFJ + F^T L(\delta F)J - F^T LFJ\left((\delta F^T)F + F^T(\delta F)\right)J\right). \quad (115)$$

Using the cyclic property of the trace and symmetry ($L$ is symmetric, $J$ is symmetric), we simplify to:

$$\delta\mathcal{H}_r = 2\,\text{tr}\left(\delta F^T\left(LFJ - FJ(F^T LF)J\right)\right). \quad (116)$$

Thus, the Euclidean gradient is:

$$\text{Grad}\mathcal{H}_r(F) = 2\left(LFJ - FJ(F^T LF)J\right). \quad (117)$$

Apply the direction $V$ to the gradient and compute the directional derivative:

$$\text{Hess}\mathcal{H}_r[V] = \frac{d}{dt}\text{Grad}\mathcal{H}_r(F + tV)\Big|_{t=0}. \quad (118)$$

Expanding the components:

- The derivative of $LFJ$ gives $LVJ - LFJ(V^T F + F^T V)J$,

- The derivative of $-FJKJ$ yields:

$$-VJKJ - F\left[-J(V^T F + F^T V)JKJ + J(V^T LF + F^T LV)J + JKJ(V^T F + F^T V)J\right]. \quad (119)$$

Combining and simplifying:

$$\text{Hess}\mathcal{H}_r[V] = 2\Big(LVJ - LFJ(V^T F + F^T V)J - VJKJ + FJ(V^T F + F^T V)JKJ - FJ(V^T LF + F^T LV)J\Big). \quad (120)$$

Further, to obtain the Riemannian gradient and Riemannian Hessian mapping, the Euclidean gradient and Euclidean Hessian mapping from the above expressions can be projected onto the RIM manifold. This allows for the optimization of the Ratio Cut loss function on the RIM manifold.

### A.10 PROOF OF THEOREM 6

**Theorem 6.** For any graph cut problem expressed as $\mathcal{H}(F) = \text{tr}((F^T LF)(F^T WF)^{-1})$, where $W$ is any symmetric matrix, the Euclidean gradient $\text{Grad}\mathcal{H}(F)$ is bounded, and satisfies:

$$\|\text{Grad}\mathcal{H}(F)\|_{\circledS} \le 2\left(\frac{\|L\|_{\circledS}\sqrt{n}}{\alpha} + \frac{\|W\|_{\circledS}\|L\|_{\circledS}n^{3/2}}{\alpha^2}\right), \quad (121)$$

where

$$\alpha = \frac{\sigma_{\min}(W) \cdot l^2}{n}, \quad (122)$$

and $\sigma_{\min}(W)$ is the smallest singular value of the matrix $W$. This implies that $\mathcal{H}(F)$ is Lipschitz continuous.

*Proof.* The spectral norm of the matrix $F$, which is its largest singular value, satisfies:

$$\|F\|_{\circledS}^2 = \sigma_{\max}(F)^2 \le \sum_{i=1}^{n}\|F_i\|_2^2 \le n \cdot 1^2 = n, \quad (123)$$

therefore, $\|F\|_{\circledS} \le \sqrt{n}$.

Let $F^j$ be the $j$-th column of the matrix $F$. Given the constraint $F^\top 1_n > l$, the $\ell_1$-norm of $F^j$ satisfies $\|F^j\|_1 = \sum_{i=1}^n F_{ij} > l$. By the Cauchy–Schwarz inequality, we have:

$$\|F^j\|_1 \leq \sqrt{n}\|F^j\|_2 \quad \Rightarrow \quad \|F^j\|_2 \geq \frac{\|F^j\|_1}{\sqrt{n}} \geq \frac{l}{\sqrt{n}}. \tag{124}$$

Next, we estimate a lower bound for the smallest singular value of the matrix $F^T W F$. For any unit vector $v \in \mathbb{R}^c$, we have:

$$\|Fv\|_2^2 \geq \sum_{j=1}^c v_j^2 \|F^j\|_2^2 \geq \frac{l^2}{n} \sum_{j=1}^c v_j^2 = \frac{l^2}{n}. \tag{125}$$

Therefore, the smallest singular value of the matrix $F$ satisfies:

$$\sigma_{\min}(F) \geq \frac{l}{\sqrt{n}}. \tag{126}$$

Since $W$ is a symmetric matrix, its singular values are the absolute values of its eigenvalues, i.e., $\sigma_i(W) = |\lambda_i(W)|$. Using the singular value inequality for matrix products, we have:

$$\sigma_{\min}(F^T W F) \geq \sigma_{\min}(F)^2 \cdot \sigma_{\min}(W). \tag{127}$$

Substituting the previously derived $\sigma_{\min}(F) \geq \frac{l}{\sqrt{n}}$, $\quad \sigma_{\min}(W) = \min_i |\lambda_i(W)|$ we obtain

$$\sigma_{\min}(F^T W F) \geq \left(\frac{l}{\sqrt{n}}\right)^2 \cdot \sigma_{\min}(W) = \frac{l^2}{n} \cdot \sigma_{\min}(W). \tag{128}$$

Furthermore, the upper bound for the spectral norm of the inverse matrix can be estimated as:

$$\|(F^T W F)^{-1}\|_\circledS = \frac{1}{\sigma_{\min}(F^T W F)} \leq \frac{n}{\sigma_{\min}(W)l^2} \equiv \frac{1}{\alpha} \tag{129}$$

and the $\alpha$ can be presented as

$$\alpha = \frac{\sigma_{\min}(W)l^2}{n}. \tag{130}$$

Using the same proof method as in A.9, we provide the gradient expression for the general graph cut objective function as:

$$\text{Grad}\mathcal{H}(F) = 2\left(LF(F^T W F)^{-1} - WF(F^T W F)^{-1}(F^T LF)(F^T W F)^{-1}\right), \tag{131}$$

and with the above technique, we can estimate its nuclear norm upper bound.

For $\|LF(F^T W F)^{-1}\|_\circledS$ Using the sub-multiplicativity of the spectral norm ($\|AB\|_\circledS \leq \|A\|_\circledS \cdot \|B\|_\circledS$):

$$\|LF(F^T W F)^{-1}\|_\circledS \leq \|L\|_\circledS \cdot \|F\|_\circledS \cdot \|(F^T W F)^{-1}\|_\circledS \tag{132}$$

Substituting the known upper bounds:

$$\|LF(F^T W F)^{-1}\|_\circledS \leq \|L\|_\circledS \cdot \|F\|_\circledS \cdot \|(F^T W F)^{-1}\|_\circledS = \|L\|_\circledS \cdot \|F\|_\circledS \cdot \frac{1}{\sigma_{\min}(F^T W F)} \tag{133}$$

$$\leq \|L\|_\circledS \cdot \|F\|_\circledS \cdot \frac{n}{\sigma_{\min}(W)l^2} = \frac{\|L\|_\circledS \cdot \|F\|_\circledS}{\alpha} \leq \frac{\|L\|_\circledS \cdot \sqrt{n}}{\alpha} \tag{134}$$

Next, we consider the second term $WF(F^T W F)^{-1}(F^T LF)(F^T W F)^{-1}$. This term can be decomposed into four parts, namely:

$$\|WF(F^T W F)^{-1}(F^T LF)(F^T W F)^{-1}\|_\circledS \leq \|WF\|_\circledS \cdot \|(F^T W F)^{-1}\|_\circledS \cdot \|F^T LF\|_\circledS \cdot \|(F^T W F)^{-1}\|_\circledS \tag{135}$$

For $\|WF\|_\circledS$, we have the following inequality:

$$\|WF\|_\circledS \leq \|W\|_\circledS \cdot \|F\|_\circledS \leq \|W\|_\circledS \cdot \sqrt{n}. \tag{136}$$

For $\|F^T L F\|_{\circledS}$, we have the following inequality:

$$\|F^T L F\|_{\circledS} \leq \|F^T\|_{\circledS} \cdot \|L\|_{\circledS} \cdot \|F\|_{\circledS} = \|F\|_{\circledS} \cdot \|L\|_{\circledS} \cdot \|F\|_{\circledS} \leq \|L\|_{\circledS} \cdot n. \tag{137}$$

Combining our estimates with the previous inequality, we obtain:

$$\|WF(F^T W F)^{-1}(F^T L F)(F^T W F)^{-1}\|_{\circledS} \tag{138}$$

$$\leq \|WF\|_{\circledS} \cdot \|(F^T W F)^{-1}\|_{\circledS} \cdot \|F^T L F\|_{\circledS} \cdot \|(F^T W F)^{-1}\|_{\circledS} \tag{139}$$

$$\leq \|W\|_{\circledS} \cdot \sqrt{n} \cdot \|L\|_{\circledS} \cdot n \cdot \left(\frac{1}{\sigma_{\min}(F^T W F)}\right)^2 \leq \|W\|_{\circledS} \cdot \sqrt{n} \cdot \|L\|_{\circledS} \cdot n \cdot \left(\frac{n}{\sigma_{\min}(W) l^2}\right)^2 \tag{140}$$

$$= \frac{\|W\|_{\circledS} \cdot \|L\|_{\circledS} \cdot n^{7/2}}{\sigma_{\min}^2(W) l^4} = \frac{\|W\|_{\circledS} \cdot \|L\|_{\circledS} \cdot n^{3/2}}{\alpha^2}. \tag{141}$$

In summary, we have

$$\|\mathrm{Grad}\mathcal{H}(F)\|_{\circledS} \leq 2\left(\frac{\|L\|_{\circledS}\sqrt{n}}{\alpha} + \frac{\|W\|_{\circledS}\|L\|_{\circledS}n^{3/2}}{\alpha^2}\right), \tag{142}$$

where

$$\alpha = \frac{\sigma_{\min}(W) \cdot l^2}{n}. \tag{143}$$

Since

$$\|\mathrm{Grad}\mathcal{H}(F)\|_F \leq \sqrt{\min(n, c)} \|\mathrm{Grad}\mathcal{H}(F)\|_{\circledS}, \tag{144}$$

it follows that $\|\mathrm{Grad}\mathcal{H}(F)\|_F$ is also bounded.

In particular, for the Ratio Cut, we know that $W = I$ is the identity matrix. Therefore,

$$\|\mathrm{Grad}\mathcal{H}_m(F)\|_{\circledS} \leq 2\left(\frac{\|L\|_{\circledS}\sqrt{n}}{\alpha} + \frac{\|L\|_{\circledS}n^{3/2}}{\alpha^2}\right), \alpha = \frac{l^2}{n}. \tag{145}$$

Furthermore, since

$$\mathrm{grad}\,\mathcal{H}_r(F) = \mathrm{Grad}_r\,\mathcal{H}(F) - \frac{1}{c}\,\mathrm{Grad}_r\,\mathcal{H}(F) 1_c 1_c^T, \tag{146}$$

it is clear that $\mathrm{grad}\,\mathcal{H}_r(F)$ is also bounded. An obvious bound is given by

$$\|\mathrm{grad}\,\mathcal{H}_r(F)\|_{\circledS} \leq \|\mathrm{Grad}\,\mathcal{H}_r(F)\|_{\circledS} + \frac{1}{c}\left(\|\mathrm{Grad}\,\mathcal{H}_r(F)\|_{\circledS} \cdot \|1_c 1_c^T\|_{\circledS}\right), \tag{147}$$

which leads to

$$\|\mathrm{grad}\,\mathcal{H}_r(F)\|_{\circledS} \leq 2\left(\frac{\|L\|_{\circledS}\sqrt{n}}{\alpha} + \frac{\|L\|_{\circledS}n^{3/2}}{\alpha^2}\right) + \frac{1}{c}\left(2\left(\frac{\|L\|_{\circledS}\sqrt{n}}{\alpha} + \frac{\|L\|_{\circledS}n^{3/2}}{\alpha^2}\right) + \sqrt{nc}\right) \tag{148}$$

$$= \left(2 + \frac{2}{c}\right)\left(\frac{\|L\|_{\circledS}\sqrt{n}}{\alpha} + \frac{\|L\|_{\circledS}n^{3/2}}{\alpha^2}\right) + \sqrt{\frac{n}{c}} \tag{149}$$

where $\alpha = \frac{l^2}{n}$.

### A.11 PROOF OF THEOREM 7

**Theorem 7.** For a general graph cut problem expressed as $\mathcal{H}(F) = \mathrm{tr}((F^T L F)(F^T W F)^{-1})$, where $W$ is an arbitrary symmetric matrix, the problem is always Lipschitz smooth. Let the corresponding smoothness Lipschitz constant be $Q$. When applying Riemannian Gradient Descent (RIMRGD) on the RIM manifold with step size $\kappa$, if $\kappa \leq \frac{1}{Q}$, then $\mathcal{H}(F)$ converges to a critical point at a rate of $\mathcal{O}(\frac{1}{T})$, i.e.,

$$\min_{0 \leq k \leq T} \left\|\mathrm{grad}\,\mathcal{H}(F^{(k)})\right\|^2 \leq \frac{2\left(\mathcal{H}(F^{(0)}) - \mathcal{H}(F^*)\right)}{\kappa(T+1)}, \tag{150}$$

where $T$ is the total number of iterations, and $\mathcal{H}(F^*)$ is the global minimum of $\mathcal{H}(F)$.

*Proof.* For a general graph cut problem, similar to Theorem A.9, the expression of the Euclidean Hessian mapping can be given.

$$\text{Hess}\mathcal{H}[V] = 2\big(LVJ - LFJ\text{sym}(V^TWF)J - WVJKJ \tag{151}$$

$$+ AFJ\text{sym}(V^TWF)JKJ - WFJ\text{sym}(V^TWF)J \tag{152}$$

$$+ WFJKJ\text{sym}(V^TWF)J\big) \tag{153}$$

Where $(F^TWF)^{-1} = J$ and $F^TLF = K$, and sym$(\cdot)$ denotes the symmetrization operation.

Similar to the previous discussion, we can decompose Hess$\mathcal{H}[V]$ into multiple parts:

$$||\text{Hess}\mathcal{H}[V]||_{\circledS} \leq 2\big(||LVJ||_{\circledS} + ||LFJ\text{sym}(V^TWF)J||_{\circledS} + ||WVJKJ||_{\circledS} \tag{154}$$

$$+ ||AFJ\text{sym}(V^TWF)JKJ||_{\circledS} + ||WFJ\text{sym}(V^TWF)J||_{\circledS} \tag{155}$$

$$+ ||WFJKJ\text{sym}(V^TWF)J||_{\circledS}\big) \tag{156}$$

So the spectral norm of each part is bounded. It is not difficult to prove that the spectral norm of Hess$\mathcal{H}[V]$ is also bounded. Furthermore, it can be shown that the Riemannian Hessian map hess$\mathcal{H}[V]$ is also bounded.

$$||\text{hess}\mathcal{H}[V]||_{\circledS} \leq ||\text{Hess}\mathcal{H}[V]||_{\circledS} + \frac{1}{c}||\text{Hess}\mathcal{H}[V]||_{\circledS} \cdot ||1_c^T 1_c||_{\circledS} \tag{157}$$

Since Theorem A.5 has already proven that we can obtain geodesics using Dijkstra's algorithm, in the subsequent proofs, we will directly assume the use of geodesics for the retraction process.

Since the Riemannian Hessian map is bounded, let its upper bound be $Q$. Using the retraction generated by the geodesic, we can expand the function $\mathcal{H}(F)$ as follows:

$$\mathcal{H}(R_F(V)) \leq \mathcal{H}(F) + \langle \text{grad}\mathcal{H}(F), V \rangle_F + \frac{Q}{2}||V||_F^2 \tag{158}$$

In the Riemannian Gradient Descent method on the RIM manifold (RIMRGD), by choosing $V = -\kappa \, \text{grad}\, \mathcal{H}(F^{(k)})$, and substituting it into the upper bound, we obtain:

$$\mathcal{H}(F^{(k+1)}) \leq \mathcal{H}(F^{(k)}) - \kappa ||\text{grad}\, \mathcal{H}(F^{(k)})||^2 + \frac{Q\kappa^2}{2}||\text{grad}\, \mathcal{H}(F^{(k)})||^2. \tag{159}$$

When the step size $\kappa \leq \frac{1}{Q}$, it simplifies to:

$$\mathcal{H}(F^{(k+1)}) \leq \mathcal{H}(F^{(k)}) - \frac{\kappa}{2}||\text{grad}\, \mathcal{H}(F^{(k)})||^2. \tag{160}$$

This indicates that at each iteration, the function value decreases by at least $\frac{\kappa}{2}||\text{grad}\, \mathcal{H}(F^{(k)})||^2$. Summing the descent over the first $k$ iterations yields:

$$\sum_{i=0}^{k} \frac{\kappa}{2}||\text{grad}\, \mathcal{H}(F^{(i)})||^2 \leq \mathcal{H}(F^{(0)}) - \mathcal{H}(F^{(k+1)}) \leq \mathcal{H}(F^{(0)}) - \mathcal{H}(F^*), \tag{161}$$

where $\mathcal{H}(F^*)$ is the infimum of $\mathcal{H}(F)$. Since the right-hand side is bounded, the series $\sum_{i=0}^{\infty} ||\text{grad}\, \mathcal{H}(F^{(i)})||^2$ converges, and thus

$$\lim_{k \to \infty} ||\text{grad}\, \mathcal{H}(F^{(k)})|| = 0. \tag{162}$$

From the inequality above, we obtain:

$$\min_{0 \leq k \leq T} ||\text{grad}\mathcal{H}(F^{(k)})||^2 \leq \frac{2(\mathcal{H}(F^{(0)}) - \mathcal{H}(F^*))}{\kappa(T+1)} \tag{163}$$

which implies a convergence rate of $O\left(\frac{1}{T}\right)$.

In addition, since the algorithm in Manopt adopts the Wolfe step size, we further provide a convergence proof of RIMRGD under the Wolfe step-size scheme. Moreover, based on our experiments, it usually yields numerical results consistent with those obtained using the Armijo step size.

Condition 1. Equation (157) shows that the Riemannian Hessian $hess$ is bounded. Therefore, we have $hess(F) \leq Q$. According to Lemma 3.5 (Retraction L-smooth) in (Kasai et al., 2018), there exists $L > 0$ such that

$$f(x_{t+1}) \leq f(x) + \langle \operatorname{grad} f(x), s \rangle + \tfrac{1}{2} L \|s\|^2, \quad x_{t+1} = R_x(s), \ s \in T_x \mathcal{M}. \tag{164}$$

Condition 2. We adopt the Wolfe step size, i.e.,

$$\begin{aligned} f(x + \kappa d) &\leq f(x) + c_1 \cdot \kappa \langle \operatorname{grad} f(x), d \rangle, \\ \langle \operatorname{grad} f(x + \kappa d), d \rangle &\geq c_2 \langle \operatorname{grad} f(x), d \rangle, \end{aligned} \tag{165}$$

where $0 < c_1 < c_2 < 1$ are hyperparameters.

Condition 3. The Ratio Cut loss is clearly lower bounded (according to the real interpretation of Ratio Cut).

Therefore, according to (Sato, 2021), the algorithm converges to a critical point.

### A.12 PROOF OF THEOREM 8

**Theorem 8.** Suppose we have an objective function $\mathcal{H}(F)$ that is $\mu$-strongly geodesically convex and Lipschitz smooth on the doubly-stochastic manifold $\{X | X > 0, X1_c = 1_n, X^T 1_n = r\}$. After relaxing it to the RIM manifold $\{X | X > 0, X1_c = 1_n, l < X^T 1_n < u\}$, let $F_1^*$ be the optimal solution on the RIM manifold and $F_2^*$ be the optimal solution on the original doubly stochastic manifold. Then we have:

$$\|F_1^* - F_2^*\| \leq \mathcal{O}\left(\frac{L}{\mu}\right) \cdot \max\left(\|r - l\|, \|u - r\|\right). \tag{166}$$

*Proof.* First, convexity is necessary. For a nonconvex problem no method can guarantee finding the global optimum, and moreover any locally optimal point may lie arbitrarily far from the global optimum, so no meaningful distance bound can be established.

Without loss of generality, assume $l < r < u$. In this case, $F_2^*$ is also a feasible point on the RIM manifold, and we can view $F_2^*$ as a perturbation of $F_1^*$ on the RIM manifold. Therefore, the problem reduces to a sensitivity analysis of convex optimization under perturbations. By strong convexity, we have:

$$\mathcal{H}(F_2^*) \geq \mathcal{H}(F_1^*) + \langle \operatorname{Grad}, \mathcal{H}(F_1^*), F_2^* - F_1^* \rangle + \frac{\mu}{2} \|F_2^* - F_1^*\|^2. \tag{167}$$

Rearranging terms gives:

$$\frac{\mu}{2} \|F_2^* - F_1^*\|^2 \leq \mathcal{H}(F_2^*) - \mathcal{H}(F_1^*). \tag{168}$$

Define the magnitude of the perturbation as $\Delta = \max(\|r - l\|, \|u - r\|)$. By sensitivity analysis (Boyd & Vandenberghe, 2004), we obtain:

$$\frac{\mu}{2} \|F_1^* - F_2^*\|^2 \leq \mathcal{H}(F_2^*) - \mathcal{H}(F_1^*) \leq L \cdot \operatorname{dist}(F_1^*, D) \tag{169}$$

where $D$ is the perturbation set induced by relaxing the constraint on $X^T 1_c = r$, and $\operatorname{dist}(F_1^*, D)$ denotes the maximum distance from $F_1^*$ to the perturbation set $D$. The size of $D$ scales proportionally with $\Delta^2 = (\max(\|r - l\|, \|u - r\|))^2$. This completes the proof.

# B  PRELIMINARIES

## B.1  NOTATIONS

Matrices are denoted by uppercase letters, while vectors are denoted by lowercase letters. Let $tr(\cdot)$ the trace of a matrix. $1_n$ denotes an $n$-dimensional column vector of all ones, and $\text{Ind}^{n \times c}$ represents the set of indicator matrices. If $F \in \text{Ind}^{n \times c}$, then $F \in \mathbb{R}^{n \times c}$ satisfies the property that each row contains exactly one element equal to 1, while all others are 0. The relaxed indicator matrix set is defined as $M = \{X \mid X1_c = 1_n, l < X^T 1_n < u, X > 0\}$, and we proved it can form a manifold $\mathcal{M}$. $T_X \mathcal{M}$ represents the tangent space of $\mathcal{M}$ at $X$. $\langle \cdot, \cdot \rangle$ denotes the Euclidean inner product, while $\langle \cdot, \cdot \rangle_X$ denotes the inner product on the manifold at $X$. $\mathcal{H}$ represents the objective function, $\text{Grad}\,\mathcal{H}$ denotes the Euclidean gradient of $\mathcal{H}$, and $\text{grad}\,\mathcal{H}$ denotes the Riemannian gradient of $\mathcal{H}$. $\text{Hess}\,\mathcal{H}(F)$ represents the Euclidean Hessian mapping, while $\text{hess}\,\mathcal{H}(F)$ represents the Riemannian Hessian mapping. $R_X$ denotes the Retraction function at $X$, which generates a curve passing through $X$, and $R_X(tV)$ represents a curve on the manifold obtained via the Retraction function, satisfying $\frac{d}{dt} R_X(0) = V$. The connection in Euclidean space is denoted as $\bar{\nabla}_V U$, while the connection on the manifold is denoted as $\nabla_V U$. The differential mapping is represented as $D\mathcal{H}(F)[V]$. Specifically, a geodesic $\gamma(t)$ is a curve on the manifold that extremizes the distance between two points. If $\frac{D}{dt} \gamma'(t) = 0$, then $\gamma(t)$ is a geodesic. $\mathcal{P}$ represents vector transport, which maps the tangent vector $V$ at point $X$ on the manifold to the tangent space $T_Y \mathcal{M}$ at another point $Y$.

We have compiled all the symbols used in this paper in Table 7, where their specific meanings are explained. Additionally, all Riemannian optimization-related symbols used in this paper follow standard conventions in the field and can also be referenced in relevant textbooks.

Table 7: Notations.

| Notation | Description |
|---|---|
| $\text{Ind}^{n \times c}$ | The set of $n \times c$ indicator matrices |
| $1_n, 1_c$ | All-ones column vectors of size $n$ or $c$ |
| $L$ | Laplacian matrix |
| $l, u$ | Lower and upper bounds of the column sum of the relaxed indicator matrix, both are $c$-dimensional column vectors |
| $\mathcal{M}$ | A set that forms a manifold |
| $\langle \cdot, \cdot \rangle$ | Inner product defined in Euclidean space, mapping two Euclidean vectors to a scalar |
| $\langle \cdot, \cdot \rangle_X$ | Inner product defined on the tangent space of $\mathcal{M}$ at $X$ |
| $T_X \mathcal{M}$ | Tangent space of the manifold $\mathcal{M}$ at $X$, which is a linear space |
| $\mathcal{H}$ | The objective function to be optimized |
| $\text{Grad}\,\mathcal{H}(F)$ | Euclidean gradient of $\mathcal{H}$ at $F$, i.e., the gradient in the embedding space |
| $\text{grad}\,\mathcal{H}(F)$ | Riemannian gradient of $\mathcal{H}$ at $F$ |
| $\bar{\nabla}_V U$ | Riemannian connection of the tangent vector field $U$ along $V$ in Euclidean space |
| $\nabla_V U$ | Riemannian connection of the tangent vector field $U$ along $V$ on the manifold |
| $\text{Hess}\,\mathcal{H}[V]$ | Riemannian Hessian mapping along tangent vector $V$ in Euclidean space |
| $\text{hess}\,\mathcal{H}[V]$ | Riemannian Hessian mapping along tangent vector $V$ on the manifold |
| $R_X(tV)$ | A curve on the manifold generated at $X$ along the tangent vector $tV$ |
| $\frac{d}{dt} R_X(tV)\big|_{t=0}$ | The derivative of $R_X(tV)$ at $t = 0$ |
| $\frac{D}{dt} \gamma'(t)\big|_{t=0}$ | Levi-Civita derivative of $\frac{d}{dt}\gamma(t)$ at $t = 0$, where $\frac{D}{dt}\gamma'(t)\big|_{t=0} = 0$ means $R_X(tV)$ generates a geodesic with parameter $t$ |
| $\text{argmin}(\cdot)$ | Returns the minimizer of an optimization problem |
| $\Omega_1, \Omega_2, \Omega_3$ | Linear submanifolds that require projection |
| $X_i$ | The $i$-th row of matrix $X$ |
| $X^j$ | The $j$-th column of matrix $X$ |
| $\text{Proj}_{\Omega_i}(X^j)$ | Orthogonal projection of the $j$-th column of matrix $X$ onto the set $\Omega_i$ |
| $\max(a, b)$ | Returns the maximum of $a$ and $b$ |
| $\min(a, b)$ | Returns the minimum of $a$ and $b$ |
| $\mathcal{L}$ | Lagrangian function for solving the optimization problem |
| $\|\cdot\|_F$ | Frobenius norm of a matrix |
| $\nu(t), \omega(t), \rho(t)$ | Lagrange multipliers in the optimization problem |
| $\frac{\partial \mathcal{L}}{\partial \nu}, \frac{\partial \mathcal{L}}{\partial \omega}, \frac{\partial \mathcal{L}}{\partial \rho}$ | Partial derivatives of $\mathcal{L}$ with respect to $\nu(t), \omega(t), \rho(t)$ |
| $\exp(\cdot)$ | Element-wise exponential function on a matrix |
| $\text{diag}(\cdot)$ | Converts a vector into a diagonal matrix |
| $D\mathcal{H}(F)[V]$ | The differential mapping of $\mathcal{H}$ at $F$ along $V$ |
| $\mathcal{S}(\cdot)$ | Sinkhorn function that outputs a doubly stochastic matrix |
| $\mathcal{P}$ | Maps the tangent vector $V$ at point $X$ on the manifold to the tangent space $T_Y \mathcal{M}$ at another point $Y$ |
| $(\cdot)^\dagger$ | Moore-Penrose pseudoinverse of a matrix |
| $tr(\cdot)$ | Trace of a matrix |
| $\oslash$ | Element-wise division |
| $\odot$ | Hadamard product (element-wise multiplication) |

## B.2 Introduction to Riemannian Optimization

Riemannian optimization optimizes functions over Riemannian manifolds, which are smooth manifolds equipped with a metric that defines distance and angles (Meghwanshi et al., 2018). It extends classical optimization to non-Euclidean spaces by replacing the Euclidean gradient with the Riemannian gradient and so on. Introduced in the 1990s in control theory and signal processing (Edelman et al., 1998; Overton & Womersley, 1995), it has since been widely adopted in machine learning, computer vision, and data science due to its ability to handle geometric constraints (Carson et al., 2017; Khan & Maji, 2021; Boumal, 2023).

The core idea is to respect the manifold's geometry during optimization. Unlike classical methods that assume Euclidean space, Riemannian optimization accounts for curvature. Early methods used steepest descent, while later developments introduced second-order methods like Riemannian conjugate gradient and Newton methods for faster convergence. Recent advancements have expanded this framework to more complex manifolds, such as Stiefel manifold.

The main advantage of Riemannian optimization lies in its ability to perform optimization directly on the manifold, ensuring that the constraints inherent to the problem are naturally respected. For example, in low-rank matrix factorization, the optimization occurs on the Stiefel manifold $\mathcal{S}t = \{X \in \mathbb{R}^{n \times k} \mid X^T X = I_k\}$, where $I_k$ is the identity matrix of size $k$, naturally respecting the orthogonality constraints of the factor matrices.

In Riemannian submanifold of Euclidean space, the Riemannian gradient $\text{grad}\mathcal{H}(F)$ at a point $F \in \mathcal{M}$ is defined as the projection of the Euclidean gradient onto the tangent space of the manifold:

$$\text{grad}\mathcal{H}(F) = \text{Proj}_{T_F \mathcal{M}} \text{Grad}\mathcal{H}(F) \tag{170}$$

This ensures that the optimization process stays within the manifold, preserving its geometric structure.

To solve optimization problems efficiently on manifolds, key operations include the Riemannian gradient, which is used in gradient-based methods. The gradient descent update rule is:

$$F^{(k+1)} = R_{F^{(k)}}(-\alpha^{(k)} \text{grad}\mathcal{H}(F^{(k)})) \tag{171}$$

where $R_F$ is the Retraction map, and $\alpha_k$ is the step size at iteration $k$. The purpose of the Retraction is to update along a curve in the manifold in a specified direction.

For second-order optimization, the Riemannian Hessian $\text{hess}\mathcal{H}(F)$ is needed. The Hessian captures the curvature of the manifold and provides more information about the local behavior of the function. The Riemannian Hessian is defined as:

$$\text{hess}\mathcal{H}(F)[V] = \nabla_V \text{grad}\mathcal{H}(F) \tag{172}$$

for any tangent vector $V \in T_F \mathcal{M}$, and is used in more sophisticated optimization algorithms to accelerate convergence.

A geodesic is a curve that connects two points on a manifold with an extremal distance, are also important in Riemannian optimization. They are used to guide the optimization process along the manifold and are defined by the differential equation:

$$\frac{d^2}{dt^2}\gamma(t) + \Gamma(\gamma(t), \dot{\gamma}(t)) = 0 \tag{173}$$

where $\Gamma$ are the Christoffel symbols that encode the manifold's curvature (Boumal, 2014; Smirnov, 2021).

The Retraction map $R_X(tV)$ is used to map from the tangent space back onto the manifold after each iteration. A common Retraction map is the exponential map (Kochurov et al., 2020; Sun et al., 2019), which can generate a geodesic.

Riemannian optimization (Yuan et al., 2026) efficiently handles manifold structures, avoiding artificial constraints and leading to faster algorithms. Second-order methods like Riemannian conjugate gradient (RCG) and Newton methods further improve convergence by utilizing curvature information. The approach is versatile, extending to manifolds such as the Stiefel, Grassmannian, and the Relaxed Indicator Matrix (RIM) manifold, which generalizes both single and double stochastic manifolds.

Overall, Riemannian optimization has become a crucial tool in solving large-scale, constrained optimization problems, particularly in machine learning, computer vision, and robotics, due to its ability to manage manifold-valued data and complex constraints.

### B.3 INTRODUCTION TO RELATED MANIFOLDS

In this section, we will introduce the single stochastic manifold, the doubly stochastic manifold, and the Stiefel manifold. For each of these manifolds, we will provide their basic definitions and discuss optimization methods on these manifolds.

### B.3.1 SINGLE STOCHASTIC MANIFOLD

The single stochastic manifold (Sun et al., 2015; Saberi-Movahed et al., 2024) consists of matrices where each element is greater than zero and the row sums are equal to one, denoted as $\{X \mid X > 0, X1_c = 1_n\}$, with a dimension of $(n-1)c$. The tangent space of a manifold $\mathcal{M}$ at a point $X$ is given by $T_X\mathcal{M} = \{U \mid X1_c = 0\}$.

In current research, the Fisher information metric is typically used as the inner product on the single stochastic manifold $\mathcal{M}$, and is defined as:

$$< U, V >_X = \sum_i \sum_j \frac{U_{ij}V_{ij}}{X_{ij}}, \quad \forall U, V \in T_X\mathcal{M}, X \in \mathcal{M}. \tag{174}$$

The Riemannian gradient $\operatorname{grad}\mathcal{H}(F)$ is the projection of the Euclidean gradient $\operatorname{Grad}\mathcal{H}(F)$:

$$\operatorname{grad}\mathcal{H}(F) = \operatorname{Proj}_{T_F\mathcal{M}}(\operatorname{Grad}\mathcal{H}(F) \odot F) \tag{175}$$

where $\operatorname{Proj}_{T_F\mathcal{M}}$ is the projection operator that projects vectors from the Euclidean space onto $T_F\mathcal{M}$. Specifically, the projection is given by:

$$\operatorname{Proj}_{T_X\mathcal{M}}(Z) = Z - (\alpha 1_c^T) \odot X, \quad \alpha = Z1_c \in \mathbb{R}^n \tag{176}$$

This projection operation involves matrix multiplication and element-wise operations, with a complexity of $\mathcal{O}(nc)$.

In the single stochastic manifold, the Retraction mapping $R_X(tV)$ is defined as:

$$X_+ = R_X(tV) = (X \odot \exp(tV \oslash X)) \oslash (X \odot \exp(V \oslash X)1_c1_c^T),$$

where the operation $\odot$ denotes element-wise multiplication, and $\oslash$ denotes element-wise division. The time complexity of this operation involves element-wise computation and normalization, resulting in a complexity of $\mathcal{O}(nc)$.

In the embedded space, the connection is considered with the Fisher metric on the set $\{X|X > 0\}$. According to the Koszul formula theorem, the unique connection in the embedded space is given by:

$$\bar{\nabla}_U V = DV[U] - \frac{1}{2}(U \odot V) \oslash X \tag{177}$$

Based on this, the unique connection on the manifold that makes the Riemannian Hessian mapping self - adjoint is:

$$\nabla_U V = \operatorname{Proj}_{T_X\mathcal{M}}(\bar{\nabla}_U V) = \operatorname{Proj}_{T_X\mathcal{M}}\left(DV[U] - \frac{1}{2}(U \odot V) \oslash X\right) \tag{178}$$

When involving directional derivatives and projections, the complexity of the operation is $O(nc)$.

By computing the connection of the Riemannian gradient, one can obtain the Riemannian Hessian mapping on the manifold. The Riemannian Hessian $\operatorname{hess}\mathcal{H}(F)[V]$ is

$$\operatorname{hess}\mathcal{H}(F)[V] = \operatorname{Proj}_{T_F\mathcal{M}}\left(D\operatorname{grad}\mathcal{H}(F)[V] - \frac{1}{2}(V \odot \operatorname{grad}\mathcal{H}(F)) \oslash F\right) \tag{179}$$

where the computation of $D\operatorname{grad}\mathcal{H}(F)[V]$ involves the Euclidean directional derivative:

$$D\operatorname{grad}\mathcal{H}(F)[V] = D\operatorname{Grad}\mathcal{H}(F)[V] \odot F + \operatorname{Grad}\mathcal{H}(F) \odot V - (\alpha 1_c^T) \odot V - (D\alpha[V]1_c^T) \odot F \tag{180}$$

where $\alpha = (\operatorname{Grad}\mathcal{H}(F) \odot F)1_c$. The time complexity of this computation involves higher-order derivatives and projections, leading to a complexity of $\mathcal{O}(nc)$. Due to the complexity of the computation, the coefficient in front of $\mathcal{O}(nc)$ is large.

### B.3.2 DOUBLY STOCHASTIC MANIFOLD

The double stochastic manifold (Shi et al., 2021; Douik & Hassibi, 2019) refers to the set of matrices where each element is greater than 0, the row sums equal 1, and the column sums equal $r$. Specifically, the manifold is defined as:

$$\{X \mid X > 0, X1_c = 1_n, X^T 1_n = r\} \tag{181}$$

with dimension $(n-1)(c-1)$. In fact, there are requirements for $r$. The more general definition is as follows.

$$\{X \mid X > 0, X1_c = 1_n, X^T 1_n = r, r^T 1_c = 1_n^T X 1_c\} \tag{182}$$

where $r$ is a general vector and the last condition ensures consistency of row and column sums. Generally, we simply denote it as (181). The tangent space of the manifold $\mathcal{M}$ at $X$ is:

$$T_X \mathcal{M} = \{U \mid X1_c = 0, X^T 1_n = 0\} \tag{183}$$

In current research, the Fisher information metric is also used as the inner product on the double stochastic manifold $\mathcal{M}$, defined as: $\langle U, V \rangle_X = \sum_i \sum_j \frac{U_{ij} V_{ij}}{X_{ij}}, \quad \forall U, V \in T_X \mathcal{M}, X \in \mathcal{M}$. The Riemannian gradient on the double stochastic manifold is given by (n=c):

$$\begin{cases} \operatorname{grad} \mathcal{H}(F) = \gamma - \left(\alpha 1_n^T + 1_n 1_n^T \gamma - 1_n \alpha^T F\right) \odot F, \\ \alpha = \left(I - FF^T\right)^\dagger \left(\gamma - F\gamma^T\right) 1_n, \quad \gamma = \operatorname{Grad} \mathcal{H}(F) \odot F. \end{cases} \tag{184}$$

Here, $(I - FF^T)^\dagger$ represents the Moore-Penrose pseudoinverse of an $n \times n$ matrix. Since computing the pseudoinverse requires at least $\mathcal{O}(n^3)$ operations, this method is impractical for large-scale datasets.

The connection on the double stochastic manifold is defined as an embedded manifold, and in the embedding space, the connection is given by $\bar{\nabla}_U V = DV[U] - \frac{1}{2}(U \odot V) \oslash X$. Further, the connection on the double stochastic manifold is given by $\operatorname{Proj}_{T_X \mathcal{M}}(\bar{\nabla}_U V) = \operatorname{Proj}_{T_X \mathcal{M}} \left(DV[U] - \frac{1}{2}(U \odot V) \oslash X\right)$.

$\operatorname{Proj}_{T_X \mathcal{M}}$ denotes the projection into the tangent space of the double stochastic manifold. The projection expression is:

$$\begin{cases} \operatorname{Proj}_{T_X \mathcal{M}}(Z) = Z - \left(\alpha 1_n^T + 1_n \beta\right) \odot X, \\ \alpha = \left(I - XX^T\right)^\dagger \left(Z - XZ^T\right) 1_n, \quad \beta = Z^T 1_n - X^T \alpha. \end{cases} \tag{185}$$

Indeed, the Riemannian Hessian mapping calculation in the referenced literature involves very complex expressions, including pseudoinverses and other operations with a time complexity of $\mathcal{O}(n^3)$, making it infeasible for large-scale datasets. In contrast, the proposed RIM manifold in this paper simplifies the calculation significantly, reducing the complexity to $\mathcal{O}(n)$.

The Riemannian Hessian is computed as follows:

$$\begin{aligned} \operatorname{hess} \mathcal{H}(F)[V] &= \operatorname{Proj}_{T_X \mathcal{M}} \left(\dot{\delta} - \frac{1}{2}\left(\delta \odot V\right) \oslash F\right) \\ \alpha &= \epsilon \left(\gamma - F\gamma^T\right) 1_n \\ \beta &= \gamma^T 1_n - F^T \alpha \\ \gamma &= \operatorname{Grad} \mathcal{H}(F) \odot F \\ \delta &= \gamma - \left(\alpha 1_n^T + 1_n \beta^T\right) \odot F \\ \epsilon &= \left(I - FF^T\right)^\dagger \\ \dot{\alpha} &= \left[\dot{\epsilon} \left(\gamma - F\gamma^T\right) + \epsilon \left(\dot{\gamma} - V\gamma - F\dot{\gamma}^T\right)\right] 1_n \\ \dot{\beta} &= \dot{\gamma}^T 1_n - V^T \alpha - F^T \dot{\alpha} \\ \dot{\gamma} &= \operatorname{Hess} \mathcal{H}(F)[V] \odot F + \operatorname{Grad} \mathcal{H}(F) \odot V \\ \dot{\delta} &= \dot{\gamma} - \left(\dot{\alpha} 1_n^T + 1_n \dot{\beta}^T\right) \odot F - \left(\alpha 1_n^T + 1_n \beta^T\right) \odot V \\ \dot{\epsilon} &= \epsilon \left(FV^T + VF^T\right) \epsilon \end{aligned} \tag{186}$$

The Retraction map uses Sinkhorn to obtain the doubly stochastic matrix. The time complexity of optimization on the doubly stochastic manifold is large, with a constant term of $\mathcal{O}(n^3)$. The aboved formulas is suitable for the case where $n = c$. However, when $n \neq c$, the calculation formula differs slightly, but the time complexity remains the same.

### B.3.3 STIEFEL MANIFOLD

The Stiefel manifold (Jiang & Dai, 2015; Li et al., 2020; Zhu, 2017) is the set of all matrices whose columns are orthonormal, i.e.,

$$\mathcal{St}(n, c) = \{X \in \mathbb{R}^{n \times c} \mid X^T X = I\}. \tag{187}$$

It can be proven that this set satisfies the requirements for a manifold, and the dimension of this manifold is given by:

$$\dim(\mathcal{St}(n, c)) = nc - \frac{c(c+1)}{2}. \tag{188}$$

At $X \in \mathcal{St}$, the tangent space of the Stiefel manifold is given by:

$$T_X \mathcal{St} = \{Z \mid Z^T X + X^T Z = 0\}. \tag{189}$$

Since the Stiefel manifold is an embedded submanifold of $\mathbb{R}^{n \times c}$, its Riemannian inner product is defined as the Euclidean inner product $\langle U, V \rangle_X = \sum_{ij} U_{ij} V_{ij}$.

The projection operator onto the tangent space $T_X \mathcal{St}$ is given by:

$$\begin{cases} \text{Proj}_{T_X \mathcal{St}}(Z) = (\hat{W} - \hat{W}^T)X, \\ \hat{W} = ZX^T - \frac{1}{2}X(X^T Z X^T). \end{cases} \tag{190}$$

Based on this, the Riemannian gradient can be directly obtained by projecting the gradient.

$$\text{grad}\,\mathcal{H}(F) = \text{Proj}_{T_F \mathcal{St}}(\text{Grad}\,\mathcal{H}(F)) = (\hat{W} - \hat{W}^T)F, \quad \hat{W} = \text{Grad}\,\mathcal{H}(F)F^T - \frac{1}{2}F(F^T\,\text{Grad}\,\mathcal{H}(F)F^T) \tag{191}$$

To compute the Retraction on the Steifel manifold, the Cayley transform method is used, given by:

$$Y(\alpha) = \left(I - \frac{\alpha}{2}W\right)^{-1}\left(I + \frac{\alpha}{2}W\right)X \tag{192}$$

Where $W = \hat{W} - \hat{W}^T$, $\alpha$ is the length on the curve. However, the inversion of $\left(I - \frac{\alpha}{2}W\right)$ is computationally expensive. To address this, Li et al. (2020) further attempts to use an iterative approach to find the solution. The Retraction is obtained by iteratively solving the following equation:

$$Y(\alpha) = X + \frac{\alpha}{2}W(X + Y(\alpha)) \tag{193}$$

Even so, each iteration still requires multiple matrix multiplications, resulting in a relatively high computational cost.

To obtain the momentum gradient descent on the Riemannian manifold, it is necessary to define the vector transport, which moves a tangent vector $V_1 \in T_{X_1}\mathcal{St}$ from the Steifel manifold at $X_1$ to the tangent space $T_{X_2}\mathcal{St}$ at $X_2$. This transport operation is denoted as:

$$\mathcal{P} : T_{X_1}\mathcal{St} \to T_{X_2}\mathcal{St}, \quad \forall V_1 \in T_{X_1}, \mathcal{P}(V_1) \in T_{X_2}\mathcal{St}. \tag{194}$$

In fact, this transport operation is general in its definition for manifolds. For the Relaxed Indicator Matrix (RIM) manifold, $T_{X_1}\mathcal{M} = T_{X_2}\mathcal{M}$ for all $X_1, X_2 \in \mathcal{M}$, which means that the vector transport is simply $\mathcal{P}(V_1) = V_1$ in the RIM manifold. However, this property does not hold on the Steifel manifold. The transport formula on the Steifel manifold is given by:

$$\mathcal{P}(V_1) = \text{Proj}_{T_{X_2}\mathcal{St}}(V_1) = (\hat{W} - \hat{W}^T)X_2, \tag{195}$$

where $\hat{W} = V_1 X_2^T - \frac{1}{2}X_2(X_2^T V_1 X_2)$, ensuring that the vector is properly projected into the tangent space at $X_2$. This projection step ensures the transfer of the vector $V_1$ from the tangent space at $X_1$ to the tangent space at $X_2$ on the Steifel manifold.

As for the computation of the connection and the Riemannian mapping matrix, although the literature does not provide explicit expressions, it can be proven that the expressions for the connection and Hessian map are as follows:

$$\begin{cases} \nabla_U V = \text{Proj}_{T_{X_2}\mathcal{St}}(DV[U]), \\ \text{hess}\mathcal{H}(F)[V] = \text{Proj}_{T_{X_2}\mathcal{St}}(\text{Hess}\mathcal{H}(F)[V]). \end{cases} \tag{196}$$

Using the above Riemannian toolbox, Riemannian optimization can be performed on the Steifel manifold. If the closed-form solution for the Retraction is directly computed, the time complexity is $\mathcal{O}(n^2 c)$. However, by using an iterative approach, the time complexity can be reduced to a large constant factor of $\mathcal{O}(n^2)$.

## B.4 MANIFOLD-BASED MACHINE LEARNING ALGORITHMS

In this section, we will introduce some classical machine learning algorithms defined on the Single stochastic, Double stochastic, and Steifel manifolds. In general, we assume the data matrix is $Z$, where $Z \in \mathbb{R}^{n \times k}$ with $n$ samples and $k$ features. Each row of $Z$ represents a sample, and $z_i$ denotes the $i$-th row of $Z$.

### B.4.1 ALGORITHMS ON THE SINGLE STOCHASTIC MANIFOLD

Fuzzy K-means (Fuzzy C-means, FCM) (Sulaiman & Isa, 2010) is an extension of the traditional K-means algorithm that allows data points to belong to multiple clusters with degrees of membership, rather than being strictly assigned to a single cluster. The core idea is to describe the relationship between data points and clusters through a membership matrix, which is suitable for clustering data with fuzzy boundaries.

Let the number of clusters be $c$, and the membership matrix $U \in \mathbb{R}^{c \times n}$, where $u_{ij}$ represents the membership degree of the $j$-th data point in the $i$-th cluster. The cluster centers are denoted as $C = \{c_1, c_2, ..., c_c\}$. The optimization goal is to minimize the following objective function:

$$J(U,C) = \sum_{i=1}^{c} \sum_{j=1}^{n} u_{ij}^m \|z_j - c_i\|^2 \tag{197}$$

The constraints are that the sum of the membership degrees for each data point equals 1: $\sum_{i=1}^{c} u_{ij} = 1 \quad (\forall j = 1, 2, ..., n)$, and the membership degrees are non-negative: $u_{ij} \in [0, 1]$. Where $m > 1$ is the fuzziness coefficient, which controls the degree of fuzziness in the clustering and $\|z_j - c_i\|$ is the Euclidean distance between data point $z_j$ and cluster center $c_i$. Thus, the final objective function and constraints can be written as:

$$\min \quad J(U,C) \quad \text{s.t.} \quad U \in \{X \in \mathbb{R}^{c \times n} \mid X > 0, X^T 1_c = 1_n\}, C \in \mathbb{R}^{c \times k} \tag{198}$$

This optimization problem is defined over the Cartesian product of the single stochastic manifold and the Euclidean space, which still constitutes a form of a single stochastic manifold.

### B.4.2 ALGORITHMS ON THE DOUBLE STOCHASTIC MANIFOLD

ANCMM (Yuan et al., 2024c) is a method for solving constrained problems on the double stochastic manifold, which can achieve adaptive neighbor clustering. Its objective function is given by:

$$\min_{S \in \mathbb{R}^{n \times n}} \sum_{i,j}^{n} \|z_i - z_j\|_2^2 S_{ij} + \alpha \|S\|_F^2 \tag{199}$$

$$\text{s.t. } S^T 1_n = 1_n, \ 0 \le s_{ij} \le 1, \ S = S^T, \ \text{rank}(L_S) = n - c$$

where $S$ is the similarity matrix, and $S_{ij}$ represents the similarity between the $i$-th and $j$-th samples. The constraint can be written as:

$$\{X \in \mathbb{R}^{n \times n} \mid X 1_n = 1_n, X^T 1_n = 1_n, X > 0\} \cap \{X \in \mathbb{R}^{n \times n} \mid X = X^T, L_S = n - c\} \tag{200}$$

where $L_S$ is the Laplacian matrix corresponding to $S$, and $L_S = n - c$ implies that the learned $S$ is naturally $c$-connected, leading to $c$ clusters. Thus, this problem can be viewed as a constrained optimization problem on the double stochastic manifold.

### B.4.3 ALGORITHMS ON THE STEIFEL MANIFOLD

The Min Cut (Fox et al., 2023) is a classic clustering method on the Steifel manifold, and its objective function and constraints are given by:

$$\min_F \text{tr}(F^T L F), \quad \text{s.t.} \ F \in \{F \in \mathbb{R}^{n \times c} \mid F^T F = I\} \tag{201}$$

This optimization problem can be solved through eigenvalue decomposition. However, it requires approximately $\mathcal{O}(n^2 c)$ time complexity, and eigenvalue decomposition alone does not provide clustering results. Additional post-processing, such as using k-means, is required. Similarly, the derived classic methods such as Ratio Cut and Normalized Cut are also classic machine learning

algorithms on the Steifel manifold. The expressions for Ratio Cut and Normalized Cut are as follows:

$$\begin{cases} \min_F \text{tr}(F^T L F (F^T F)^{-1}), & \text{s.t. } F \in \{F \in \mathbb{R}^{n \times c} \mid F^T F = I\} \\ \min_F \text{tr}(F^T L F (F^T D F)^{-1}), & \text{s.t. } F \in \{F \in \mathbb{R}^{n \times c} \mid F^T F = I\} \end{cases} \tag{202}$$

In addition, algorithms such as MinMax Cut (Nie et al., 2010), Principal Component Analysis (PCA) (Abdi & Williams, 2010), Robust PCA (Hubert et al., 2005), and others are also classic machine learning algorithms defined on the Steifel manifold.

### B.5 OTHER RELATED WORK AND BACKGROUND INTRODUCTION

In this section, we first review our contributions and then introduce other related work beyond manifold optimization.

As mentioned in our paper, there are currently three main approaches to relaxing the indicator matrix (ours being the fourth). For the first three, the optimization methods themselves have seen little change, but have instead been applied to different models. For example:

The earliest approach relaxes to the singly stochastic manifold (Bezdek et al., 1979), which actually has a history of more than 45 years. More recent applications in clustering include (Bao et al., 2024), which employs momentum methods to solve the constraint, and Zhao et al. (2022), which introduces auxiliary variables and updates via coordinate descent. The main drawback of this relaxation is its inability to incorporate prior information about class sizes into the model.

Another line of work relaxes to the Stiefel manifold, starting from (Ng et al., 2001), which spurred the development of spectral graph theory and has now a history of about 20 years. The basic idea is to construct forms like $\text{tr}(F^T L F)$ and perform spectral decomposition, as in (He et al., 2025). The limitation here is that the resulting $F$ lacks the interpretability of an indicator matrix, requiring a subsequent K-Means step, with a computational complexity of $\mathcal{O}(n^3)$. Moreover, this approach also cannot incorporate any class-related information.

A more recent direction is doubly stochastic relaxation, with representative work Fettal et al. (2024), which solves the problem via optimal transport, and Douik & Hassibi (2019), which adopts manifold optimization. The challenge here is that the constraints can be overly strict and counterproductive to the model, and manifold optimization still requires $\mathcal{O}(n^3)$.

Some works in optimal transport are also related to ours. For example, Chapel et al. (2020) introduces Partial Optimal Transport, which is a less strict form of optimal transport. This idea is similar to ours in spirit; however, our algorithm is designed for arbitrary functions defined on manifolds, whereas theirs focuses on classical linear problems.

In addition, Benamou et al. (2014) shows that optimal transport problems can be solved using Bregman Projections. This is close in spirit to the original motivation behind our Retraction design. We further demonstrate that our Retraction corresponds to a geodesic, while also simplifying the overall algorithmic procedure.

## C  OPTIMIZATION ALGORITHMS ON THE RIM MANIFOLD

In this section, we will introduce three renowned Riemannian optimization algorithms that are utilized in this paper: the Riemannian Gradient Descent method, the Riemannian Conjugate Gradient method, and the Riemannian Trust-Region method. For each algorithm, we will present its fundamental concepts and provide pseudocode. For detailed implementations of these algorithms, one may refer to the open-source manifold optimization package, Manopt (Boumal et al., 2014).

### C.1  GRADIENT DESCENT ON THE RIM MANIFOLD

The Gradient Descent on the RIM Manifold method generalizes the classical gradient descent in Euclidean space to Riemannian manifolds by replacing the traditional gradient with the Riemannian gradient, ensuring that the iterations remain on the manifold. The key idea is to utilize the manifold's geometric structure to adjust the gradient direction, and then use Retraction to map the updated point back onto the manifold. The process begins with initialization, where an initial point $F_0$ is chosen on the manifold, and a step size is chosen. In the next step, the Euclidean gradient of the objective function is computed at the current point $F^{(k)}$. Then, the Euclidean gradient is projected onto the tangent space of the manifold to obtain the Riemannian gradient, which involves adjusting the gradient by subtracting the normal component. The updated point is then computed along the Riemannian gradient direction, and Retraction (such as exponential mapping or projection) is used to ensure that the new point remains on the manifold. The process continues iteratively until the gradient norm or the change in the objective function becomes smaller than a predefined threshold. The reference pseudo code is in Algorithm 3.

### C.2  CONJUGATE GRADIENT METHOD ON THE RIM MANIFOLD

The Conjugate Gradient Method on the RIM Manifold introduces conjugate directions to reduce the redundancy in search directions during iterations, thereby speeding up convergence by incorporating information from previous search directions. The core idea is to define and update conjugate directions on the manifold. The method begins with initialization, where the initial point $F_0$ is chosen, the initial Riemannian gradient $g_0$ is computed, and the initial search direction is set as $d_0 = -g_0$. Then, the optimal step size in the direction of $d_k$ is determined through a line search, using conditions like Armijo's rule. The point is updated along $d_k$, and Retraction is applied to map it back onto the manifold. In the next step, the conjugate direction is updated using the current gradient $g_{k+1}$ and the previous direction $d_k$, with formulas such as the Polak-Ribière method to compute the new conjugate direction $d_{k+1}$. On the RIM manifold, the transport of tangent vectors is equivalent to the vectors themselves. This property simplifies the process of the Riemannian Conjugate Gradient Method. The process is repeated until convergence is achieved. The reference pseudo code is in Algorithm 4.

### C.3  TRUST REGION METHOD ON THE RIM MANIFOLD

The Trust Region Method on the RIM Manifold constructs a local quadratic model in each iteration and constrains the step size within a trust region to ensure stability. The trust region radius is dynamically adjusted to balance the accuracy of the model with the step size. The method starts with initialization, where the initial point $F_0$ and trust region radius $\Delta_0$ are set. The Riemannian gradient $g_k$ and the approximate Hessian $H_k$ are computed at $F^{(k)}$. The next step involves solving the constrained quadratic optimization problem in the tangent space, given by:

$$\min_{d \in T_{F^{(k)}}\mathcal{M}, \|d\| \leq \Delta_k} \left( g_k^T d + \frac{1}{2} d^T H_k d \right) \qquad (203)$$

Following this, the method updates the point and adjusts the trust region radius $\Delta_k$ based on the ratio of the actual decrease in the objective function to the model's predicted decrease. Finally, Retraction is used to project the updated point back onto the manifold. This method is known for its strong stability and is particularly suited for highly nonlinear problems. However, it requires frequent Hessian

calculations, resulting in a high computational cost. The reference pseudo code is in Algorithm 5.

---

**Algorithm 3:** Riemannian Gradient Descent Algorithm on RIM Manifold

---

**Input:** RIM manifold $\mathcal{M} = \{X \mid X1_c = 1_n, l < X^T 1_n < u, X > 0\}$
Objective function $\mathcal{H}(F)$, Retraction $R_X(tV)$, transport $\mathcal{P}$. Initial point $F_0 \in \mathcal{M}$
**Output:** Sequence of iterates $\{F^{(k)}\}$ converging to a stationary point of $\mathcal{H}$

1   Initialize $k = 0$ **while** *not converged* **do**
2      Compute Euclidean gradient $\mathrm{Grad}\,\mathcal{H}(F^{(k)})$
3      Compute Riemannian gradient: $\mathrm{grad}\,\mathcal{H}(F^{(k)}) = \mathrm{Grad}\,\mathcal{H}(F^{(k)}) - \frac{1}{c}\,\mathrm{Grad}\,\mathcal{H}(F^{(k)})1_c 1_c^T$
4      The line search step size: $\kappa^{(k)}$
5      Perform Retraction: $F^{(k+1)} = R_{F^{(k)}}(\kappa^{(k)}\,\mathrm{grad}\,\mathcal{H}(F^{(k)}))$
6      $k \leftarrow k + 1$
7   **end**
8   **return** $F^{(k)}$

---

**Algorithm 4:** Riemannian Conjugate Gradient Algorithm on RIM Manifold

---

**Input:** RIM manifold $\mathcal{M} = \{X \mid X1_c = 1_n, l < X^T 1_n < u, X > 0\}$
Objective function $\mathcal{H}(F)$, Retraction $R_X(tV)$, Initial point $F_0 \in \mathcal{M}$.
**Output:** Sequence of iterates $\{F^{(k)}\}$ converging to a stationary point of $\mathcal{H}$

1   Initialize $k = 0$;
2   Compute initial Riemannian gradient, $d_0 \leftarrow -\,\mathrm{grad}\,\mathcal{H}(F^{(0)})$;
3   **while** *not converged* **do**
4      Compute line search step size $\kappa^{(k)}$
5      Perform Retraction: $F^{(k+1)} = R_{F^{(k)}}(\kappa^{(k)}d^{(k)})$
6      Compute new gradient $\mathrm{grad}\,\mathcal{H}(F^{(k+1)})$
7      Compute the conjugate direction $d^{(k+1)} = -\,\mathrm{grad}\,\mathcal{H}(F^{(k+1)}) + \beta^{(k)}\mathcal{P}(d^{(k)})$
8      Compute $\beta^{(k)}$: $\beta^{(k)} = \frac{\langle \mathrm{grad}\,\mathcal{H}(F^{(k+1)}), \mathrm{grad}\,\mathcal{H}(F^{(k+1)}) - \mathrm{grad}\,\mathcal{H}(F^{(k)})\rangle}{\langle \mathrm{grad}\,\mathcal{H}(F^{(k)}), \mathrm{grad}\,\mathcal{H}(F^{(k)})\rangle}$
9      $k \leftarrow k + 1$
10   **end**
11   **return** $F^{(k)}$

---

**Algorithm 5:** Riemannian Trust Region Algorithm on RIM Manifold

---

**Input:** RIM manifold $\mathcal{M} = \{X \mid X1_c = 1_n, l < X^T 1_n < u, X > 0\}$
Objective function $\mathcal{H}(F)$, Retraction $R_X(tV)$, Initial point $F_0 \in \mathcal{M}$, Initial trust region radius $\Delta_0$.
**Output:** Sequence of iterates $\{F^{(k)}\}$ converging to a stationary point of $\mathcal{H}$

1   Initialize $k = 0$ Initialize $\Delta_0$ **while** *not converged* **do**
2      Compute Riemannian gradient $\mathrm{grad}\,\mathcal{H}(F^{(k)})$
3      Compute the Riemannian Hessian $\mathrm{hess}\,\mathcal{H}(F^{(k)})$
4      Solve the trust region subproblem: $\Delta^{(k)} = \arg\min_{\|d\| \le \Delta_k} \mathcal{H}(F^{(k)} + d)$
5      Compute the step size $\kappa^{(k)}$ using a line search or heuristic method
6      Perform Retraction: $F^{(k+1)} = R_{F^{(k)}}(\kappa^{(k)}d^{(k)})$
7      Update the trust region radius $\Delta_{k+1}$;
8      $k \leftarrow k + 1$
9   **end**
10   **return** $F^{(k)}$

---

# D    DETAILS OF THE EXPERIMENTAL SETUP

## D.1    EXPERIMENT 2 SETUP

In the first problem of Experiment 2, due to the particularity of the manifold, it is known that the optimal solution on manifold $\mathcal{M}$ is $A$, at which point the value of the objective function is 0. Therefore, by comparing the losses of different algorithms under various parameters, the one with the smallest loss is the optimal result.

## D.2    EXPERIMENT 3 SETUP

For Experiment 3, we compared the cases when $l = u$ and $l \neq u$. For $l \neq u$, we set $l = 0.9 \lfloor \frac{n}{c} \rfloor$ and $u = 1.1 \lfloor \frac{n}{c} \rfloor$. When $l = u$, the RIM manifold degenerates into the double stochastic manifold, and we can compare it with algorithms on the double stochastic manifold. When $l = 0.9 \lfloor \frac{n}{c} \rfloor$ and $u = 1.1 \lfloor \frac{n}{c} \rfloor$, more general methods such as the Frank-Wolfe Algorithm (FWA) and Projected Gradient Descent (PGD) are used for comparison. For the case where the RIM manifold degenerates into the double stochastic manifold, we also compared the Riemannian Gradient Descent (DSRGD) and Riemannian Conjugate Gradient (DSRCG) on the double stochastic manifold. A brief introduction to these algorithms is provided as follows:

- The Frank-Wolfe algorithm (Xie et al., 2025) is a well-known method for solving nonlinear constrained optimization problems. The core idea is to find the direction within the constraint set that is closest to the negative gradient direction, and search and descend along this direction to optimize the objective function.

- The Projected Gradient Descent algorithm (Chen et al., 2021) is also a method for solving nonlinear constrained problems. The process involves searching along the gradient direction, and when leaving the constraint set, the point is projected back onto the constraint set.

- Riemannian optimization on the double stochastic manifold (Douik & Hassibi, 2019): This includes Double Stochastic Riemannian Gradient Descent, Double Stochastic Riemannian Conjugate Gradient methods. The algorithm process is similar to the RIM manifold methods, except that the Retraction and Riemannian gradient computation methods are different.

The PGD method differs greatly from Riemannian optimization methods, including the search direction. Projected Gradient Descent follows the Euclidean gradient, but the Euclidean gradient may contain irrelevant information on the constraint set. Riemannian optimization removes the redundant information and searches along the Riemannian gradient direction.

The Retraction process also differs; the projection process in Projected Gradient Descent may not be easy to compute and the result may not be unique, whereas Riemannian optimization can choose an appropriate Retraction process, which is faster and more convenient.

The generality is also different: Riemannian optimization not only has Riemannian descent but can also be naturally extended to methods like Riemannian Conjugate Gradient, Riemannian Coordinate Descent, etc., while Projected Gradient Descent has fewer such extensions.

The convergence properties differ as well; for example, Projected Gradient Descent typically requires convexity to converge to the global optimum, while Riemannian optimization only requires geodesic convexity, and there are cases where non-convex problems are geodesically convex.

We compared the final results obtained by optimizing with these algorithms and the total time required, and we organized the data into tables in the main text and appendix, along with visualizations through plotting.

## D.3    EXPERIMENT 4 SETUP

For RIMRcut, we apply the same initialization as (Xie et al., 2025) and perform RIM optimization on Rcut based on the initialization. When applying the RIM manifold to the Rcut, we compare it with ten benchmark clustering algorithms across eight real-world datasets. These algorithms include KM-based methods, bipartite graph clustering techniques, and various balanced clustering

approaches. By solving the Ratio Cut problem on the RIM manifold, the clustering results are more balanced, as the number of samples within each cluster is constrained to a reasonable range. A detailed introduction to each algorithm is provided below.

- KM partitions data into predefined clusters by minimizing the sum of squared distances between data points and their corresponding cluster centers. It is simple but sensitive to initial centroids and struggles with non-spherical clusters.
- CDKM (Nie et al., 2021) improves KM by utilizing coordinate descent method to directly solve the discrete indicator matrix instead of alternative optimization. It could optimize the solution of KM further.
- Rcut minimizes the cut between two sets in a graph while considering the size of the sets, aiming to balance the partition.
- Ncut improves on Ratio-Cut by normalizing the cut, balancing the partition while considering the total graph weight. It's better suited for non-convex and unevenly distributed clusters.
- Nystrom (Chen et al., 2011) method approximates large kernel matrices using a subset of data, making spectral clustering scalable and efficient for large datasets.
- BKNC (Chen et al., 2022a) (Balanced K-Means with a Novel Constraint) extends K-Means by introducing a balance-aware regularizer, allowing flexible control over cluster balance. It is solved using an iterative optimization algorithm and achieves better balance and clustering performance than existing balanced K-Means variants.
- FCFC (Liu et al., 2018) is an efficient clustering algorithm that combines K-means with a balance penalty, ensuring flexible cluster sizes. It scales well to large datasets and outperforms existing methods in efficiency and clustering quality.
- FSC (Zhu et al., 2017) improves spectral clustering efficiency by using Balanced K-means based Hierarchical K-means (BKHK) to construct an anchor-based similarity graph. It achieves high performance on large-scale data.
- LSCR (Chen & Cai, 2011) randomly selects landmarks instead of using K-Means, making it faster but potentially less accurate than LSCK in capturing data structure.
- LSCK selects representative landmarks via K-Means to construct a smaller graph, reducing computational cost while preserving clustering quality.

To evaluate the clustering performance comprehensively, three metrics are applied, which are clustering accuracy (ACC), normalized mutual information (NMI) and adjusted rand index (ARI). The calculation of these three metrics are displayed below.

### D.3.1 CLUSTERING ACCURACY (ACC)

Clustering Accuracy (Yuan et al., 2024a;b) measures the proportion of correctly clustered data points by aligning predicted cluster labels with ground truth labels. Since clustering algorithms do not inherently assign specific labels, a permutation mapping is applied, often using the Hungarian algorithm, to maximize alignment. The formula for ACC is:

$$\text{ACC} = \frac{\delta(map(\hat{y}_i), y_i)}{n} \tag{204}$$

where $\delta(a, b)$ is an indicator function defined as:

$$\delta(a, b) = \begin{cases} 1, & \text{if } a = b \\ 0, & \text{otherwise,} \end{cases} \tag{205}$$

Here, $\hat{y}_i$ is the predicted label, $y_i$ is the true label, $n$ is the total number of data points, and $map(\hat{y}_i)$ is the permutation mapping function that aligns predicted labels with ground truth labels. ACC ranges from 0 to 1, with higher values indicating better clustering performance.

### D.3.2 NORMALIZED MUTUAL INFORMATION (NMI)

Normalized Mutual Information (Zhong et al., 2021) quantifies the mutual dependence between clustering results and ground truth labels, normalized to account for differences in label distributions.

It evaluates the overlap between clusters and true classes using information theory. Given predicted partitions $\{\hat{C}_i\}_{i=1}^{c}$ and ground truth partitions $\{C_i\}_{i=1}^{c}$, NMI is calculated as:

$$\text{NMI} = \frac{\sum_{i=1}^{c}\sum_{j=1}^{c}\left|\hat{C}_i \cap C_j\right|\log\frac{n\left|\hat{C}_i \cap C_j\right|}{|\hat{C}_i||C_j|}}{\sqrt{\left(\sum_{i=1}^{c}\left|\hat{C}_i\right|\log\frac{|\hat{C}_i|}{n}\right)\left(\sum_{j=1}^{c}|C_j|\log\frac{C_j}{n}\right)}} \tag{206}$$

Here, $|\cdot|$ denotes the size of a set, and $\hat{C}_i \cap C_j$ represents the number of data points belonging to both the $i$-th predicted cluster and the $j$-th ground truth class. NMI ranges from 0 to 1, where 1 indicates perfect agreement between clustering results and ground truth. It is particularly effective in scenarios with imbalanced class distributions.

### D.3.3 ADJUSTED RAND INDEX (ARI)

The Adjusted Rand Index (Dang et al., 2021) measures the similarity between predicted clustering and ground truth by comparing all pairs of samples and evaluating whether they are assigned to the same cluster in both results. A contingency table $H$ is first constructed, where each element $h_{ij}$ represents the number of samples in both predicted cluster $\hat{C}_i$ and ground truth cluster $C_j$. The formula for ARI is:

$$\text{ARI}(\bar{C}, C) = \frac{\sum_{ij}\binom{n_{ij}}{2} - \left[\sum_i\binom{n^i}{2}\sum_j\binom{n_j}{2}\right]/\binom{n}{2}}{\frac{1}{2}\left[\sum_i\binom{n^i}{2} + \sum_j\binom{n_j}{2}\right] - \left[\sum_i\binom{n^i}{2}\sum_j\binom{n_j}{2}\right]/\binom{n}{2}} \tag{207}$$

where $\binom{n_{ij}}{2} = \frac{n_{ij}(n_{ij}-1)}{2}$. ARI ranges from -1 to 1, where 1 indicates perfect clustering, 0 represents random assignments, and negative values indicate worse-than-random clustering. ARI is robust to differences in cluster sizes and does not favor a large number of clusters.

### D.3.4 INTRODUCTION OF REAL DATASETS

The real-world datasets includes: COIL20, Digit, JAFFE, MSRA25, PalmData25, USPS20, Waveform21 and MnistData05. These datasets are selected for their diversity in data types (images, waveforms, and biometric data) and their widespread use in benchmarking machine learning and computer vision algorithms. They provide a comprehensive evaluation framework for testing the robustness and generalization capabilities of the proposed methods. The detailed description of them are displayed below.

- The COIL20 dataset [1] contains 1,440 images of 20 distinct objects, with each object captured from different angles. Each image has 1,024 dimensions, making it suitable for object recognition and clustering tasks.

- The Digit dataset consists of 1,797 instances of handwritten digits, ranging from 0 to 9. Each sample has 64 dimensions, representing low-resolution grayscale images.

- The JAFFE dataset includes 213 facial expression images from 10 subjects, covering seven basic emotions. Each image has 1,024 dimensions, making it suitable for facial expression recognition and emotion analysis.

- The MSRA25 dataset is a widely used benchmark for face recognition task. It consists of 1,799 grayscale face images, each resized to $16\times16$ pixels. The dataset includes 12 clusters, representing different individuals or categories.

- The PalmData25 [2] dataset consists of 2,000 palmprint images, each with 256 dimensions. It includes 100 clusters.

- The USPS20 dataset is a subset of the USPS handwritten digit dataset, containing 1,854 instances. Each sample has 256 dimensions, representing grayscale images of digits.

---

[1] http://www.cad.zju.edu.cn/home/dengcai/Data/data.html
[2] https://www.scholat.com/xjchensz

- The Waveform21 dataset [3] contains 2,746 instances of synthetic waveform data, each with 21 dimensions. It includes 3 clusters.
- The MnistData05 dataset is a subset of the MNIST dataset, containing 3,495 instances of handwritten digits. Each sample has 784 dimensions, representing 28×28 grayscale images. It is widely used for digit recognition, classification, and clustering tasks, providing a benchmark for evaluating machine learning models.

### D.3.5   How to Choose $l$ and $u$

$l$ and $u$ are pivotal parameters within the RIM manifold. When the values of $l$ and $u$ are set to be equal, an approximation of the doubly stochastic manifold can be achieved. When $l$ and $u$ are not equal, their application to practical problems holds significant meaning, particularly in the context of unbalanced scenarios. For instance, in clustering tasks, the RIM manifold encompasses all indicator matrices, with $l$ and $u$ representing the minimum and maximum number of samples within each cluster, respectively. The magnitude of these parameters can be estimated based on the total number of samples and the known number of clusters. Alternatively, they may be assigned according to certain prior knowledge. However, it is noteworthy that in the absence of prior information, the values of $l$ and $u$ can be set within a broader range. In addition, a suitable choice of $l$ and $u$ can also be determined through multiple trials.

The parameter in RIM optimization is listed in Table 8.

Table 8: Values of $l$ and $u$ on different data sets for RIMRcut

| Datasets | $l$ | $u$ |
|---|---|---|
| COIL20 | [0.6*n/c] | [1.2*n/c] |
| Digit | [0.4*n/c] | [1.6*n/c] |
| JAFFE | [0.4*n/c] | [1.6*n/c] |
| MSRA25 | [0.4*n/c] | [1.6*n/c] |
| PalmData25 | [0.4*n/c] | [1.8*n/c] |
| USPS20 | [0.6*n/c] | [2.0*n/c] |
| Waveform21 | [0.4*n/c] | [1.8*n/c] |
| MnistData05 | [0.8*n/c] | [1.4*n/c] |

Subsequently, we will perform clustering using the data in this table and visualize the clustering results, as shown in Figure 7 and Figure 8.

Moreover, we acknowledge that precisely choosing $l$ and $u$ is a challenging task, as it is essentially equivalent to obtaining prior information about the dataset. Our study is conducted under the assumption that such prior information is available. Nevertheless, we also provide a possible way to estimate this prior knowledge, namely by running **K-Means**[*] to approximate the cluster proportions. For instance, on the MnistData05 dataset, the estimation yields $(l, u) = (0.86 \times \frac{n}{c}, 1.22 \times \frac{n}{c})$, which is close to the values we selected. Among them, it is preferable to select a balanced K-Means variant (Chang et al., 2014) or a K-Means method equipped with balance regularization.

| | COIL20 | Digit | JAFFE | MSRA25 | PalmData25 | USPS20 | Waveform21 | MnistData05 |
|---|---|---|---|---|---|---|---|---|
| $(l, u)$ **K-Means**[*] | (0.41×, 1.47×) | (0.52×, 1.66×) | (0.24×, 1.59×) | (0.47×, 1.92×) | (0.33×, 1.85×) | (0.53×, 1.71×) | (0.72×, 1.36×) | (0.86×, 1.22×) |
| $(l, u)$ Selected | (0.6×, 1.2×) | (0.4×, 1.6×) | (0.4×, 1.6×) | (0.4×, 1.6×) | (0.4×, 1.8×) | (0.6×, 2.0×) | (0.4×, 1.8×) | (0.8×, 1.4×) |

Table 9: $(l, u)$ values from datasets.

At the same time, although the algorithm is sensitive to $(l, u)$, the sensitivity is not high. Taking the MnistData05 dataset as an example, the performance metrics under different values of $l$ and $u$ are as follows. Here, $a\times$ denotes $a \times \frac{n}{c}$.

Table 10: Performance under different $(l, u)$ values on the *MnistData05* dataset.

| $(l, u)$ | (0×, 2×) | (0.3×, 1.7×) | (0.5×, 1.5×) | (0.6×, 1.4×) | (0.7×, 1.3×) | (0.8×, 1.4×) | (0.9×, 1.1×) |
|---|---|---|---|---|---|---|---|
| ACC | 61.23 | 61.61 | 62.86 | 63.12 | 64.26 | 65.55 | 66.09 |
| NMI | 54.96 | 55.53 | 56.68 | 57.54 | 58.68 | 59.35 | 61.93 |
| ARI | 46.02 | 46.25 | 49.37 | 50.73 | 51.82 | 52.87 | 53.02 |

In addition, we also provide results showing how the accuracy varies with $(l, u)$ across other datasets in Table 11.

---

[3]http://archive.ics.uci.edu/datasets

Table 11: ACC metric under different $(l, u)$ values on the other dataset.

| $(l, u)$ | $(0\times, 2\times)$ | $(0.3\times, 1.7\times)$ | $(0.5\times, 1.5\times)$ | $(0.6\times, 1.4\times)$ | $(0.7\times, 1.3\times)$ | $(0.8\times, 1.4\times)$ | $(0.9\times, 1.1\times)$ |
|---|---|---|---|---|---|---|---|
| Waveform21 | 69.16 | 68.90 | 69.99 | 72.17 | 71.81 | 74.80 | 71.08 |
| USPS20 | 68.99 | 70.39 | 67.53 | 68.28 | 65.16 | 65.70 | 68.23 |
| PalmData25 | 87.10 | 86.15 | 87.70 | 86.60 | 91.05 | 89.45 | 91.10 |
| MSRA25 | 54.92 | 55.09 | 55.86 | 56.97 | 57.42 | 55.59 | 61.09 |
| JAFFE | 86.38 | 92.95 | 96.71 | 96.71 | 96.71 | 96.71 | 92.95 |
| Digit | 78.46 | 81.52 | 84.14 | 82.47 | 79.74 | 80.80 | 77.90 |
| COIL20 | 72.92 | 76.94 | 77.99 | 78.13 | 78.68 | 77.22 | 70.49 |

# E    ADDITIONAL EXPERIMENTAL RESULTS

In this section, we present additional experimental results to better demonstrate the advantages of the proposed algorithm.

## E.1    RESULTS OF EXPERIMENTAL 1

Experiment 1 compares the running times of three Retraction methods under different matrix sizes. In Table 12, we compare the running times when $l$ is not equal to $u$. The results that run the fastest under each set of experiments are highlighted in **red**. Additionally, for better visualization, we present a three-dimensional bar chart showing the performance of multiple Retraction methods, as illustrated in Figure 3. The experimental results reveal that when the matrix dimension is small, Sinkhorn outperforms the other two methods in terms of speed, while Dykstras shows an advantage when the matrix dimension is larger. This conclusion holds true both when $l$ equals $u$ and when $l$ does not equal $u$. While the efficiency of dual method is always inferior than other methods.

Table 12: Table of Execution Time when $l \neq u$ for Different Retraction Algorithms($s$)

| Row&Col | Dual | | | | | | Sinkhorn | | | | | | Dykstras | | | | | |
|---|---|---|---|---|---|---|---|---|---|---|---|---|---|---|---|---|---|---|
| | 500 | 1000 | 3000 | 5000 | 7000 | 10000 | 500 | 1000 | 3000 | 5000 | 7000 | 10000 | 500 | 1000 | 3000 | 5000 | 7000 | 10000 |
| 5 | 0.015 | 0.025 | 0.056 | 0.083 | 0.109 | 0.140 | **0.001** | **0.004** | 0.017 | 0.042 | 0.085 | 0.166 | 0.011 | 0.005 | **0.011** | **0.018** | **0.027** | **0.037** |
| 10 | 0.020 | 0.039 | 0.082 | 0.111 | 0.145 | 0.183 | **0.001** | **0.003** | 0.017 | 0.042 | 0.081 | 0.179 | 0.009 | 0.005 | **0.015** | **0.022** | **0.031** | **0.044** |
| 50 | 0.053 | 0.106 | 0.763 | 1.353 | 1.934 | 2.738 | **0.001** | **0.005** | **0.021** | 0.056 | 0.109 | 0.226 | 0.006 | 0.010 | 0.022 | **0.038** | **0.052** | **0.072** |
| 100 | 0.014 | 0.156 | 1.556 | 2.747 | 3.948 | 5.675 | **0.002** | **0.005** | **0.029** | 0.079 | 0.149 | 0.288 | 0.009 | 0.012 | 0.030 | **0.054** | **0.071** | **0.100** |
| 500 | 0.060 | 0.119 | 7.296 | 12.208 | 17.021 | 23.773 | **0.006** | **0.014** | 0.114 | 0.305 | 0.577 | 1.119 | 0.018 | 0.032 | **0.089** | **0.157** | **0.207** | **0.299** |
| 1000 | 0.103 | 0.172 | 15.483 | 25.830 | 37.027 | 58.107 | **0.018** | **0.036** | **0.194** | 0.500 | 0.889 | 1.781 | 0.036 | 0.071 | 0.204 | **0.367** | **0.522** | **0.763** |

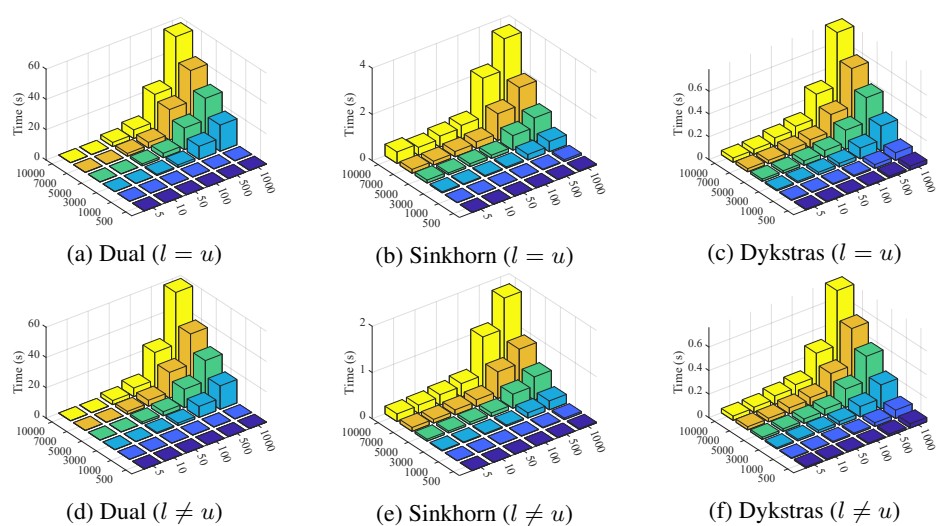

|          |          |          |
|----------|----------|----------|
| (a) Dual ($l = u$) | (b) Sinkhorn ($l = u$) | (c) Dykstras ($l = u$) |
| (d) Dual ($l \neq u$) | (e) Sinkhorn ($l \neq u$) | (f) Dykstras ($l \neq u$) |

Figure 3: Comparison of running time for different Retraction algorithms.

## E.2    RESULTS OF EXPERIMENTAL 2

For the first question in Experiment 2, we compare the application of gradient descent, conjugate gradient, and trust-region methods on the RIM manifold. The value of cost function and running time of gradient descent and conjugate gradient on RIM manifold are display in Table 13 and 14. As can be seen from the two tables, regardless of the optimization method employed, the loss function values and running time of the RIM manifold approach are superior to those of the doubly stochastic manifold method. This advantage is attributed to the lower computational complexity of gradient

Table 13: Cost and Time on the RIM Manifold and Doubly Stochastic Manifold(RGD).

| Row&Col | RIM Manifold | | | | | | Doubly Stochastic Manifold | | | | | |
| | Cost | | | Time | | | Cost | | | Time | | |
| Size | 5000 | 7000 | 10000 | 5000 | 7000 | 10000 | 5000 | 7000 | 10000 | 5000 | 7000 | 10000 |
| 5 | 4.74E-14 | 1.14E-13 | 1.05E-13 | 1.233 | 0.974 | 1.225 | 4.96E-07 | 6.08E-07 | 9.01E-07 | 17.19 | 18.07 | 38.73 |
| 10 | 1.28E-13 | 4.48E-05 | 7.04E-15 | 0.864 | 2.686 | 1.311 | 1.22E-06 | 7.73E-07 | 2.39E-06 | 12.76 | 19.20 | 22.45 |
| 20 | 5.39E-14 | 1.09E-14 | 1.89E-13 | 0.779 | 1.266 | 1.914 | 3.07E-06 | 2.79E-06 | 5.46E-06 | 18.34 | 20.08 | 27.02 |
| 50 | 1.95E-13 | 8.12E-14 | 1.84E-13 | 1.442 | 2.780 | 2.663 | 3.71E-06 | 6.38E-06 | 9.27E-06 | 48.72 | 37.79 | 75.39 |
| 70 | 1.72E-13 | 4.47E-13 | 1.73E-13 | 2.350 | 2.811 | 4.356 | 7.91E-06 | 9.68E-06 | 1.82E-05 | 39.37 | 64.68 | 56.13 |
| 100 | 1.58E-15 | 1.12E-14 | 2.32E-13 | 3.086 | 3.242 | 4.126 | 1.37E-05 | 1.89E-05 | 2.99E-05 | 46.06 | 93.26 | 105.8 |

and Hessian matrix calculations on the RIM manifold. For example, when the matrix size is 100 by 10,000, for the RTR method, the running time is increased by approximately **200** times. For the RGD method, the time required is only one-twenty five of that for the doubly stochastic manifold. As for RCG method, the running time is increased by approximately **75** times. Meanwhile, optimization methods on RIM manifolds often yield solutions closer to zero (the ratio of losses can even reach **1E10**) compared to methods on doubly stochastic manifolds.

The second issue pertains to the problem of image restoration. We introduced varying levels of noise into two images and then compared the visual outcomes of the RIM manifold-based method with those of the DSM-based method in restoring the original images from their noisy counterparts. The visual results are displayed in Figure 4, which also annotates the values of the parameter $\xi$. Regardless of the intensity of the noise, the images restored by the RIM method are clearer and retain better texture information.

Table 14: Cost and Time on the RIM Manifold and Doubly Stochastic Manifold(RCG).

| Row&Col | RIM Manifold | | | | | | Doubly Stochastic Manifold | | | | | |
| | Cost | | | Time | | | Cost | | | Time | | |
| Size | 5000 | 7000 | 10000 | 5000 | 7000 | 10000 | 5000 | 7000 | 10000 | 5000 | 7000 | 10000 |
| 5 | 3.74E-14 | 4.63E-13 | 1.20E-13 | 0.285 | 0.375 | 0.683 | 8.16E-10 | 7.78E-10 | 2.57E-09 | 4.624 | 10.22 | 15.42 |
| 10 | 6.22E-14 | 2.56E-13 | 4.92E-14 | 0.161 | 0.307 | 0.579 | 1.81E-09 | 2.71E-09 | 1.53E-09 | 6.230 | 11.29 | 13.96 |
| 20 | 1.03E-13 | 1.08E-13 | 1.52E-15 | 0.396 | 0.817 | 0.558 | 4.75E-09 | 3.53E-09 | 2.67E-09 | 11.91 | 16.91 | 17.13 |
| 50 | 5.69E-14 | 1.56E-13 | 2.51E-13 | 0.859 | 1.047 | 1.774 | 3.74E-09 | 5.49E-09 | 4.81E-09 | 30.11 | 45.33 | 58.19 |
| 70 | 2.22E-13 | 1.74E-13 | 6.37E-17 | 0.932 | 1.603 | 1.024 | 4.36E-09 | 2.52E-09 | 4.81E-09 | 46.83 | 73.80 | 60.49 |
| 100 | 4.21E-13 | 1.89E-15 | 1.61E-14 | 1.542 | 0.960 | 2.045 | 4.03E-09 | 5.01E-09 | 7.99E-09 | 55.70 | 81.65 | 158.7 |

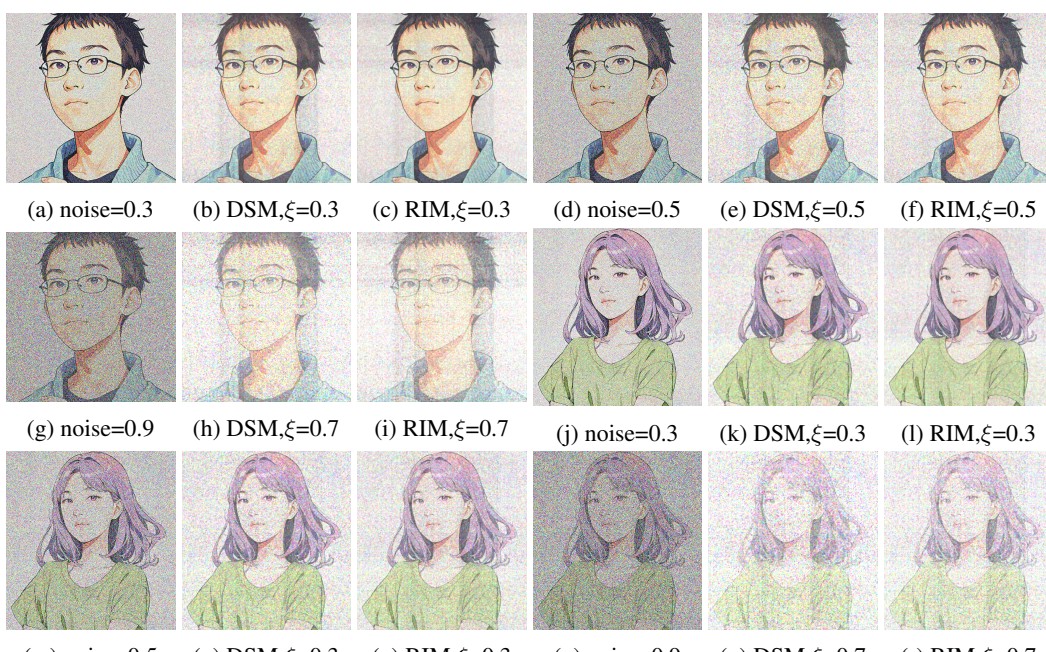

(a) noise=0.3  (b) DSM,$\xi$=0.3  (c) RIM,$\xi$=0.3  (d) noise=0.5  (e) DSM,$\xi$=0.5  (f) RIM,$\xi$=0.5

(g) noise=0.9  (h) DSM,$\xi$=0.7  (i) RIM,$\xi$=0.7  (j) noise=0.3  (k) DSM,$\xi$=0.3  (l) RIM,$\xi$=0.3

(m) noise=0.5  (n) DSM,$\xi$=0.3  (o) RIM,$\xi$=0.3  (p) noise=0.9  (q) DSM,$\xi$=0.7  (r) RIM,$\xi$=0.7

Figure 4: mage Denoising Results.

Table 15: Time and Loss of Different Optimization Algorithms on Ratio Cut when $l \neq u$

| Datasets&Methods | FWA | | PGD | | RIMRGD | | RIMRCG | | RIMRTR | |
|---|---|---|---|---|---|---|---|---|---|---|
| | Time | Cost | Time | Cost | Time | Cost | Time | Cost | Time | Cost |
| COIL20 | 6.220 | 3.908 | 6.061 | 0.7108 | 8.040 | 0.5306 | **2.601** | **0.494** | 15.97 | 0.588 |
| Digit | 5.878 | 0.389 | 6.063 | 0.817 | 7.355 | **0.652** | **1.443** | 0.755 | 13.92 | 0.661 |
| JAFFE | 0.257 | 0.207 | 1.019 | 0.294 | **0.116** | 1.110 | 0.260 | 0.154 | 3.741 | 0.103 |
| MSRA25 | 6.238 | 0.253 | 6.444 | 0.048 | 9.123 | 0.037 | 9.787 | **0.000** | 15.95 | 0.033 |
| PalmData25 | 77.69 | 16.40 | 71.73 | 3.299 | 25.54 | **0.984** | **5.635** | 6.686 | 19.05 | 12.78 |
| USPS20 | 6.133 | 1.631 | 6.109 | 1.563 | 7.025 | **1.544** | **1.309** | 1.729 | 17.72 | 1.551 |
| Waveform21 | 12.62 | 0.405 | 8.529 | 0.452 | 9.650 | **0.366** | **1.571** | 0.373 | 46.20 | **0.366** |
| MnistData05 | 19.86 | **0.538** | 17.09 | 12.36 | 16.52 | 1.693 | **1.876** | 2.467 | 30.47 | 1.677 |

### E.3 RESULTS OF EXPERIMENTAL 3

Experiment 3 compared the objective function values and running times of the rim manifold-based approach with other solution methods on real datasets when the objective function was the Ratio Cut. The results for the case where $l$ equals $u$ are shown in Table 5, while the results for $l$ not equal to $u$ are reported in Table 15. It can be observed that the RIMRCG method achieves the lowest running time on most datasets. Meanwhile, the RIMRGD method can reach the minimum in terms of loss. Furthermore, for each dataset, we have plotted the iteration curves of the objective function values against the number of iterations for various optimization methods. These are displayed in Figures 5 and 6, respectively. From the convergence curves in Figure 5, it is evident that the Rim manifold-based methods enable the objective to decrease more rapidly within a shorter number of iterations. In contrast, the descent curves of the PGD and DSRGD methods are more gradual. A similar experimental outcome is also presented in Figure 6.

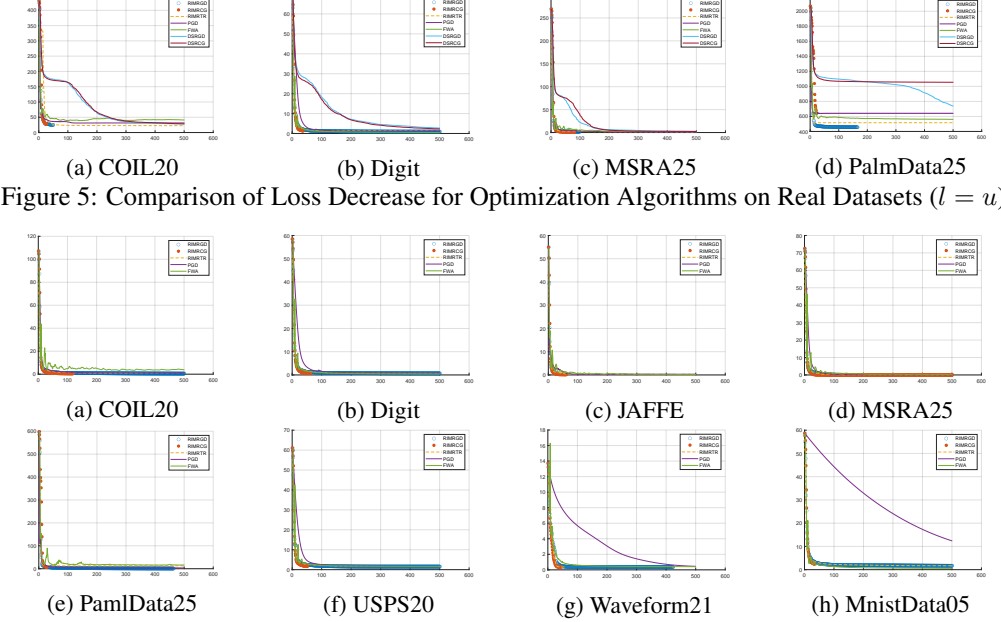

Figure 5: Comparison of Loss Decrease for Optimization Algorithms on Real Datasets ($l = u$).

(a) COIL20    (b) Digit    (c) MSRA25    (d) PalmData25

(a) COIL20    (b) Digit    (c) JAFFE    (d) MSRA25

(e) PamlData25    (f) USPS20    (g) Waveform21    (h) MnistData05

Figure 6: Comparison of Loss Decrease for Optimization Algorithms on Real Datasets ($l \neq u$).

### E.4 RESULTS OF EXPERIMENTAL 4

In this section, we mainly provide two supplementary materials. First, we verify whether Riemannian optimization on the RIM manifold ensures that the distribution of each column lies within the prescribed range. Second, we visualize the learned indicator matrix to examine whether each entry $F_{ij} \in [0, 1]$ is satisfied.

Figure 7 illustrates the column sum distributions of the relaxed indicator matrix obtained via Riemannian gradient descent (RIMRGD) on the RIM manifold for different datasets. The dashed lines represent the values of the upper bound $u$ and lower bound $l$. As shown, all column sums eventually lie within the specified interval $[l, u]$.

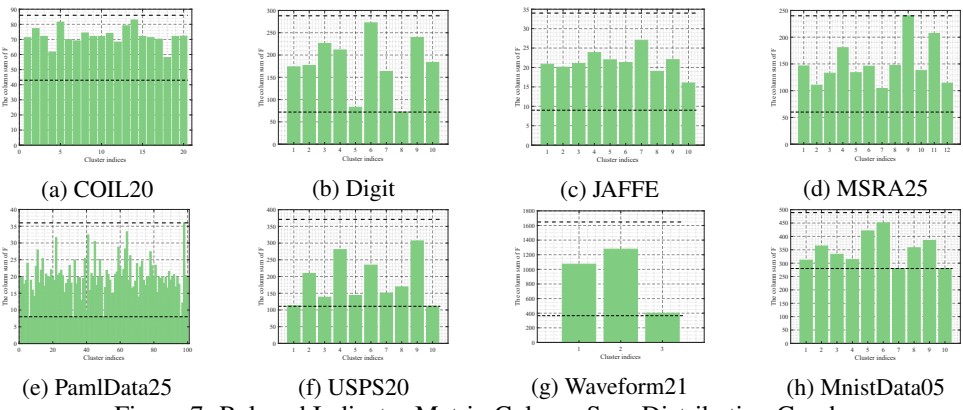

Figure 7: Relaxed Indicator Matrix Column Sum Distribution Graph.

It is worth noting that, under the specified bounds $[l, u]$, not all bounds are necessarily active for every dataset. For instance, in the Digit dataset, the lower bound $l$ is active, as the sum of the 8th column reaches the lower bound, while no column reaches the upper bound $u$. In contrast, for the MSRA25 dataset, the upper bound $u$ is active, but the lower bound $l$ is not. For some datasets like COIL20, neither the lower nor the upper bounds are active, possibly because the dataset naturally leads to a balanced partitioning.

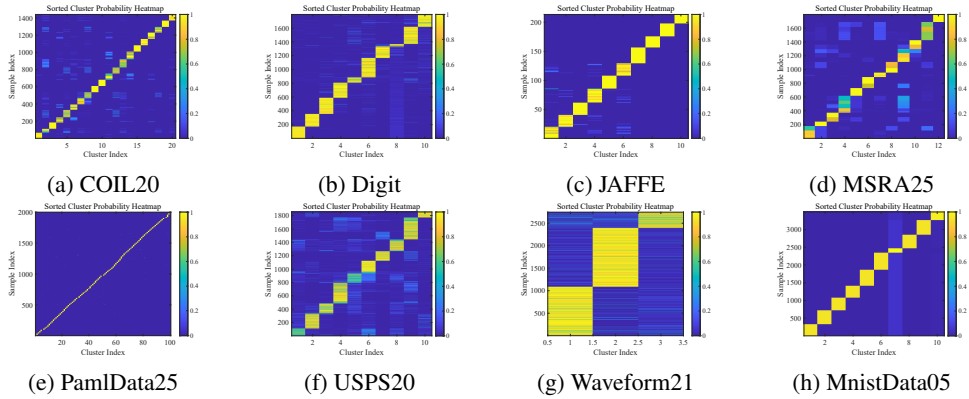

Figure 8: Visualization of the Relaxed Indicator Matrix.

Figure 8 presents the visualization results of the relaxed indicator matrix. It can be observed that each element $F_{ij}$ lies within the interval $[0, 1]$, and the indicator matrix exhibits a clear clustered structure. This structure indicates distinct clustering results, suggesting that learning on the relaxed indicator matrix manifold effectively captures the underlying structure of the graph.

## F  RIM MANIFOLD EQUIPPED WITH FISHER METRIC

In this section, we will explain why we assign the Euclidean inner product to $\mathcal{M} = \{X | X1_c = 1_n, l < X^T 1_n < u, X > 0\}$ instead of the currently more commonly used Fisher information metric. The RIM manifold is defined as $\mathcal{M} = \{X | X1_c = 1_n, l < X^T 1_n < u, X > 0\}$, where the row sums are equal to 1. Therefore, each element's rows on the RIM manifold can be considered as a probability distribution and can equip a Fisher information metric. Specifically, for each point $X$ on the RIM manifold, there exists a tangent space $T_X \mathcal{M}$ which is a linear space. Previously, the Euclidean metric was equipped on this linear space, that is, $\forall U, V \in T_X \mathcal{M}, < U, V >_X = \sum_{i=1}^n \sum_{j=1}^n U_{ij} V_{ij}$. This section will discuss the impact on optimization over the manifold when the Fisher information metric is equipped on $T_X \mathcal{M}$, that is,

$$\forall U, V \in T_X \mathcal{M}, < U, V >_X = \sum_{i=1}^n \sum_{j=1}^n \frac{U_{ij} V_{ij}}{X_{ij}} \tag{208}$$

To distinguish it from the previous RIM manifold, we call the RIM manifold equipped with the Fisher information metric the Fisher RIM manifold, abbreviated as FRIM manifold.

### F.1  DIMENSION AND TANGENT SPACE

Regarding dimension and tangent space, their definitions depend only on the manifold itself and are independent of the metric equipped on it. Therefore, for the same set $\mathcal{M}$, whether it is equipped with the Euclidean metric or the Fisher information metric, it has the same dimension and tangent space. That is, both the RIM manifold and the FRIM manifold have a dimension of $(n-1)c$, and the tangent space is $T_X \mathcal{M} = \{U \mid U1_c = 0\}$. The proof can be found in Theorem A.1

### F.2  RIEMANNIAN GRADIENT, RIEMANNIAN CONNECTION AND RIEMANNIAN HESSIAN

When the Fisher metric is assigned to $\{X \in \mathbb{R}^{n \times c} | X > 0\}$, the gradient of $\mathcal{H}$ at $X$ in $\{X \in \mathbb{R}^{n \times c} | X > 0\}$ is given by $\text{Grad}\mathcal{H} \odot X$, where $\text{Grad}\mathcal{H}$ is the Euclidean gradient. At this time, the FRIM manifold is a Riemannian embedded submanifold of $\{X \in \mathbb{R}^{n \times c} | X > 0\}$. The Riemannian gradient on the FRIM manifold is the orthogonal projection under the Fisher metric. The expression of this orthogonal projection is

$$\text{Proj}_{T_X \mathcal{M}}(Z) = Z - (\alpha 1_c^T) \odot X, \quad \alpha = Z1_c \in \mathbb{R}^n \tag{209}$$

The Riemannian connection on the FRIM manifold is the orthogonal projection of the connection under the Fisher metric, where the connection on $\{X \in \mathbb{R}^{n \times c} | X > 0\}$ can be expressed as

$$\bar{\nabla}_U V = DV[U] - \frac{1}{2}(U \odot V) \oslash X, \quad U, V \in \mathbb{R}^{n \times c} \tag{210}$$

The Riemannian connection on the FRIM manifold is given by

$$\begin{cases} \nabla_U V = \text{Proj}_{T_X \mathcal{M}}(\bar{\nabla}_U V) = \text{Proj}_{T_X \mathcal{M}}\left(DV[U] - \frac{1}{2}(U \odot V) \oslash X\right) \\ U, V \in T_X \mathcal{M}, X \in \{X \in \mathbb{R}^{n \times c} | X > 0\} \end{cases} \tag{211}$$

Furthermore, the Riemannian Hessian mapping is given by

$$\text{hess }\mathcal{H}(X)[V] = \text{Proj}_{T_X \mathcal{M}}\left(D \text{ grad }\mathcal{H}(X)[V] - \frac{1}{2}(V \odot \text{grad }\mathcal{H}(X)) \oslash X\right) \tag{212}$$

It can be seen that the Riemannian gradient, Riemannian connection and Riemannian Hessian on the FRIM manifold are the same as those on the single stochastic manifold equipped with the Fisher information metric. Since the FRIM manifold itself can be regarded as a Riemannian embedded submanifold of the single stochastic manifold, it is not surprising that these three Riemannian tools are the same as those on the single stochastic manifold. However, the existing Retraction mapping on the single stochastic manifold cannot be applied to the FRIM manifold, because it cannot be guaranteed that the curve generated by the Retraction mapping on the single stochastic manifold will always lie on the FRIM manifold (it may violate the column constraint).

### F.3 RETRACTION MAPPING

Although the Retraction mapping on the single stochastic manifold cannot be used as the Retraction mapping on the FRIM manifold, the Retraction mapping on the RIM manifold proposed in this paper can naturally serve as the Retraction mapping on the FRIM manifold. That is, the FRIM manifold naturally has three Retraction methods respectively given by Theorems A.6, A.7 and A.8. However, Theorem A.5 indicates that the result obtained by Theorem A.6 is a geodesic on the RIM manifold. However, on the FRIM manifold, Theorem A.6 is not an orthogonal projection under the Fisher information metric, so the geodesic on the FRIM manifold cannot be obtained. That is to say, although the three methods in Theorems A.6, A.7 and A.8 can all be used as Retractions, none of them is a second-order Retraction.

### F.4 WHICH TO USE?

Although there is also a set of Riemannian tools available on the FRIM manifold, according to the analysis above, the Riemannian toolbox under the RIM manifold and the Riemannian toolbox on the FRIM manifold have almost the same time complexity and can use the same Retraction. However, when using the Dykstras Retraction, a geodesic can be quickly obtained on the RIM manifold, while it is impossible to obtain a geodesic on the FRIM manifold, meaning a second-order Retraction cannot be achieved. This may have a certain impact on the convergence of the algorithm. Therefore, we recommend using the RIM manifold, which restricts the Euclidean inner product to the manifold rather than the Fisher information metric.

# G    EXPLANATION REGARDING DETAILS

In this section, we provide detailed answers to three questions raised by the community. These questions essentially represent the special techniques that enable the RIM manifold to operate at singularities.

- **Question 1:** Why is projection chosen as the Retraction onto $\{X \mid X \geq 0, X1_n = 1_n, l \leq X^T1_n \leq u\}$? What if strict inequalities are required?
- **Question 2:** For the RIM manifold $\{X \mid X > 0, X1_n = 1_n, l < X^T1_n < u\}$, does it become an empty set when $l = u = r$? Why can the algorithm function normally?
- **Question 3:** How does it behave near the boundary of the RIM manifold?

## G.1    ANSWER TO THE QUESTION 1:

Choosing projection as the Retraction is partly due to its speed, but more importantly, due to its stability. When strict inequalities are required, the following analysis can be made:

When $l \neq u$, one can always choose a sufficiently small $\varepsilon$ such that the constraints effectively map. Specifically, given $l$ and $u$, if we must require the resulting $X$ to satisfy $\{X \mid X > 0, X1_n = 1_n, l < X^T1_n < u\}$, we only need to pick an $\varepsilon$ and project onto the set:

$$\{X \mid X \geq 0 + \varepsilon, X1_n = 1_n, l + \varepsilon \leq X^T1_n \leq u - \varepsilon\} \tag{213}$$

Here, $\varepsilon$ can be chosen to be very small, for example, $\varepsilon = 10^{-12}$ or even smaller than machine precision. This is also very easy to implement in the RIM toolbox: simply set $u' = u - \varepsilon$ and $l' = l + \varepsilon$, and then input $(u', l')$ into the RIM toolbox. At the same time, the exact same method can be used to handle $\{X \mid X > 0\}$ and $\{X \mid X > 0 + \varepsilon\}$. In fact, when $\varepsilon$ is sufficiently small, it does not affect the final result at all.

When $l = u$, the advantage of choosing projection as the Retraction becomes even more apparent, as we will point out in Answer 2.

Furthermore, it must be noted that in scenarios where the RIM manifold is typically used, such as machine learning, one usually does not need to care about whether values are taken at the boundary. In scenarios like classification and clustering, one typically only concerns themselves with the maximum or minimum elements of the Relaxed Indicator Matrix. From a numerical optimization perspective, a correction using an $\varepsilon$ can yield significant acceleration, making it a worthwhile trade-off.

## G.2    ANSWER TO THE QUESTION 2:

When $l = u$, since the Retraction we actually adopt projects onto $\{X \mid X \geq 0, X1_n = 1_n, l \leq X^T1_n \leq u\}$, which in this case means projecting onto:

$$\{X \mid X \geq 0, X1_n = 1_n, X^T1_n = r\} \tag{214}$$

This is the reason why the RIM manifold is always able to obtain points on the doubly stochastic manifold when setting $l = u$.

Moreover, considering the Riemannian gradient, this is equivalent to projecting the Euclidean gradient, causing components that deviate from the tangent direction to become zero. This is equivalent to a gradient projection method corrected by a Lagrange multiplier, which accelerates convergence.

## G.3    ANSWER TO THE QUESTION 3:

When $l \neq u$, we can always leverage the equivalence condition mentioned earlier. When considering the properties of the RIM manifold $\{X \mid X > 0, X1_n = 1_n, l < X^T1_n < u\}$ near $l$ and $u$, it is always possible to slightly expand the boundaries, i.e., to consider the interior properties of $\{X \mid X > 0, X1_n = 1_n, l - \varepsilon < X^T1_n < u + \varepsilon\}$. In particular, since the Retraction projects onto the boundary, we can always locate extreme points on the boundary, for instance, where $X^T1_n = u$.

## H  REFERENCE CODE FOR RIM MANIFOLD RIEMANNIAN TOOLBOX

```matlab
function M = RIMfactory(n, c, row,upper,lower)

    maxDSiters = min(1000, n*c);
    if size(row, 1) ~= n
        error('row should be a column vector of size n.');
    end
    if size(upper, 1) ~= c
        error('upper should be a column vector of size c.');
    end
    if size(lower, 1) ~= c
        error('lower should be a column vector of size c.');
    end

    M.name = @() sprintf('%dx%d matrices with positive entries F1_c=1_n,l
        <F1_n<u', n, c);
    M.dim = @() (n-1)*c;
    M.hash = @(X) ['z' hashmd5(X(:))];
    M.lincomb = @matrixlincomb;
    M.zerovec = @(X) zeros(n, c);
    M.transp = @(X1, X2, d) ProjToTangent(d);
    M.vec = @(X, U) U(:);
    M.mat = @(X, u) reshape(u, n, c);
    M.vecmatareisometries = @() true;
    M.inner = @iproduct;

    function ip = iproduct(X,eta, zeta)
        ip = sum((eta(:).*zeta(:)));
    end
    M.norm = @(X,eta) sqrt(M.inner(X,eta, eta));
    M.typicaldist = @() n+c;
    M.rand = @random;
    function X = random(X)
        Z = abs(randn(n, c));
        X = Dykstras(Z, row, lower, upper, maxDSiters);
    end
    M.randvec = @randomvec;
    function eta = randomvec(X)
        Z = randn(n, c);
        eta = ProjToTangent(Z);
    end
    M.proj = @projection;
    function etaproj = projection(X,eta)
        etaproj = ProjToTangent(eta);
    end
    M.tangent = M.proj;
    M.tangent2ambient = @(X,eta) eta;
    M.egrad2rgrad = @egrad2rgrad;
    function rgrad = egrad2rgrad(X,egrad)
        rgrad = ProjToTangent(egrad);
    end
    M.retr = @Retraction;
    function Y = Retraction(X, eta, t)
        if nargin < 3
            t = 1;
        end
        Y=Dykstras(X+t*eta, row, lower, upper, maxDSiters);
    end
    M.ehess2rhess = @ehess2rhess;
    function rhess = ehess2rhess(X, egrad, ehess, eta)
        rhess = ProjToTangent(ehess);
    end

end
```

In this section, we will provide reference code for the RIM manifold toolbox. Our code is compatible with the well-known open-source manifold optimization toolbox Manopt (Boumal et al., 2014), allowing the direct use of Manopt's algorithms to implement Riemannian optimization on the RIM manifold. The first code block creates a factory named "RIM", which allows for the direct call to the RIM factory to obtain the basic description of the RIM manifold, covering the essential information about the manifold and the invocation of basic Riemannian operations.

Dykstras algorithm is one of the methods for implementing Retraction. Its process involves iterative projections and the condition for determining when to exit the loop.

```
function [P] = Dykstras(M, a, b_l, b_u, N)
    if b_l==b_u
        tol=1e-2;
    else
        tol=1e-1;
    end
    rng(1);
    [mn, mc] = size(M);
    P = M;
    z1 = zeros(mn, mc);
    z2 = zeros(mn, mc);
    z3 = zeros(mn, mc);

    for iter = 1:N
        for i = 1:mn
            prev_row = P(i, :) + z1(i, :);
            P(i, :) = EProjSimplex_new(prev_row, a(i));
            z1(i, :) = prev_row - P(i, :);
        end

        for j = 1:mc
            prev_col = P(:, j) + z2(:, j);
            current_sum = sum(prev_col);
            if current_sum >= b_l(j)
                z2(:, j) = 0;
                P(:, j) = prev_col;
            else
                delta = (b_l(j) - current_sum) / mn;
                new_col = prev_col + delta * ones(mn, 1);
                z2(:, j) = prev_col - new_col;
                P(:, j) = new_col;
            end
        end

        for j = 1:mc
            prev_col = P(:, j) + z3(:, j);
            current_sum = sum(prev_col);
            if current_sum <= b_u(j)
                z3(:, j) = 0;
                P(:, j) = prev_col;
            else
                delta = (b_u(j) - current_sum) / mn;
                new_col = prev_col + delta * ones(mn, 1);
                z3(:, j) = prev_col - new_col;
                P(:, j) = new_col;
            end
        end

        if norm(P*ones(mc,1)-a, 'fro') < tol && all(P(:)>=-tol)
            disp(['Converged at iteration: ', num2str(iter)]);
            break;
        end
    end
end
```

In the Dykstras algorithm process, the first step is to project onto the simplex, where the projection function is EProjSimplex_new. The code for this is provided below. During usage, you can create a file named EProjSimplex_new and call the EProjSimplex_new algorithm in each iteration of the Dykstras algorithm process.

```matlab
function [x ft] = EProjSimplex_new(v, k)
    if nargin < 2
        k = 1;
    end;
    ft=1;
    n = length(v);
    v0 = v-mean(v) + k/n;
    vmin = min(v0);
    if vmin < 0
        f = 1;
        lambda_m = 0;
        while abs(f) > 10^-10
            v1 = v0 - lambda_m;
            posidx = v1>0;
            npos = sum(posidx);
            g = -npos;
            f = sum(v1(posidx)) - k;
            lambda_m = lambda_m - f/g;
            ft=ft+1;
            if ft > 100
                x = max(v1,0);
                break;
            end;
        end;
        x = max(v1,0);
    else
        x = v0;
end;
```

The function ProjToTangent is a simple projection function onto the tangent space.

```matlab
function P = ProjToTangent(X)
    c=size(X,2);
    P=X-1/c*X*ones(c,c);
end
```

**When running the code, please create four separate MATLAB files for RIMfactory, Dykstras, EProjSimplex_new, and ProjToTangent, and place them in the manopt folder following this structure:**

```
-manopt;
 --manifolds;
   ---multinomial;
       ----RIMfactory;
       ----Dykstras;
       ----EProjSimplex_new;
       ----ProjToTangent;
```

**Then you can call the functions in the general way as per manopt.**

```matlab
RIM_manifold = RIMfactory(n,c,row,upper,lower);
problem.M = RIM_manifold;
problem.cost = @(X) ...;
problem.egrad = @(X) ...;   % Euclidean gradient
[X_rim,~,info_rim,~] = steepestdescent(problem);
```

Furthermore, we provide reference code for the dual gradient and Sinkhorn algorithms, which allow the Retraction operation to be performed in other ways. Overall, we still recommend using Dykstras algorithm under the Euclidean inner product for descent along the geodesics of the RIM manifold.

```matlab
function F = dual_gradient(Z, l, u, max_iter)
    [n, c] = size(Z);
    l = l(:);
    u = u(:);

    nu = ones(n, 1);
    omega = ones(c, 1);
    rho = ones(c, 1);

    step_size = .05;

    for iter = 1:max_iter
        term = Z - nu * ones(1, c) - ones(n, 1) * omega' + ones(n, 1) *
            rho';
        F_current = max(term, 0);

        grad_nu = F_current * ones(c, 1) - ones(n, 1);
        grad_omega = F_current' * ones(n, 1) - u;
        grad_rho = -F_current' * ones(n, 1)+l;

        nu = nu + step_size * grad_nu;
        omega = omega + step_size * grad_omega;
        rho = rho + step_size * grad_rho;

        omega = max(omega, 0);
        rho = max(rho, 0);
    end
    term = Z - nu * ones(1, c) - ones(n, 1) * omega' + ones(n, 1) * rho';
    F = max(term, 0);
end

function P = sinkR(X, a, l, u, N)
    rng(1)
    [n, c] = size(X);
    K = X;
    u_vec = ones(n, 1);
    q_vec = ones(c, 1);
    v_vec = ones(c, 1);

    for i = 1:N
        u_vec = a ./ (K * (q_vec .* v_vec));

        sum_P_t = sum((u_vec .* K), 1)';
        q_vec = max(l(:) ./ sum_P_t, ones(c, 1));

        sum_P_t = sum((u_vec .* K) .* q_vec', 1)';
        v_vec = min(u(:) ./ sum_P_t, ones(c, 1));

        P = diag(u_vec) * K * diag(q_vec .* v_vec);
        P_liehe = P'*ones(n,1);

        if norm(P*ones(c,1)-ones(n,1), 'fro') < 1e-2 && all(P(:)>=-1e-2)
            && all(P_liehe>=l-1e-2) && all(P_liehe<=u+1e-2)
            break;
        end
    end
end
```

An alternative version of this paper is available on arXiv (Yuan et al., 2025).

