# OpenReview forum: "Riemannian Optimization on Relaxed Indicator Matrix Manifold"
_ICLR.cc/2026/Conference — ICLR 2026 Poster_

### Official Review · Reviewer_m9ij · 2025-10-30

**Soundness:** 2
**Presentation:** 2
**Contribution:** 2
**Rating:** 4
**Confidence:** 4

**Summary:**

The paper introduces the RIM (Relaxed Indicator Matrix) manifold as a generalization of the doubly stochastic and single stochastic manifolds. It proves Riemannian geometric properties of this set and develops a full Riemannian toolbox for optimization, including gradient, Hessian, and multiple retraction methods. The method is applied to large-scale tasks like Ratio Cut clustering and image denoising.

**Strengths:**

1. This paper proposes a manifold that interpolates between existing relaxations of indicator matrices.

2. The authors develops practical Riemannian algorithms (including a retraction via projection).

3. The authors shows speedups over doubly stochastic manifold optimization in experiments.

**Weaknesses:**

1. The paper defines the RIM manifold as a subset of Euclidean space with strict inequalities: M = {X | X1 = 1, l < Xᵀ1 < u, X > 0}
However, this is an open set, and hence optimization problems posed over this set may have no solution, since the infimum may lie on the boundary (e.g., when X^T 1=u). The paper does not discuss this issue, and Theorem 1 claims it's an "embedded submanifold" without clarifying what happens near the boundary. This has major implications — e.g., the projection-based retraction (Theorem 5) may not be well-defined if the minimum lies at the boundary where feasibility breaks.

2. The paper claims that optimization on the Stiefel manifold has time complexity
O(n^3), but this is misleading. For typical tall matrices (n≫c), the complexity is only O(nc^2), as shown in Wen and Yin (2013). Please correct this and cite:

       -Wen and Yin, A feasible method for optimization with orthogonality constraints, Math. Program. (2013)

Also, for retraction definitions, please cite standard sources:

       -Boumal, “An Introduction to Optimization on Smooth Manifolds”, 2023, or

       -Absil et al., “Optimization Algorithms on Matrix Manifolds”, Princeton, 2008

In Theorem 5, the projection used as a retraction may not always be well-defined — the feasibility set is open, and projection may fail to produce a point inside. The paper should acknowledge this and cite related projection-based retractions such as:

       Absil & Malick, Projection-like retractions on matrix manifolds, SIAM J. Optimization, 2012.

3. Every minor step is called a "Theorem", including obvious projections and inner product definitions. This makes the paper harder to read. Please distinguish between core theoretical contributions and auxiliary results.

**Questions:**

1. How do you guarantee the solution remains in the interior of the RIM manifold? Do you clip or project back when the optimizer moves outside?

2. Can the projection in Theorem 5 fail if the retracted point lies on the boundary (e.g., due to positivity or sum constraints)?

---

> ### Author Response · Authors · 2025-11-13
> **Part One**
>
> Hello Reviewer m9ij 😀,
>
> I am astonished by your talent and profound insight! You are undoubtedly one of the most visionary reviewers for ICLR 2026, as you have identified one of the most brilliant aspects of the RIM manifold's design at a glance! I am delighted to explain it to you in detail!
>
> You and another equally visionary reviewer, Y5CU, have pointed out two sides of our intricate design. What's even more fascinating is that your questions can actually answer each other's—this is the most interesting thing I've encountered!
>
> ---
>
> >  However, this is an open set, and hence optimization problems posed over this set may have no solution, since the infimum may lie on the boundary (e.g., when $X^T \mathbf{1}=u$).
>
> You are absolutely correct! We also explained in **Appendix G** that our projection is, in practice, onto the set $\{X | X \ge 0, X\mathbf{1}_n = \mathbf{1}_n, l \le X^T\mathbf{1}_n \le u\}$. As we explained to Reviewer Y5CU, this is done so that our algorithm can still operate effectively at the "singularity" (i.e., when $l=u$).
>
> Specifically:
>
> 1.  When $l \ne u$, this has no real impact. We can consider a tiny quantity $\varepsilon$ such that $l \le X^T\mathbf{1}_n \le u \Leftrightarrow l-\varepsilon < X^T\mathbf{1}_n < u+\varepsilon$. In the machine learning community, we usually don't worry about whether the inequality is strict (I will explain why below). Even if we strictly require $ l < X^\top \mathbf{1}_n < u $, we only need to define the RIM manifold as
> $$
> X \mid X > 0+\varepsilon, X\mathbf{1}_n = \mathbf{1}_n, l+\varepsilon < X^\top\mathbf{1}_n < u-\varepsilon
> $$
> By projecting onto $ l+\varepsilon \le X^\top\mathbf{1}_n \le u-\varepsilon $ as described in our paper, we can ensure that $ l < X^\top\mathbf{1}_n < u $, with $\varepsilon$ chosen arbitrarily small.
>
>
> 2.  However, when $l=u=r$, our retraction is equivalent to projecting onto $\{X | X \ge 0, X\mathbf{1}_n = \mathbf{1}_n, r \le X^T\mathbf{1}_n \le r\}$. **This is the key** to why we can operate at the singularity. In this special case, our algorithm is equivalent to a projection method with a specific gradient correction. This correction ensures that during optimization, no components are generated in meaningless directions (i.e., non-tangential directions), which is one reason why the RIM manifold converges faster!
>
> ---
>
> So, regarding your comment:
> > The paper does not discuss this issue, and Theorem 1 claims it's an "embedded submanifold" without clarifying what happens near the boundary.
>
> In fact, we did provide a simple explanation in Appendix G. When we impose $l \le X^\top \mathbf{1}_n \le u $, we can always equivalently rewrite it as
> $ l - \varepsilon < X^\top \mathbf{1}_n < u + \varepsilon $.
> The latter uses strict inequalities and thus forms a submanifold. Since this equivalence is general and holds universally, it is more convenient and clearer to simply write $ l < X^\top \mathbf{1}_n < u $ in practice.
>
> You also don't need to worry about this:
> >  This has major implications — e.g., the projection-based retraction (Theorem 5) may not be well-defined if the minimum lies at the boundary where feasibility breaks.
>
> On the contrary! It is precisely *because* **we project onto the boundary** that we are able to find an extremum *on* the boundary. If we strictly used inequality signs ($<$ and $>$), we would only be able to find a value that infinitely approaches it.
>
> Another interesting question is why, in the machine learning community, the boundary cases of the RIM manifold are usually not a concern. In fact, I have a lot of experience with optimization problems involving indicator matrices. We typically need these matrices for machine learning tasks like classification/clustering [1] [2]. We are concerned with constraining the *column sums* of the indicator matrix to control the total number in each class, and we care about which entry has the maximum value in each row.
>
> Even from a purely numerical optimization perspective, if we *really* wanted to avoid landing on the boundary, we would just need to add/subtract a tiny $\varepsilon$ correction. This is not complicated and, in return, provides a **massive speedup**!
>
> ---
>
> So, to answer your two questions directly:
>
> > How do you guarantee the solution remains in the interior of the RIM manifold? Do you clip or project back when the optimizer moves outside?
>
> > Can the projection in Theorem 5 fail if the retracted point lies on the boundary (e.g., due to positivity or sum constraints)?
>
> 1.  In our code, we use **Dykstra's algorithm to project onto the boundary or the interior**.
> 2.  **No, it will not fail.** Even in the extreme case where we want to strictly avoid the boundary, we would only need to add a tiny $\varepsilon$.
>
> We hope this answer is satisfactory!
>
> ---
>
> We hope this has answered your questions. Thank you for your support!
>
> [1] Unsupervised Feature Selection via Multi-Structure Learning and Indicator Matrix
>
> [2] Visualization of the cluster indicator matrix

---

> > ### Author Response · Authors · 2025-11-17
> > **An easy experiment**
> >
> > Furthermore, we have added another experiment to alleviate your concern. We consider a simple loss function:
> > $$
> > \min_X \|X-B\|^2, \quad X \in \mathcal{M}
> > $$
> > where $B =[ 1,0; 1,0; 1,0] $ (i.e., the first column is all ones, and the second column is all zeros). $X$ is on the RIM manifold, with $l=0$ and $u=3$.
> >
> > Your concern was:
> > > This has major implications — e.g., the projection-based retraction (Theorem 5) may not be well-defined if the minimum lies at the boundary where feasibility breaks.
> >
> > > Can the projection in Theorem 5 fail if the retracted point lies on the boundary (e.g., due to positivity or sum constraints)?
> >
> > Our experiment yielded the following results:
> >
> > **Run Log:**
> > ```
> >  iter     cost val              grad. norm
> >    0    +1.6070002164248229e+00  1.79276335e+00
> >    1    +3.1423686537994899e-01  7.92763351e-01
> >    2    +0.0000000000000000e+00  0.00000000e+00
> > Gradient norm tolerance reached; options.tolgradnorm = 1e-06.
> > Total time is 0.011777 [s] (excludes statsfun)
> > ```
> >
> > **Converged Result:**
> > ```
> > Optimized X =
> >      1     0
> >      1     0
> >      1     0
> >
> > Final cost = 0.000000
> > Column sums: 3     0
> > Row sums:    1     1     1
> > ```
> >
> > As you can see, the optimal solution $X$ is exactly $B$, which lies on the boundary (column sums are 3 and 0, which are the exact values of $u$ and $l$).
> >
> > It is precisely because Dykstra's algorithm (which our projection is based on) projects **onto** the boundary (rather than strictly *into* the interior) that we are able to find and converge to a point that lies exactly on the boundary. This demonstrates that feasibility is not broken, and the projection remains well-defined.
> >
> > We hope this simple experiment can fully resolve your concern.
> >
> >
> > Our code is following:
> > ```
> > clc; clear; close all;
> >
> > n = 3;
> > c = 2;
> >
> > % Target matrix B
> > B = [1 0;
> >      1 0;
> >      1 0];
> >
> > % RIM constraints
> > row_sum = ones(n,1);   % rows must sum to 1
> > l = zeros(c,1);        % 0 < column sum
> > u = 3*ones(c,1);       % column sum < 3
> >
> > % Build manifold
> > manifold = Relaxd_Indicator_Matrix_factory(n, c, row_sum, u,l);
> >
> >
> > X0 = manifold.rand();
> >
> > % Quadratic objective
> > problem.M = manifold;
> > problem.cost  = @(X) 0.5 * norm(X - B, 'fro')^2;
> > problem.egrad = @(X) (X - B);
> >
> > % Solve
> > [Xopt, costopt, info] = steepestdescent(problem, X0);
> >
> > % Results
> > disp("Optimized X = ");
> > disp(Xopt)
> > fprintf("Final cost = %.6f\n", costopt);
> > disp("Column sums: "); disp(sum(Xopt,1));
> > disp("Row sums: "); disp(sum(Xopt,2));
> > ```

---

> ### Author Response · Authors · 2025-11-13
> **Part Two**
>
> Furthermore, we have checked and updated all citations in the latest version, and we have also changed the original Theorems 1, 2, 3, and 4 to Lemmas. We have completed the revisions to the paper. We look forward to your reply.
>
> Thank you once again for your review. Your kindness and outstanding insight are truly memorable. It would be my honor to discuss this further and learn from you.😀

---

### Official Review · Reviewer_dz1d · 2025-10-31

**Soundness:** 3
**Presentation:** 2
**Contribution:** 2
**Rating:** 4
**Confidence:** 2

**Summary:**

The paper introduces the Relaxed Indicator Matrix Manifold (RIM). The authors prove (M) is an embedded submanifold. They equip (M) with the Euclidean metric restricted to the manifold and derive a simple projection formula for the Riemannian gradient.
For retraction, they give two options: (1) a norm‑minimizing projection that they show is a geodesic for small steps (Dykstra‑style algorithm, Theorem 5, and (2) a Sinkhorn‑style mapping characterized by diagonal scalings.

A central claim is lower per‑step complexity on RIM vs the doubly stochastic manifold (DSM) (Table 1).

Experiments:
(i) retraction timing favors Dykstra as size grows (Table 2).
(ii) “Experiment 2” compares RIM vs DSM on two problems: a convex norm‑approximation and TV image denoising.
(iii) “Experiment 3” applies RIM to Ratio‑Cut with closed‑form Euclidean gradient/Hessian (Appendix A.9) and shows losses/times vs several baselines
(iv) “Experiment 4” reports clustering metrics (ACC/NMI/ARI) across datasets (Table 5).

Authors also provide a convergence proof of RIM‑RGD to stationarity at (O(1/T)) under standard smoothness and Armijo/Wolfe conditions.

**Strengths:**

1. The RIM construction interpolates between single‑stochastic and doubly‑stochastic models via ([l,u]) bounds, letting practitioners encode prior class information.
2. Clean Riemannian operators on RIM reduce to simple column‑mean corrections for the gradient and avoid DSM’s pseudoinverses, yielding the (O(nc)) vs (O(n^3)) complexity gap (Table 1).
3. Multiple retractions are implemented and compared; Dykstra is both fast at large sizes and, by Theorem 5, induces a geodesic in the small‑step regime.
4. Useful coverage of Ratio‑Cut with explicit Euclidean gradient and Hessian (Appendix A.9) that can be projected to RIM.
5. Convergence guarantees for RIM‑RGD with Armijo/Wolfe steps are provided.

**Weaknesses:**

1. No approximation guarantees to the discrete indicator problem. The paper proves geometry and algorithmic convergence, not approximation quality: there is no integrality gap, rounding guarantee, or conditions under which RIM recovers the discrete optimum. The main theory sections and appendices focus on the toolbox, complexity, convergence, not on approximation bounds. A relaxation cannot just be fast, it must actually be a good approximation to the original problem (discrete indicator).
2. TV denoising evidence is weak. The claim relies on an ~10% objective gap (1.05e5 vs 1.17e5) and “zoomed‑in” visual inspection; no PSNR/SSIM/LPIPS or even data‑term MSE are reported near those figures/tables [p.8, text around Table 3; Fig.4].
3. Compute fairness needs clarification. Speedups depend on stopping rules, retraction choice, and any hyperparameter tuning (e.g., (\xi) in TV); these details are not fully standardized across RIM vs DSM in the description of Experiment 2.
4. Choice of ([l,u]) is heuristic and dataset‑dependent. Appendix D.3.5 admits the difficulty, proposes using (n/c) or K‑means proportions, and shows a sensitivity table only for one dataset (MnistData05).
5. Claims that “RIM images are clearer (regardless of noise level)” (page 47) are again qualitative and not well justified.

**Questions:**

1. Can you provide any approximation or rounding guarantee that connects a stationary point on RIM to a discrete indicator solution (or to DSM) with a bounded increase in the objective?
2. How should ([l,u]) be picked in practice without prior labels Do you recommend a data‑driven estimator beyond K‑means, and how sensitive are results across datasets?
3. In Experiment 2, did both manifolds use identical stopping rules and the same retraction and line‑search settings If not, how do the results change when matched?
4. Can authors provide concrete denoising metrics such as PSNR/SSIM/LPIPS/MSE for RIM vs DSM denoising experiments?

---

> ### Author Response · Authors · 2025-11-13
> **Part One**
>
> Hello Reviewer dz1d! 😊
>
> We are very happy to receive your reply. You, along with Reviewers Y5CU and m9ij, are exceptionally thorough and kind reviewers. We look forward to a close 3-week discussion, and I will answer each of your questions and address the weaknesses you've pointed out, one by one, until you are satisfied! We welcome you to raise more questions at any time. Let's work together to make this paper even better! 🧡
>
> ---
>
> > Can you provide any approximation or rounding guarantee that connects a stationary point on RIM to a discrete indicator solution (or to DSM) with a bounded increase in the objective?
>
> We completely agree with your assessment that "A relaxation cannot just be fast, it must actually be a good approximation to the original problem (discrete indicator)." However, optimizing a discrete indicator matrix is typically not only non-convex but also **NP-hard**.
>
> Let's assume the true discrete optimal solution is $ Ind^* $. To find a solution in an acceptable amount of time, we must relax the problem. After relaxation, we at best convert an NP problem to a P problem, but it remains non-convex. Our algorithm can only guarantee finding a *locally* optimal relaxed indicator matrix $ F_1^* $ (in fact, no algorithm can guarantee finding the *globally* optimal solution). Therefore, obtaining a precise bound between $ F_1^* $ and $ Ind^* $ is practically impossible.  Of course, a simple bound is easy, e.g.,
>  $|| F_1^* - Ind^* ||_F^2 < \sum_1^n (1^2+1^2) = 2n$  ,
> but such obvious bounds rarely provide useful insights.
>
> However, **a effective bound relative to the DSM optimization problem *can* be given.** Let's assume we are considering a $\mu$-strongly convex and $L$-Lipschitz optimization problem with respect to DSM (we must assume this, as non-convex problems can have local optima with vastly different properties, making analysis impossible). Let the optimal doubly stochastic matrix be $ F_2^* $, where its row and column sums are all $r$. To speed up computation, we relax the constraint to the RIM manifold, setting the column sums to be between $ l $ and $ u $ ($l \le \text{column sum} \le u$). If we find the globally optimal solution $F_1^*$ on this relaxed manifold, we have:
>
> $$
> \|F_1^*−F_2^∗\|≤\mathcal{O}(\frac{L}{\mu})⋅\max(|r−l|, |u−r|)
> $$
>
> This bound is derived from the **sensitivity theorem of convex relaxation** [1]. We will provide the full proof in the updated PDF.
>
> Furthermore, we can state that our relaxation is **provably better than existing singly stochastic and doubly stochastic relaxations!** This is because if we take the uninformative prior for the RIM manifold, $l = -\infty$ and $u = +\infty$, our constraint becomes equivalent to the singly stochastic manifold. If we set $l=u=r$, it is equivalent to the doubly stochastic manifold. This means the RIM manifold always searches for the optimal solution within a more reasonable constraint space, while also being faster than the doubly stochastic manifold. This is our contribution.
>
> ---
>
> >  How should (l,u) be picked in practice without prior labels? Do you recommend a data-driven estimator beyond K-means, and how sensitive are results across datasets?
>
> This is an excellent question! As we mention in the paper, the purpose of the RIM manifold is to introduce prior information into the model. The absence of priors is indeed a tricky problem.
> * One less-than-ideal method is to set $l = -\infty, u = +\infty$, which defaults to the singly stochastic manifold.
> * Another method, as we suggested, is to use existing data mining algorithms to find priors for the model. In the paper, we recommend using *K-Means** [2] (or other fast algorithms with a balance regularizer) because it is fast and stable.
>
> We can test this with the simple experiment below. Other unsupervised algorithms could also be considered.
>
> | Dataset | (l,u) K-Means | (l,u) selected |
> | :--- | :---: | :---: |
> | COIL20 | (0.41x, 1.47x) | (0.6x, 1.2x) |
> | Digit | (0.52x, 1.66x) | (0.4x, 1.6x) |
> | JAFFE | (0.24x, 1.59x) | (0.4x, 1.6x) |
> | MSRA25 | (0.47x, 1.92x) | (0.4x, 1.6x) |
> | PalmData25 | (0.33x, 1.85x) | (0.4x, 1.8x) |
> | USPS20 | (0.53x, 1.71x) | (0.6x, 2.0x) |
> | Waveform21 | (0.72x, 1.36x) | (0.4x, 1.8x) |
> | MnistData05 | (0.86x, 1.22x) | (0.8x, 1.4x) |

---

> ### Author Response · Authors · 2025-11-13
> **Part Two**
>
> We also provide a supplementary sensitivity analysis showing how the Accuracy (ACC) metric changes with different $(l,u)$ ranges. As you can see, the results are relatively robust to the choice of $(l,u)$. The metric only changes partially when the $(l,u)$ range becomes excessively narrow or wide.
>
> | Data/ACC   | (0.0x, 2.0x) | (0.3x, 1.7x) | (0.5x, 1.5x) | (0.6x, 1.4x) | (0.7x, 1.3x) | (0.8x, 1.4x) | (0.9x, 1.1x) |
> | :--------- | :----------: | :----------: | :----------: | :----------: | :----------: | :----------: | :----------: |
> | Waveform21 |    69.16     |    68.90     |    69.99     |    72.17     |    71.81     |    74.80     |    71.08     |
> | USPS20     |    68.99     |    70.39     |    67.53     |    68.28     |    65.16     |    65.70     |    68.23     |
> | PalmData25 |    87.10     |    86.15     |    87.70     |    86.60     |    91.05     |    89.45     |    91.10     |
> | MSRA25     |    54.92     |    55.09     |    55.86     |    56.97     |    57.42     |    55.59     |    61.09     |
> | JAFFE      |    86.38     |    92.95     |    96.71     |    96.71     |    96.71     |    96.71     |    92.95     |
> | Digit      |    78.46     |    81.52     |    84.14     |    82.47     |    79.74     |    80.80     |    77.90     |
> | COIL20     |    72.92     |    76.94     |    77.99     |    78.13     |    78.68     |    77.22     |    70.49     |
>
> ---
>
> >  In Experiment 2, did both manifolds use identical stopping rules and the same retraction and line-search settings? If not, how do the results change when matched?
>
> Thank you for asking about this. In Experiment 2, we strictly ensured that **all external settings were identical**. The only differences were due to the manifolds themselves. The RIM manifold used the code we provided in the appendix (we provided the complete code), and the doubly stochastic manifold used the code from **Manopt**.
>
> Besides that, the line search, stopping conditions, etc., were all direct calls to the files in `manopt/manopt/solvers`.
>
> Regarding the retraction: all existing doubly stochastic manifolds (in packages like Manopt) uniformly use the **Fisher inner product** and the accompanying **Sinkhorn retraction**. Our RIM manifold innovatively uses the **Euclidean inner product** and the corresponding **Dykstra's retraction**.
>
> Experiment 2 was designed to verify that the RIM manifold is faster than the doubly stochastic manifold. Table 3 already proves that one reason RIM is faster is that Dykstra's retraction is faster. Since Dykstra's retraction is one of our contributions, we believe Experiment 2 is a fair comparison.
>
> ---
>
> > Can authors provide concrete denoising metrics such as PSNR/SSIM/LPIPS/MSE for RIM vs DSM denoising experiments?
>
> We completely agree with you! Denoising was part of our Experiment 2, and its purpose was to answer the question: "When $l = u$, does the Riemannian optimization algorithm on the RIM manifold outperform the Riemannian optimization algorithm on the doubly stochastic manifold in terms of effectiveness and speed?"
>
> Denoising was the objective function we chose for this experiment, which is why we initially only reported the time and final loss. Your suggestion is very reasonable. We provide the various denoising metrics under different noise levels below.
>
> | (noise, $\xi$) | Method  |     MSE      |    PSNR     |    SSIM    |    LPIPS     |
> | :------------- | :-----: | :----------: | :---------: | :--------: | :----------: |
> | (0.3, 0.3)     | **RIM** | **0.011829** | **19.2707** | **0.5019** | **0.561309** |
> | (0.3, 0.3)     |   DSM   |   0.014676   |   18.3340   |   0.4116   |   0.671187   |
> | (0.3, 0.7)     | **RIM** | **0.019672** | **17.0615** | **0.4339** | **0.719043** |
> | (0.3, 0.7)     |   DSM   |   0.027178   |   15.6579   |   0.2557   |   0.928825   |
> | (0.5, 0.3)     | **RIM** | **0.022064** | **16.5631** | **0.2822** | **0.741554** |
> | (0.5, 0.3)     |   DSM   |   0.026411   |   15.7823   |   0.2468   |   0.823607   |
> | (0.5, 0.7)     | **RIM** | **0.023341** | **16.3190** | **0.3266** | **0.775428** |
> | (0.5, 0.7)     |   DSM   |   0.032183   |   14.9237   |   0.2085   |   0.967826   |
> | (0.9, 0.3)     | **RIM** | **0.084558** | **10.7285** | **0.1068** | **0.968648** |
> | (0.9, 0.3)     |   DSM   |   0.105894   |   9.7513    |   0.0962   |   1.020374   |
> | (0.9, 0.7)     | **RIM** | **0.043725** | **13.5927** | **0.1830** | **0.802953** |
> | (0.9, 0.7)     |   DSM   |   0.051412   |   12.8900   |   0.1478   |   1.020374   |
>
> As you can see, the RIM manifold consistently outperforms DSM in terms of these metrics. Please note that the absolute values of these metrics are not the main point (as TV norm is not a state-of-the-art denoising algorithm); rather, **the relative values are what matter**. This experiment was conducted to verify that RIM is better than DSM!
>
> ---

---

> ### Author Response · Authors · 2025-11-13
> **Part Three**
>
> Once we are in agreement, we will add all of this to the paper!
>
> 😀 We know you have put a lot of effort into reviewing this paper, and we are working together to improve it! If you feel that our joint efforts have made the paper stronger, we would be grateful if you would consider raising your score!
>
> **References**
>
> [1] Convex Optimization
>
> [2] Balanced k-Means and Min-Cut Clustering

---

> > ### Comment · Reviewer_dz1d · 2025-11-26
> >
> > I thank the authors for the detailed rebuttal. I think most of my concerns are addressed. I am not fully convinced by the $\mu$-sc $L$-lipschitz assumption in the answer to question 1, but I am not sure what is a reasonable guarantee to ask for outside of convexity.
> >
> > I will raise my score to 6, but will defer more to the opinion of other reviewers since I am not as familiar with this area.

---

> ### Author Response · Authors · 2025-11-26
> **Thank You and Further Discussion**
>
> **First, thank you very much for your support. As you can see, we have put a great deal of effort into this work. The paper is currently on the borderline, and your support means a lot to us—we sincerely appreciate it.**
>
> In addition, we would be very happy to further discuss Question 1 with you (**independently of the score**, as it is genuinely an interesting problem).
>
> To begin with, we can agree on the following point: if we want to bound the distance between $F^\*_1$ and $F^\*_2$ , convexity alone is not sufficient—**strong convexity is essential**. For example, consider a function $\mathcal{H}(F)=1$, which is a constant and convex function. Every point is a global optimum. In this case, there is no meaningful way to bound the distance between the relaxed optimum $ F^\*_1 $ and the original optimum $ F^\*_2 $, since $ F^\*_2 $ can be arbitrarily far away.
>
> However, if we do **not** require bounding $F^\*_1$ and $F^\*_2$ themselves, but only require bounding the values $\mathcal{H}(F^\*_1)$ and $\mathcal{H}(F^\*_2)$, then strong convexity is no longer necessary—convexity alone is enough. A convex function may have infinitely many optimal points, but they all share the same function value, and the optimal set is always “connected” in an appropriate sense.
>
> For a general **non-convex** problem, the situation becomes completely different. For a general $\mathcal{H}(F)$, we often do not even know the global optimum, since in general **no algorithm can reliably find the global solution of a non-convex problem** [1]. What we usually have is only a lower bound, e.g., based on domain-specific knowledge such as $\mathcal{H}(F) > 0$. While this gives some form of bound, such bounds are usually not meaningful enough to provide practical insight—let alone the fact that general graph-cut problems are NP-hard discrete optimization problems.
>
> Therefore, the research community has focused on finding *more meaningful relaxations*, such as the doubly-stochastic relaxation [2], in order to obtain richer structural information about the model. Our work also follows this line of research.
>
> Again, thank you very much for your support. Your reply truly made our day. We hope you are having a wonderful one as well.
>
> [1] Non-convex Optimization for Machine Learning
>
> [2] Graph Cuts with Arbitrary Size Constraints Through Optimal Transport

---

### Official Review · Reviewer_Y5CU · 2025-11-03

**Soundness:** 3
**Presentation:** 2
**Contribution:** 3
**Rating:** 6
**Confidence:** 3

**Summary:**

A new relaxation is introduced for the indicator matrix optimization problem, leading to a manifold with a simple structure. Algorithms are designed based on this manifold.

**Strengths:**

New retractions proposed and a class of efficient manifold algorithms are developed. Extensive numerical experiments have been conducted to validate the effectiveness of the proposed algorithms.

**Weaknesses:**

Presentation can be improved, e.g., "Our Code is presented in Appendix H", "The proof is included in A.5", etc.

**Questions:**

It is claimed after (1) that when $l=u=r$, the relaxation becomes $\\{X | X1_c=1_, X^T1_n=r, X>0\\}$. However, it seems to me that when $l=u=r$, the set in (1) is empty. What the tangent space of the manifold when $l=u$? Can you verify that the proposed retraction is still valid for this case?

---

> ### Author Response · Authors · 2025-11-13
> **Part One**
>
> Hello Reviewer Y5CU! 😀
>
> You are undoubtedly one of the most insightful reviewers for ICLR 2026, demonstrating a true talent for research. In a very brief review, you have pointed out one of the most subtle and ingenious design aspects of the RIM manifold! We are more than happy to explain this further to you!
>
> We mentioned in **Appendix G** that many people are curious about the case $l=u$, which appears to be a singularity. How can the RIM manifold magically operate in this case and ensure the points remain on the doubly stochastic manifold? We gave a brief explanation in Appendix G and are delighted to elaborate for you here!
>
> Let's look at why our method can run successfully when $l=u$ and what set it is actually operating on. We know that the two most important structures in manifold optimization are the **Riemannian gradient** and the **Retraction**.
>
> 1.  **Riemannian Gradient:** Regardless of whether $l=u$, we can always compute Equation (2)(project the gradient to tangent space). This is, of course, mathematically valid, as it simply involves a row-wise normalization of the gradient.
>
> 2.  **Retraction:** Please pay close attention to the projection onto $\Omega_2$ in our Dykstra's algorithm, Equation (7). Note that this uses the $\le$ symbol, not $<$. In practice, we are always projecting onto the set $\{X | X \ge 0, X\mathbf{1}_n = \mathbf{1}_n, l \le X^T\mathbf{1}_n \le u\}$.
>     * When $l \ne u$, this has no real impact. We can consider a tiny quantity $\epsilon$ such that $l \le X^T\mathbf{1}_n \le u \Leftrightarrow l-\epsilon < X^T\mathbf{1}_n < u+\epsilon$. We usually don't worry about whether the inequality is strict.  Even if we strictly require $ l < X^\top \mathbf{1}_n < u $, we only need to define the RIM manifold as
> $$
> X \mid X > 0, X\mathbf{1}_n = \mathbf{1}_n, l+\varepsilon < X^\top\mathbf{1}_n < u-\varepsilon
> $$
> By projecting onto $ l+\varepsilon \le X^\top\mathbf{1}_n \le u-\varepsilon $ as described in our paper, we can ensure that $ l < X^\top\mathbf{1}_n < u $, with $\varepsilon$ chosen arbitrarily small.
>
> * However, when $l=u=r$, our retraction is equivalent to projecting onto $\{X | X \ge 0, X\mathbf{1}_n = \mathbf{1}_n, r \le X^T\mathbf{1}_n \le r\}$. **This is the key** to why we can operate on this "singularity."
>
> In this special case ($l=u=r$), our algorithm is equivalent to a **projection method with a specific gradient correction**. This correction ensures that during optimization, no components are generated in meaningless directions (i.e., non-tangential directions). This is one of the reasons why the RIM manifold converges faster!
>
> ---
>
> Without a doubt, your insight is one of the best at ICLR 2026. Your and m9ij's perceptiveness, along with dz1d's rigor, make you all strong contenders for an **Outstanding Reviewer** award. I recommend that the AC nominate you,  dz1d and m9ij  for your professionalism (if that award is still given this year).
>
> We hope this has answered your questions. Thank you for your support! If we have resolved your concerns, we would be grateful if you would consider raising your score.

---

> > ### Author Response · Authors · 2025-11-18
> > **An easy experiment**
> >
> > We wish to further alleviate your concern with one more simple experiment to verify our algorithm's effectiveness at a singularity. We again consider the simple objective:
> > $$
> > \min_X \|X-B\|^2, \quad X \in \mathcal{M}
> > $$
> > where  $B =[1,0 ; 1,0;1, 0]$. $X$ is on the RIM manifold, but this time we set the singularity condition $l=1.5$ and $u=1.5$.
> >
> > You were concerned:
> > > What is the tangent space of the manifold when l=u? Can you verify that the proposed retraction is still valid for this case?
> >
> > First, we inspected the tangent space. We display a random tangent vector using `disp(problem.M.randvec())`:
> > ```
> >    -0.5879    0.5879
> >     0.5199   -0.5199
> >    -0.0671    0.0671
> > ```
> > As you can see, at this singularity, we maintain the same tangent space definition as in the non-singular case. The tangent vectors (like the one above) correctly satisfy the row-sum constraint (i.e., their row sums are 0).
> >
> > Furthermore, our optimization algorithm (which relies on the retraction) successfully converges. The output is:
> >
> > ```
> > Optimized X =
> >      0.5000    0.5000
> >      0.5000    0.5000
> >      0.5000    0.5000
> >
> > Final cost = 0.750000
> > Column sums:
> >      1.5000    1.5000
> > Row sums:
> >      1
> >      1
> >      1
> > ```
> > This result shows that our algorithm effectively finds the optimal solution, converging precisely to the singularity boundary ($l=u=1.5$) while perfectly maintaining the row-sum constraints. This confirms that our retraction remains valid.
> >
> > We hope this simple experiment can fully resolve your concern.
> >
> > For full transparency, the code for this experiment is provided below:
> > ```matlab
> > %% RIM Boundary Verification Test -- Singularity (l=u)
> > clc; clear; close all;
> >
> > n = 3;
> > c = 2;
> >
> > % Target matrix B
> > B = [1 0;
> >      1 0;
> >      1 0];
> >
> >
> > % RIM constraints
> > row_sum = ones(n,1);   % rows must sum to 1
> > l = 1.5*ones(c,1);     % Lower bound
> > u = 1.5*ones(c,1);     % Upper bound (l=u)
> >
> > % Build manifold
> > manifold = Relaxd_Indicator_Matrix_factory(n, c, row_sum, u, l);
> >
> > % Starting point
> > X0 = manifold.rand();
> >
> > % Quadratic objective
> > problem.M = manifold;
> > problem.cost  = @(X) 0.5 * norm(X - B, 'fro')^2;
> > problem.egrad = @(X) (X - B);
> >
> > % Solve
> > [Xopt, costopt, info] = steepestdescent(problem, X0);
> >
> > % Results
> > disp('Random tangent vector:');
> > disp(problem.M.randvec()); % Display a random tangent vector
> >
> > disp("Optimized X = ");
> > disp(Xopt)
> > fprintf("Final cost = %.6f\n", costopt);
> > disp("Column sums: "); disp(sum(X_opt, 1));
> > disp("Row sums: "); disp(sum(X_opt, 2));
> > ```

---

### Author Response · Authors · 2025-11-13
**Thank all reviewers**

Dear AC and PC,

We sincerely thank all reviewers for their thorough and constructive feedback. We appreciate the opportunity to clarify our work and address the points raised.

We have provided detailed responses to each reviewer, including new experimental results and theoretical clarifications, which we believe fully address their concerns. We are confident that these clarifications will resolve any misunderstandings and demonstrate the contribution of our paper. It was truly a fantastic experience.

Many thanks again to Y5CU and m9ij for their unique insights and academic discoveries, and to dz1d for rigor and pragmatism. All of this left a deep impression on us. They are definitely top reviewers.

We look forward to a productive discussion.

---

### Author Response · Authors · 2025-11-17
**Authors' Response to Reviews**

Dear Reviewers,

We sincerely thank you for your insightful comments and valuable feedback on our paper. We have thoroughly addressed all points raised and have submitted a revised manuscript. All major revisions are marked in green in the updated PDF.

Below are our point-by-point responses:

### **For Reviewer Y5CU:**

  * We have carefully revised the manuscript's presentation. For instance, Line 24 now reads, "Our code is presented in Appendix H," and the proof statement is, "The proof is included in A.1."
  * In Lines 192-195, we have briefly addressed the concern raised by both you and Reviewer m9ij, and we have directed readers to Appendix G for a detailed discussion.

### **For Reviewer dz1d:**

  * We now point to our analysis of the optimal solution relationship in the main paper (Lines 317-319), with the full proof provided in the appendix (Lines 1748-1780).
  * We have added supplementary experiments and a sensitivity analysis for $(l, u)$ in Lines 2461-2490.
  * In Lines 369-371, we clarify that our other conditions remain consistent with those in RIM and DSM.
  * We have supplemented the description of the denoising process evaluation (Lines 429-451) and provided the final results (Lines 487-491).

### **For Reviewer m9ij:**

  * In Lines 192-195, we acknowledge the boundary point issue you raised and point to a detailed discussion in Appendix G. We have also added a "Limitation" section (Lines 470-478) to clarify that an $\epsilon$-correction is needed for strict boundary requirements.
  * We have corrected the time complexity for Stiefel manifold optimization (Lines 38-39, 107-108) and included the correct citation.
  * We have added the appropriate citation for the definition of "retraction" (Lines 160-161). We also cite projection-based retractions and discuss your related query in Lines 192-195.
  * We have relabeled the original "Theorems 1, 2, 3, and 4" as "Lemmas 1, 2, 3, and 4," respectively.

-----

We once again thank all reviewers for their time and effort in reviewing our manuscript. We hope our revisions have satisfactorily addressed all concerns and look forward to your re-evaluation.

---

> ### Author Response · Authors · 2025-11-27
> **Revision 2.0**
>
> To further address the inquiries from Reviewers Y5CU and m9ij, we have expanded Appendix G into a dedicated full page, located on page 53. As before, these revisions are highlighted in green, and we look forward to your further feedback.
>
> We deeply appreciate the significant effort the three reviewers have invested in this paper, we intend to acknowledge their contributions in the Acknowledgments section of the RIM manifold paper and our project homepage.

---

### Author Response · Authors · 2025-11-29
**Summary: We are ready to share our work**

Dear AC, SAC, and PC,

As a brief summary, we would first like to sincerely thank you and all reviewers for your time, efforts, and constructive feedback. Our paper makes the following contributions:

1. **We propose the RIM manifold**, which incorporates class information into the underlying geometric structure of machine learning systems. This represents a **conceptual advance** compared with single-stochastic or double-stochastic relaxations.

2. **We accelerate double-stochastic manifolds from (\mathcal{O}(n^3)) to (\mathcal{O}(nc))**, achieving a significant and practically meaningful speedup.

3. **We develop a complete RIM optimization toolbox**, fully compatible with Manopt, with “one-click” runnable code to support reproducibility and inspire follow-up research.

4. **We conduct four extensive experiments**, including cases with **over ten million variables(10000000)**, to thoroughly validate our method.

5. **We provide nearly 60 pages** of detailed theoretical development and application discussion.

6. **We dedicate an entire Appendix (G)** to analyzing potential singularities of the algorithm and experimentally verifying its robustness.

Our work has already received **initial positive feedback from the community**. We believe that a contribution combining **theoretical innovation**, **large-scale experimental validation**, and **reproducible open-source tooling** represents a *pretty solid paper even by ICLR standards*.

We are confident that we can address any additional concerns you may have in the camera-ready version. We would be excited to share this work with the Riemannian optimization and machine learning communities at the conference, and we hope you will give our submission your serious consideration.

---

### Meta-Review · Area_Chair_Ksam · 2026-01-04

**Summary:**

This paper introduces the RIM (Relaxed Indicator Matrix) manifold as a generalization of the doubly stochastic and single stochastic manifolds. It follows the standard arguments in Riemannian optimization by providing Riemannian geometric computations.

There are three reviewers. All reviewers acknowledge the novel contributions. Most concerns are well addressed by the rebuttal. Specifically, Reviewer dz1d mentioned the possibility that he/she would raise his/her score to 6. In summary, I think this paper generally meets the ICLR threshold.

**Reviewer Concerns:**

Most concerns are well addressed by the authors, and they provide detailed responses to these rebuttals.

**Reviewer Scores:**

Reviewer dz1d explicitly mentioned the possibility that he/she would raise his/her score to 6. In addition, I think Reviewer m9ij might increase his/her score if he/she had been able to participate fully in the discussion.

---

### Decision · Program_Chairs · 2026-01-26

Accept (Poster)